# Delocalized electronic engineering of TiNb₂O₇ enables low temperature capability for high-areal-capacity lithium-ion batteries

Yan Zhang[1,4], Yingjie Wang[2,4], Wei Zhao[1], Pengjian Zuo [ORCID][1], Yujin Tong[3], Geping Yin [ORCID][1] ✉, Tong Zhu [ORCID][2] ✉ & Shuaifeng Lou [ORCID][1] ✉

High areal capacity and low-temperature ability are critical for lithium-ion batteries (LIBs). However, the practical operation is seriously impeded by the sluggish rates of mass and charge transfer. Herein, the active electronic states of TiNb₂O₇ material is modulated by dopant and O-vacancies for enhanced low-temperature dynamics. Femtosecond laser-based transient absorption spectroscopy is employed to depict carrier dynamics of TiNb₂O₇, which verifies the localized structure polarization accounting for reduced transport overpotential, facilitated electron/ion transport, and improved Li⁺ adsorption. At high-mass loading of 10 mg cm⁻² and −30 °C, TNO$_{-x}$@N microflowers exhibit stable cycling performance with 92.9% capacity retention over 250 cycles at 1 C (1.0-3.0 V, 1 C = 250 mA g⁻¹). Even at −40 °C, a competitive areal capacity of 1.32 mAh cm⁻² can be achieved. Such a fundamental understanding of the intrinsic structure-function put forward a rational viewpoint for designing high-areal-capacity batteries in cold regions.

High areal capacity is essential for commercial batteries with high energy density, which requires a high mass loading of active components on the electrode[1,2]. Nevertheless, thick electrodes usually cause a proportional increase in the ions/electrons transport distance and resistance, accompanied by high polarization and low utilization[3–6]. As the temperature falls below 0 °C, these issues are further aggravated, arising from the decreased ion conductivity of electrolytes, restricted desolvation processes, and reduced solid-phase diffusion of Li⁺ in the active materials, resulting in a reduction in cell capacity and power[7,8]. Particularly, the fast-charging capability of conventional graphite presents obvious decreases under low temperatures and high rate conditions due to the huge polarization and even propels the intercalation potential of the graphite anode below 0 V (*vs*. Li⁺/Li), which makes it difficult to charge the batteries limited by the cut-off voltage. A more critical concern is that the lithium plating on the graphite electrode tends to become "dead lithium" due to the lithium dendrite damage, leading to the loss of active lithium and rapid capacity decay, rendering the

practical application of high-energy-density batteries in cold regions challenging[9,10].

Niobium-based oxides (TiNb₂O₇), as a potential anode electrode with the potential of 1.65 V *vs*. lithium anodes, have received extensive attention owing to the safe lithiation potential and suitable channels for Li⁺ migration related to the defective ReO₃-like structure[11–13]. Despite great progress in the fast-charging ability, limited areal capacity and low-T performance (below −30 °C) are still critical concerns. The large bandgap of TiNb₂O₇ gives rise to large charge transfer barriers during the first insertion of lithium ions, resulting in an insufficient ability to accommodate the fast-passing Li⁺ during the solid solution transformation process, especially under low temperatures[14,15]. Meanwhile, the poor electronic conductivity can cause the electrode potential to deviate from the equilibrium potential during cycling, generating a larger voltage polarization (especially at fast-charging conditions), which will decrease the effective potential and reduce the available Li ions. In general, the apparent Li⁺ diffusion rate depends on the intrinsic structure and particle size[16]. Hence, nano-engineering is typically used to shorten

[1]State Key Laboratory of Space Power-Sources, Harbin Institute of Technology, Harbin, China. [2]Laser Micro/Nano Fabrication Laboratory, School of Mechanical Engineering, Beijing Institute of Technology, Beijing, China. [3]Faculty of Physics, University of Duisburg-Essen, Duisburg, Germany. [4]These authors contributed equally: Yan Zhang, Yingjie Wang. ✉e-mail: yingeping@hit.edu.cn; Tongzhubit@bit.edu.cn; shuaifeng.lou@hit.edu.cn

the ionic transfer distance and increase the solid-liquid interfaces for sufficient Li$^+$ insertion (nano-effects). Nevertheless, a major drawback of nanoparticles is the low tap density, which reduces the packing efficiency of the electrode material, yielding poor volumetric energy density[17,18]. Therefore, exploiting a practical route to enhance the charge-transfer kinetics of micron-sized TiNb$_2$O$_7$ for high-areal-capacity batteries under low temperatures is urgently desired.

Micron-scale TiNb$_2$O$_7$ composed of compacted nanosheets can circumvent the problems introduced by nanoparticles and retain the nano-effects for favorable low-temperature performance[19]. The introduction of dopants with larger atomic sizes and lower electronegativity into the lattice is an effective approach to adjusting the local electronic states, which is predicted to produce impurity bands between the valence and conduction bands, facilitating the generation of holes and free electrons, thereby improving electronic conductivity. Beyond that, anionic defects are precisely introduced by in situ doping, which can act as driving sites to enhance the low-temperature dynamics for Li$^+$ diffusion during solid-solution transformation[20,21]. To the best of our knowledge, the Li-storage behavior of the high-mass-loading Ti$_x$Nb$_y$O$_z$-based electrodes below −30 °C has not yet been reported; and also, the correlation between the intrinsic electronic structure and low-temperature properties remains unclear.

Herein, we proposed delocalized electronic engineering to manipulate the active electronic states in TiNb$_2$O$_{7-x}$@N (TNO$_{-x}$@N) microflowers to boost ion-diffusion kinetics at low temperatures (Fig. 1). Such compact nano-structured electrode materials can achieve rational tap density and provide abundant storage sites for fast Li$^+$ transport. DFT calculations further demonstrated that the modulation of local electronic properties can facilitate electron transport, ion diffusion, and Li$^+$ adsorption, endowing high Li$^+$ diffusion coefficients and fast-charging capability at low temperatures. It is important to emphasize that, the results from transient absorption spectroscopy directly verify the theoretical calculations, which established the relationship between the anion engineering, energy band structure, and intrinsic electronic conductivity. Benefitting from this, the TNO$_{-x}$@N microflowers delivered excellent rate capability (147.6 mAh g$^{-1}$ at 6 C) and high durability with a low capacity decline rate of 0.028% per cycle over 250 cycles under high mass loading and −30 °C. Even when the temperature drops to −40 °C, a high areal capacity of 1.32 mA h cm$^{-2}$ (12 mg cm$^{-2}$) was maintained, demonstrating potential application for fast-charging batteries in cold regions.

## Results

### Morphology, structure and kinetics of the TNO$_{-x}$@N

The TNO$_{-x}$@N microflowers materials with gradient anion doping and rich defects are prepared by hydrothermal reaction and followed by nitriding treatment, as presented in Fig. 2a. X-ray diffraction (XRD) is employed to identify the crystalline structures (Fig. 2b), and all Bragg diffraction peaks of TNO$_{-x}$@N can be well indexed as the monoclinic phase of TiNb$_2$O$_7$ (JCDPS No. 72-0116). Notably, the main peaks of TNO$_{-x}$@N are slightly shifted to a lower angle compared to that of TNO, and the Rietveld refinements of XRD reveal that the unit cell volume of TNO$_{-x}$@N (796.175 Å$^3$) towards larger than TNO (795.316 Å$^3$), which is ascribed to the increase in interplanar spacing induced by anion doping[22]. Further analysis based on Raman spectra confirms the XRD results. The stretching vibration bands of TNO$_{-x}$@N are negatively shifted to the low-frequency range compared to TNO, indicating the decreased symmetry of the Nb/Ti−O bonds[23] (Supplementary Fig. 1).

X-ray photoelectron spectroscopy (XPS) measurements are performed to explore the surface chemical valence of TNO$_{-x}$@N (Supplementary Fig. 2). The high-resolution N 1$s$ spectra located at 399.1 eV (Supplementary Fig. 3) confirms the valid nitrogen doping. For the high-resolution Nb 3$d$ spectrum of TNO$_{-x}$@N, two peaks at binding energies of 209.7 and 206.9 eV are assigned to Nb$^{5+}$, while two additional small peaks attributed to the Nb$^{4+}$ can be also observed at 209.1 and 206.0 eV[24] (Supplementary Fig. 4a). In Supplementary Fig. 4b, two additional peak at 463.4 and 457.3 eV are detected in the Ti 2$p$ spectra of TNO$_{-x}$@N in comparison to TNO, corresponding to the characteristic positions of Ti$^{3+}$ cations[25]. Compared to the TNO material, the peak position in the Nb 3$d$ spectra of TNO$_{-x}$@N shows an obvious blue shift of about 0.2 eV, which may be the presence of charge redistribution[26,27]. Similar shift trend is also presented in the Ti 2$p$ XPS spectra. X−ray absorption near−edge structure of TNO and TNO$_{-x}$@N are examined to confirm the existence of energy shift. In Fig. 2c, the XANES profile of TNO$_{-x}$@N lies between Nb foil and TNO, which indicates an average oxidation state of Nb that is intermediate between the two reference samples, further suggesting that an enhanced electron redistribution occurs after the ammonia annealing treatment. The O 1$s$ spectrum (Supplementary Fig. 4c) in TNO$_{-x}$@N can be divided into three peaks located at 529.6, 531.2, and 532.7 eV, belonging to lattice O (metal−oxygen bands), oxygen vacancies, and oxygen species in hydroxyl oxygen, respectively[28]. The peak area ratio of O vacancy to lattice O for TNO$_{-x}$@N (0.24) is much higher than that of TNO (0.12),

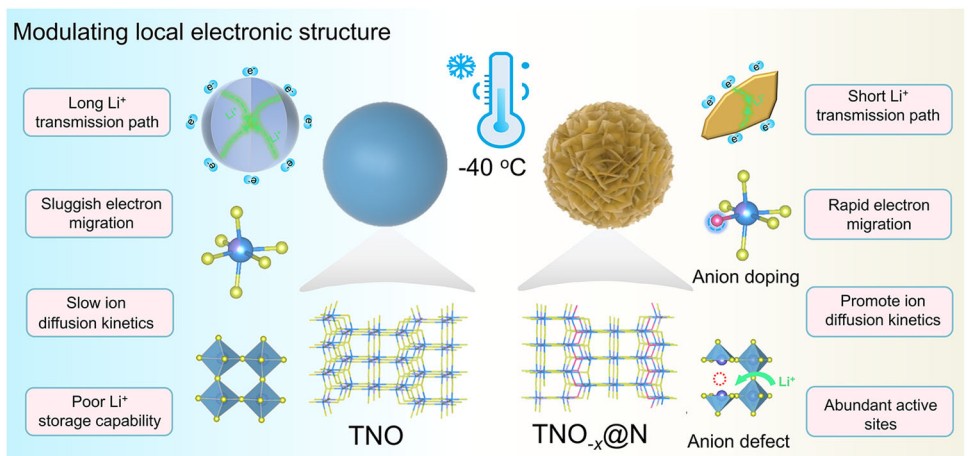

**Fig. 1 | Schematic illustration of the multiple improvement of TNO$_{-x}$@N compared with bulk TNO operated at low temperatures.** The diagram reveal the different lithation kinetics mechanisms between TNO and TNO$_{-x}$@N. For bulk TNO, sluggish apparent Li$^+$ diffusion rate and poor intrinsic electronic conductivity are obstacles for high-energy-density batteries in cold regions. For the TNO$_{-x}$@N microflowers, the synergistic effect of N-incorporation and O-vacancies can induce localized structure polarization, improving electron transfer, reducing Li$^+$ diffusion energy barriers, and promoting Li$^+$ adsorption at low-temperature conditions.

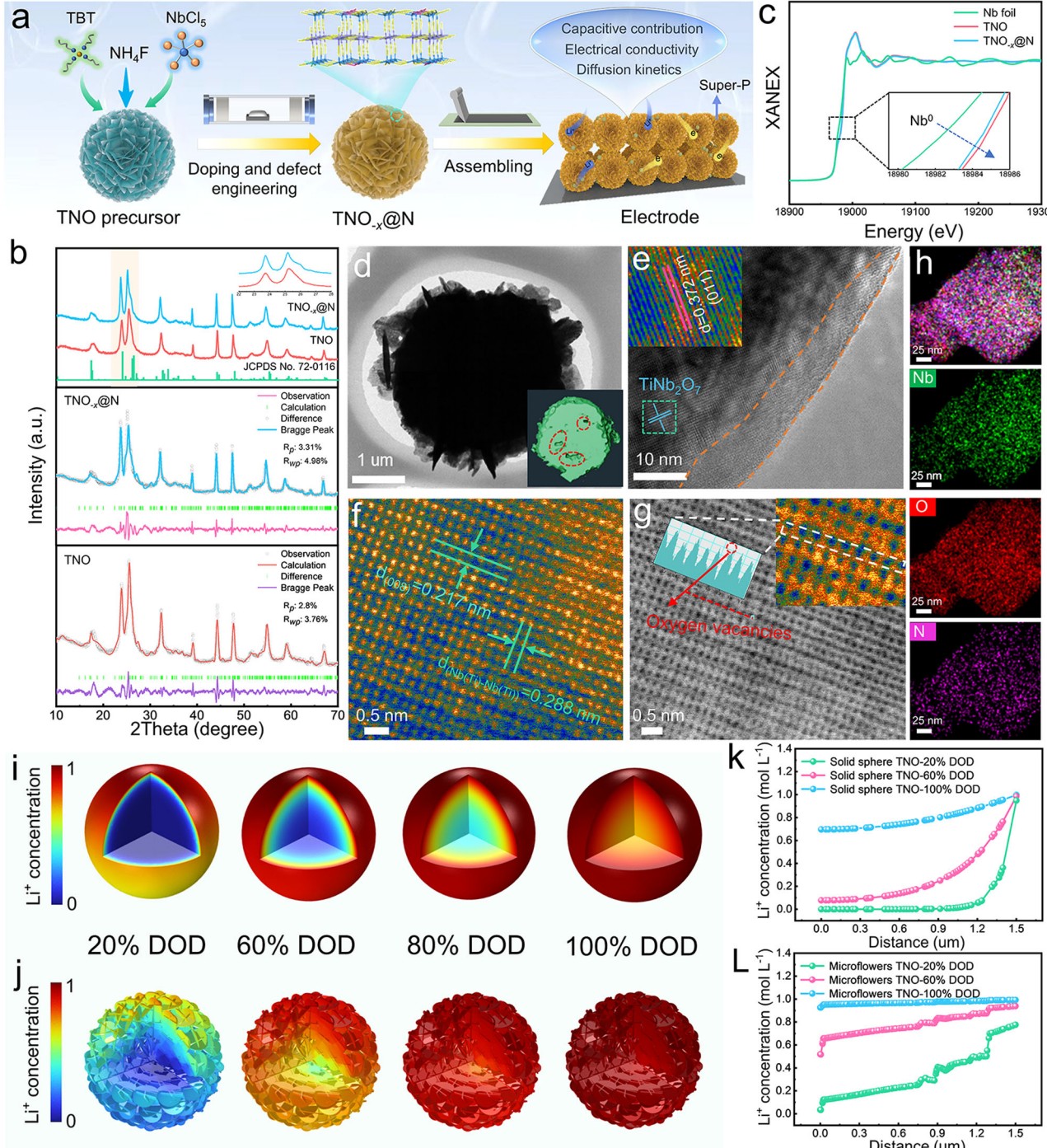

**Fig. 2 | Fabrication, microstructure analysis and lithiation simulation of the TNO$_{-x}$@N composite. a** Schematic diagram of the synthesis process of TNO$_{-x}$@N. **b** Powder XRD and Refined XRD. **c** Nb K-edge XANES of Nb foil, TNO, and TNO$_{-x}$@N. **d** TEM image of TNO$_{-x}$@N (the inset demonstrates the 3D image of TNO$_{-x}$@N by Nano-CT technology). **e** HR-TEM image of TNO$_{-x}$@N. **f** Aberration-corrected STEM-HADDF image of the TNO$_{-x}$@N. **g** Oxygen defect analysis with ABF image of the TNO$_{-x}$@N. **h** Atomic-resolution EDS mappings of the TNO$_{-x}$@N. **i, j** Finite element simulation models of Li$^+$ concentration of solid sphere (**I**) and microflowers (**j**) for different lithiation states at the current density of 5 C. **k, l** Evolution of Li$^+$ concentration in the solid sphere (**k**) and microflowers (**l**).

confirming the introduction of oxygen vacancies in TNO$_{-x}$@N. Electron paramagnetic resonance (EPR) measurements are carried out to further detect the oxygen defects. As shown in Supplementary Fig. 5, TNO$_{-x}$@N shows a distinct EPR signal peaks at g = 2.003, whereas TNO exhibits a slight resonance. This result strongly proves that the thermal reduction process due to the ammonia annealing treatment leads to the appearance of bulk-phase oxygen vacancies in TNO$_{-x}$@N in addition to surface vacancies.

Morphology and structural features are investigated by scanning electron microscopy (SEM) and transmission electron microscope (TEM). The as-prepared TNO precursor exhibits a rose-like structure assembled by smooth nanosheets with an average size of 2~3 μm (Supplementary Fig. 6). After thermal treatment, the TNO sample preserves the 3D structural feature (Supplementary Fig. 7). However, the surface of TNO$_{-x}$@N micro flowers becomes rough after nitridation because of the etching effect of the thermal NH$_3$ atmosphere

(Supplementary Fig. 8). The TEM images in Fig. 2d show that the assembled nanosheets are irregular and dense (size, ~300 nm), which is beneficial for increasing the tap density of the electrodes. In addition, advanced synchrotron nano-tomography is employed to investigate the morphological and structural of 2D slice of TNO$_{-x}$@N composite (Fig. 2d inset). The results further confirmed the dense structure of the microspheres. From the magnified TEM image of TNO$_{-x}$@N (Supplementary Fig. 9), it can be monitored that rich mesopores are formed on the surface of the nanosheets. According to the Barrett–Joyner−Halenda (BJH) results the pore size distributions of TNO$_{-x}$@N are mainly concentrated at around 20-40 nm, further proving a mesoporous structure (Supplementary Fig. 10). In addition, the Brunauer−Emmett−Teller (BET) results exhibit that the specific surface area of TNO$_{-x}$@N micro flowers reaches 26.42 m$^2$ g$^{-1}$, which is larger than the TNO (20.76 m$^2$ g$^{-1}$) and solid TNO (1.20 m$^2$ g$^{-1}$). Such a striking feature provides a large surface-to-volume ratio compared to TNO and bulk samples (bulk morphology in Supplementary Fig. 11), enabling abundant Li$^+$ storage sites and excellent transport ability in the TNO$_{-x}$@N sample.

High-resolution TEM (HRTEM) image of TNO$_{-x}$@N shows the lattice spacing of 0.372 nm, corresponding to (011) crystal planes of the monoclinic TiNb$_2$O$_7$ (Fig. 2e). Note that an amorphous layer with a thickness of ~8 nm can be observed in the edge region, which may be attributed to the partial nitriding of the surface TNO material due to the outward-inward expansion of the NH$_3$ heat treatment[29,30]. Meanwhile, there are no obvious characteristic peaks in the XRD pattern in Fig. 2b, which may be attributed to the low nitride content and the generation of amorphous nitrides. The atomic-level vacancy structure of TNO$_{-x}$@N is visually examined by high-angle annular dark-field scanning transmission electron microscopy (HAADF-STEM). In Fig. 2f, the presence of three-dimensional and interconnected cation active sites is observed and confirmed in the TiNb$_2$O$_7$ crystalline structure, which is conducive to serving as a high-flux and interconnected transport pathway. Further, the O-vacancies of TNO$_{-x}$@N can be directly visualized via annular bright-field (ABF) imaging. As shown in Fig. 2g, the area marked with faint dark spots indicates the presence of O-vacancies, which can be further identified by the corresponding intensity line profiles derived from the contrast differences (Fig. 2g inset). The corresponding element mapping presents a homogeneous distribution of Nb, Ti, N, and O in TNO$_{-x}$@N microflowers (Fig. 2h and Supplementary Fig. 12).

The rational structural design of electrode materials with rapid transport kinetics for both Li$^+$ and electrons is a key strategy to improve low-temperature performance. The nanosheet-assembled compact structure possesses short distance and large electrode-electrolyte contact area for increased Li$^+$ flux per unit area. Inspired by this, the Finite element simulations are conducted to capture the real-time Li$^+$ concentration distribution between the solid sphere (R = 1.5 um) and microflowers (R = 1.5 um, L = 300 nm, W = 400 nm, δ = 10 nm) upon the lithiation (5 and 15 C discharge) process, where R, L, W, δ are the radius of a model particle, the length, the width, and the thickness of each nanosheet, respectively. As shown in Fig. 2i and Supplementary Fig. 13, the Li$^+$ concentration in the solid spheres steadily increases from the outside to the inside during lithiation on account of the lithium diffusion inwards from the entire periphery. Particle underutilization caused by non-uniform Li$^+$ concentration appears even at 100% depth-of-discharge (DOD) state, due to the low active surface area and restricted transport channels of the solid spheres. Figure 2j shows that the Li$^+$ concentration of the microflowers models is more homogeneous compared to the solid spheres models during discharging, suggesting that the nanosheet-assembled compact microflowers can reduce the Li$^+$ diffusion distance and increase the contact area of the solid-liquid interfaces, resulting in a faster ion-diffusion kinetics. Meanwhile, the inhomogeneity of the particles is more pronounced at the higher current densities. Such results are

further verified by the evolution of the Li$^+$ concentration distribution achieved along the radial direction (Fig. 2k, I and Supplementary Fig. 14).

To depict the real kinetics behavior of the two model electrodes for LIBs, the Li$^+$ concentration variations are further simulated at the electrode-level models with different depths of discharge (Supplementary Fig. 15). Taking the initial (40% DOD) and full depth of discharge (100% DOD) as two key nodes, the Li$^+$ concentration presents a gradient feature along the depth direction of the electrolyte. Typically, the reaction at high C-rates is relatively fast so that the sluggish mass transport in the 3D pore phase cannot deliver the Li ions into the deep electrode at a sufficient rate[31]. At 100% DOD state, the bottom layer (near the current collector) of the solid sphere electrode is not lithiated fully. In contrast, the microflowers electrode allows for a uniform lithiation state. These simulation results highlight the tremendous ion diffusion and storage capability in the compact nanostructure, which is expected to achieve superior low-temperature performance.

## Theoretical insight into the electron delocalization effect

Density functional theory (DFT) calculations are performed to understand the N doping preference position and the delocalized electronic engineering on the kinetics nature of TNO$_{-x}$@N. As shown in Supplementary Table 1, the substitution energy is lowest when it occupies the tetra-coordination site (Supplementary Fig. 16 and Fig. 3a), indicating that the tetra-coordination is probably the N-doped sites in the TNO$_{-x}$@N sample. Meanwhile, various O-vacancy concentration models (varying from 1% to 9%, denoted as Ti$_{18}$Nb$_{36}$N$_{12}$O$_{114-x}$) are used to clarify how anion defects contribute to the electronic structure. Supplementary Fig. 17 shows the locations where oxygen defects are likely to form. As illustrated in Fig. 3b and Supplementary Table 2, the average O-vacancy formation energy ($E_{vac}$) is positively correlated with the number of O-vacancies. Notably, it is more energetically favorable for the formation of O-vacancies near the triple-coordinated Ti site (1.43 eV, Ti$_{18}$Nb$_{36}$N$_{12}$O$_{113}$) compared to the Nb site (3.9 eV), manifesting that Ti$^{4+}$ is more prone to be reduced. The drastically increased average O-vacancy formation energy (>2.2 eV) triggers a relatively unstable state when more than six O-vacancies are present in the model unit of Ti$_{18}$Nb$_{36}$N$_{12}$O$_{114-x}$.

The corresponding electron localization functions (ELF) of TNO$_{-x}$@N and TNO (Fig. 3c and Supplementary Fig. 18) show that the electron delocalization is caused by the weak interactions between the TiNb$_2$O$_7$ and N-terminal with low electronegativity, indicating that N atoms have a weak electron absorption ability, and the electron localization is weaker and more free electrons are generated in TNO$_{-x}$@N, which may be conducive to the improvement of the electrical conductivity of TNO materials and reduce the initial reaction barrier[32,33]. The density of states (DOS), as a basic concept describing the number of electron states, is utilized to examine the electronic structure of samples (Fig. 3d). For the valence band of the perfect TNO, the strong hybridization between Ni$_{3d}$-O$_{2p}$ and Ti$_{3d}$-O$_{2p}$ is revealed by their obvious overlap and consistent amplitudes. In addition, the calculated TNO bandgap of around 2.2 eV indicates an insulating nature. By contrast, the electrical conductivity of TNO$_{-x}$@N can be significantly improved because of the appearance of the impurity bands between the valence and conduction band by N-incorporation, leading to an increase in carrier concentration. In addition, the Ultraviolet−visible diffuse reflectance spectra (UV−vis DRS) are carried out to further estimate the bandgap values of the TNO and TNO$_{-x}$@N. Based on the Tauc plots based on the Kubelka−Munk equation, as shown in Supplementary Fig. 19, the bandgap values of TNO and TNO$_{-x}$@N are 2.95 and 2.62 eV, respectively. This agrees well with the DFT calculation, but this value is larger than the bandgap calculated by the DFT, which is mainly attributed to the fact that the DFT theory itself tends to underestimate the energy gap[34]. The electronic conductivities of the TNO and TNO$_{-x}$@N powders are evaluated by the chronoamperometry

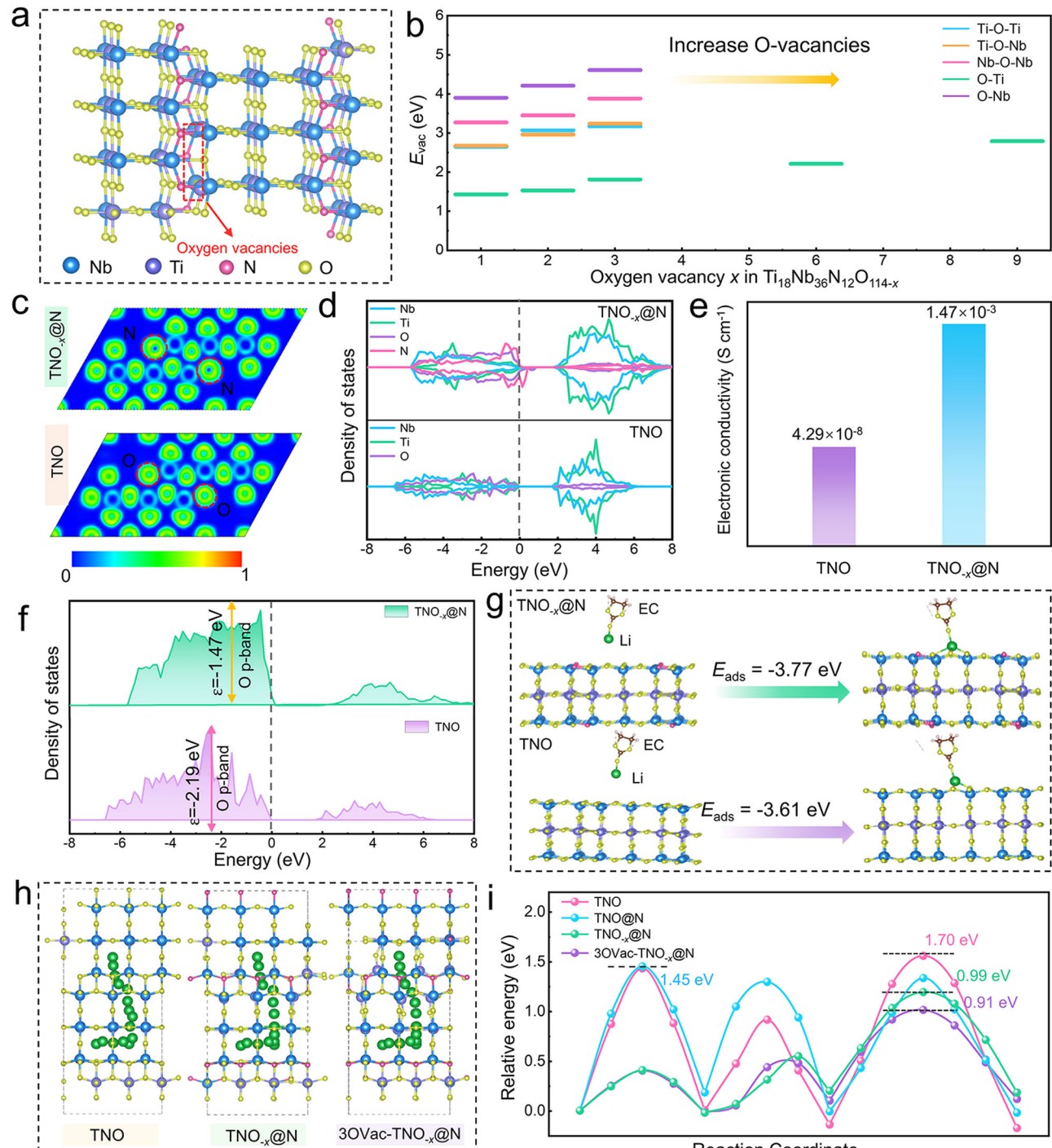

**Fig. 3 | Electron delocalization analysis and effects on Li⁺ transport. a** The optimized structure of TNO$_{-x}$@N. **b** The dependent relationship of the formation energy of O-vacancy with concentration and position in TNO$_{-x}$@N structure. **c** The corresponding ELF plots of TNO$_{-x}$@N and TNO. **d** The calculated DOS of TNO$_{-x}$@N and TNO. **e** The electronic conductivities of TNO and TNO$_{-x}$@N. **f** PDOS values of the O $p$ orbitals of TNO$_{-x}$@N and TNO. **g** Interfacial Li⁺ adsorption energy (the yellow, purple, pink, green and blue balls represent O, Ti, N, Li and Nb atoms, respectively). **h** Li⁺ migration pathways of TNO, TNO$_{-x}$@N, and 3OVac-TNO$_{-x}$@N. **i** The migration energy barriers in the TNO, TNO@N, TNO$_{-x}$@N, and 3OVac-TNO$_{-x}$@N.

(CA) method (Supplementary Fig. 20). As shown in Fig. 3e, the electrical conductivity of TNO$_{-x}$@N is $1.47 \times 10^{-3}$ S cm$^{-1}$, nearly 5 orders of magnitude higher than that of TNO ($4.29 \times 10^{-8}$ S cm$^{-1}$).

To further determine the band-structure relationship, the $p$-band centers of the PDOS of O in TNO and TNO$_{-x}$@N are calculated. In Fig. 3f, the $p$-band center energy of TNO$_{-x}$@N ($-1.47$ eV) exhibits an up-shift trend with respect to TNO ($-2.19$ eV), suggesting that the local electron redistribution can enhance the interaction with

adsorbate. Furthermore, the corresponding adsorption energies of Li⁺ on TNO$_{-x}$@N and TNO are calculated, as depicted in Fig. 3g. The binding energy ($E_{ads}$) between the TNO$_{-x}$@N surface and solvated Li-EC ($-3.77$ eV) is higher than that of TNO ($-3.61$ eV), signifying that the anion engineering can provide effective adsorption for contributing to the surface-induced pseudocapacitive charge storage[20].

As another key evaluation index reflecting the mass transport, Li⁺ transport pathways (Fig. 3h) and the corresponding diffusion energy

barriers (Fig. 3i) are also calculated. It can be seen that the relatively low diffusion barriers of TNO@N manifests the positive role of N-incorporation for Li⁺ migration. Particularly, Li⁺ diffusion energy barriers decrease with the increase in oxygen deficiency in $Ti_{18}Nb_{36}N_{12}O_{114-x}$, further indicating the beneficial effects of rational O-vacancies in promoting diffusion kinetics. We, therefore, confirm that the synergistic effect of N-incorporation and O-vacancies can induce localized structural polarization, enhance Li⁺ adsorption and facilitate electron/ion mobility.

## Ultrafast spectroscopy analysis of electron dynamics and charge migration

Transient absorption (TA) spectroscopy is a powerful tool for monitoring the dynamics of excited-state carriers on ultrafast timescales[35]. An illustration of the experimental setup is shown in Fig. 4a (details in the **Experimental Section**). Here, the carrier dynamics of pure-phase TNO and $TNO_{-x}$@N samples on sub-picosecond to hundreds of picosecond time delay are investigated. According to the steady-state absorption spectrum of TNO and $TNO_{-x}$@N (Supplementary Fig. 19), the pump energy of TA is set to 3.55 eV (350 nm) for exciting electrons from the valence band to the conduction band. Figure 4b, c illustrates pseudo color TA plots of the two samples, and Fig. 4d, e presents TA spectra at different time delays. A negative signal within initial 1 picosecond (ps) represents a photoinduced bleach signal caused by band-filling effect of free carrier[36]. A significant redshift and broadening of $TNO_{-x}$@N bleach indicate that the band-filling-state carriers contain less energy after relaxation to the band edges, evidencing a narrowing of the bandgap[37], consistent with the calculations in Fig. 3d.

A broad and positive photoinduced absorption signal that appears after a few ps in the TA spectra is mainly contributed by the optical response of polarons[38], a quasi-particle of excited carriers in local lattice distortions caused by the carriers' charge[39], which have been commonly observed in transition metal oxides[40]. Figure 4f shows the extracted dynamics at the 625 nm probe, where the bi-exponential function is employed to fit the dynamic curve. Other relaxation processes with much longer time constants are out of the time window of our experiments. The fitting results of the two curves are presented in Supplementary Table 3. Notably, the carrier decay lifetime $\tau_2$ is extended by ~2 times (from 14.85 to 30.22 ps) after N-incorporation. Due to the oxygen-vacancies introduced by N dopant, the slow process $\tau_2$ is attributed to oxygen-defect-assisted recombination, and the increased lifetime can effectively promote charge separation and transfer[41]. Another time constant $\tau_1$ for a faster process of bi-exponential function fitting results is 1.06 ps and 1.52 ps for TNO and $TNO_{-x}$@N, respectively. We excluded the possibility of the fast process contributed by hot-carrier relaxation and nonlinear processes by performing pump wavelength and excitation fluence-dependent experiments (Supplementary Fig. 21). Therefore, we attribute this ~1 ps process to the formation of polaron, of which the time constant is consistent with the time scale reported previously[42]. Figure 4g represents a schematic diagram of the relaxation process of excited state carriers.

By comparing the normalized dynamics after a few picoseconds, $TNO_{-x}$@N is found to form an increased concentration of polarons (Fig. 4f), which is likely attributed to structural distortion in the lattice of $TNO_{-x}$@N caused by the introduction of oxygen-vacancies. It is expected that the Li-ion migration barrier could be lowered by the increment of polarons concentration due to the strong screening experienced by Li-ion during migration through polarons[43,44]. On the other hand, rational anion engineering can effectively reduce the stabilization energy gained upon lattice distortion and localization (the energy difference between the delocalized and polaronic states, $\Delta E_{pol}$). The decreased $\Delta E_{pol}$ indicates that the polarons are less stable and more susceptible to being re-excited to a delocalized state, which is transported in a more efficient band-like manner[45] (Fig. 4g). Thus, a

lower $\Delta E_{pol}$ has a suppressive effect on polaron formation, which is consistent with the increased trend in the time constant $\tau_1$ of the polaron formation process inferentially.

In fact, electric current flows under voltage holding bias via polaron conduction process known as polaron hopping, which requires thermal vibrational energy to hop from one site to the next (Fig. 4h). Polaron hopping has been explicitly proposed as one of the main factors determining electronic conductivity and electrochemical performance of solid-state electrode materials in many prior works[46]. The structural distortion caused by the incorporation of N dopants is ascribed to the accelerated polaron hopping by lowering of the polaron hopping activation energy $E_a$[47] (Fig. 4i), resulting in almost 5 orders of magnitude higher measured electrical conductivity of $TNO_{-x}$@N. Considering all the above, we believe that electric delocalization plays an important effect on the formation and transport of polarons, and is one of the main reasons that may contribute to the enhanced electric conductivity and reduced Li ion transport barrier.

## Temperature-dependent electrochemical performance

To explore the lithium diffusion kinetics of the $TNO_{-x}$@N electrode, a series of electrochemical measurements are conducted (Fig. 5). Obviously, the CV curves of $TNO_{-x}$@N show a higher redox current and a lower polarization potential than those of pure-phase TNO with flower structure and bulk phase, suggesting a dramatically improved reactivity and efficiency (Supplementary Fig. 22). Figure 5a demonstrates excellent cycling reliability of the $TNO_{-x}$@N electrode at 0.5 C, with a high reversible capability of 254.7 mAh g⁻¹ and capacity retention of 93.0% over 130 cycles, outperforming the pure TNO flower (244.1 mAh g⁻¹, 87.0%) and solid TNO (232.7 mAh g⁻¹, 91.5%). When the current rate is increased to 1 C, the $TNO_{-x}$@N electrode still achieves a high capacity of 257.4 mAh g⁻¹ with 0.0104% capacity decay per cycle for up to 300 cycles (Supplementary Fig. 23). Such high cycling stability in $TNO_{-x}$@N can be ascribed to enlarged interlayer distance for Li⁺ diffusion due to the synergistic effect of N-incorporation and O-vacancies. Meanwhile, the mixture of graphite and $TNO_{-x}$@N (denoted as G/TNO) according to a mass ratio of 1:1 demonstrates excellent cycling reliability at 0.5 C, with a high reversible capability of 273.4 mAh g⁻¹ and capacity retention of 100% over 50 cycles (Supplementary Fig. 24).

Further evidence for the fast-charging capability of $TNO_{-x}$@N electrode in the current density range from 0.2 to 60 C is shown in Fig. 5b and Supplementary Fig. 25. A stable reversible capacity of 186.7 mAh g⁻¹ can still be achieved even at a high current density of 60 C (60 s), which surpasses that of TNO, solid TNO, and the majority of the recently reported Nb-based electrodes[29,30,48–60] (Fig. 5c). Even when the rate is switched to 0.2 C, the output reversible capacity of $TNO_{-x}$@N can be restored to the initial level, revealing high kinetic reversibility. At the high current density, the increase in capacity can be due to the concentration polarization, resulting in the inability to deintercalation active lithium during the initial process. Meanwhile, we found that the majority of the Nb-based oxides possess a capacity increase trend at high current densities[53,61–63], which may also be related to the intrinsic properties and pseudo-capacitive behavior of Nb-based oxides. Additionally, the impacts of $TNO_{-x}$ sample with just containing O defects feature on the battery performances are conducted. As shown in Supplementary Fig. 26, the $TNO_{-x}$ electrode exhibits the specific capacity of 251.6 mAh g⁻¹ after 100 cycles at 0.5 C. As the rate increased to 60 C, it delivers a capacity of 179.1 mAh g⁻¹, approaching $TNO_{-x}$@N (186.7 mAh g⁻¹), further revealing the O-vacancies feature is the main contribution to the improvement of the rate performances of the TNO material. The prolonged cycling stability of the $TNO_{-x}$@N is further evaluated at 10 C in Supplementary Fig. 27. A high reversible capability of 184.4 mAh g⁻¹ of $TNO_{-x}$@N after

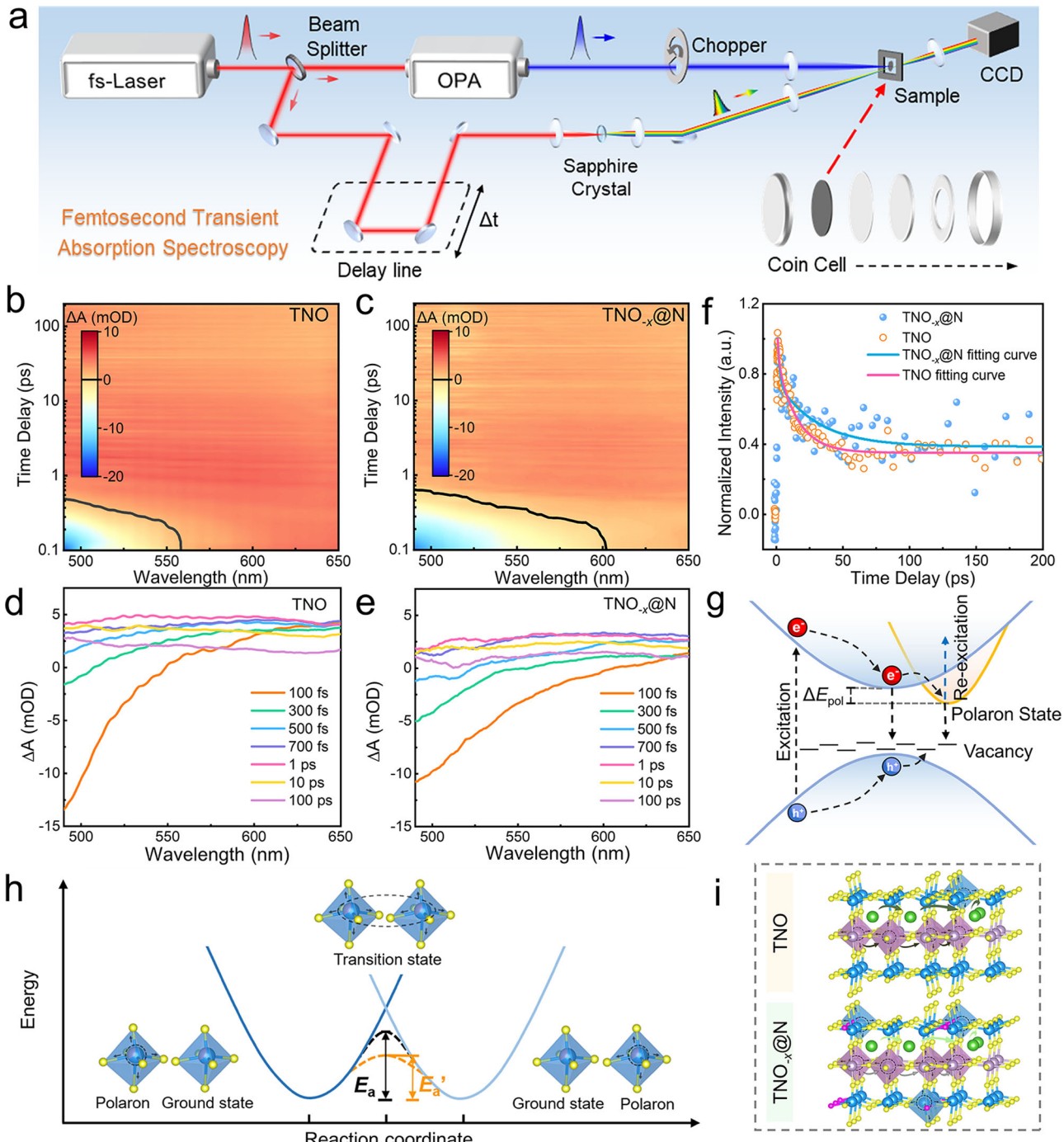

**Fig. 4 | Transient absorption (TA) study of charge carrier dynamics and migration. a** Schematic illustration of principles for femtosecond transient absorption spectroscopy. **b, c** The pseudo color TA spectra plots of TNO (**b**) and TNO$_{-x}$@N (**c**) measured after excitation at 3.55 eV, ~1.7 mJ cm⁻². The black lines represent the position of positive and negative boundaries of the TA signal. **d, e** TA spectra at different time delays between pump and probe pulse of TNO (**d**) and TNO$_{-x}$@N (**e**). **f** Comparison of extracted dynamics at 625 nm probe and bi-exponential function fit curve. **g** The energy-band structure diagram and

recombination processes of excited-state carriers in TNO$_{-x}$@N. **h** Schematic diagram of polaron hopping between two identical metal ions with potential energy landscapes. The dark and orange dashed curves indicate adiabatic surfaces of TNO and TNO$_{-x}$@N. $E_a$ and $E_a'$ represent the polaron hopping activation energy of TNO and TNO$_{-x}$@N, respectively. **i** Schematic illustration of Li⁺ migration accompanies with polaron hopping process (the yellow, purple, pink, green and blue balls represent O, Ti, N, Li and Nb atoms, respectively).

1000 cycles can be achieved, much better than that of the TNO and solid TNO counterpart.

Given the urgent demand for highenergy-density LIBs, cycling performances of the different anode electrodes with the high mass loading are evaluated (Fig. 5d and Supplementary Fig. 28). From the cross-sectional SEM image in Fig. 5e, the high-loading electrode

(10 mg cm⁻²) indicates a much higher thickness than that of the regular electrode. The thick TNO$_{-x}$@N electrode exhibits a high capacity retention of 82.3% with an stable capacity of 185.9 mAh g⁻¹ at 1 C (1 C= after 250 cycles, which is much larger than that of the TNO (166.5 mAh g⁻¹, 75.3%, the 2ⁿᵈ as the initial calculated capacity) and solid TNO electrode (114.7 mAh g⁻¹, 56.8%). Even if the mass loading is

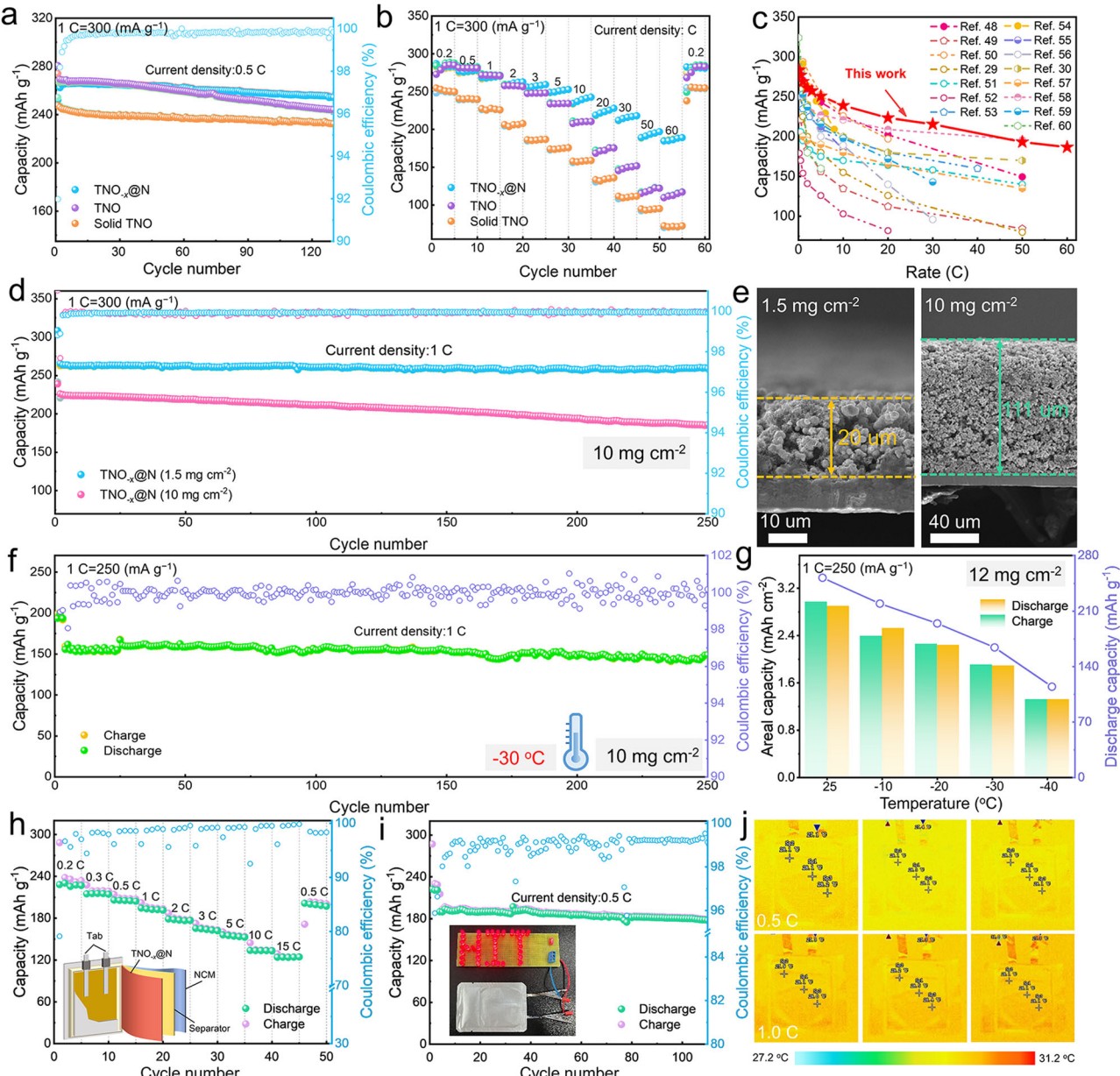

**Fig. 5 | Lithium storage performance of TNO$_{-x}$@N, TNO, and solid TNO with high mass loading at low temperature. a** Cyclability and **b** rate capability of the three electrodes with the loading mass of about 1.5 mg cm$^{-2}$. (1 C = 300 mA g$^{-1}$) **c** Comparison of rate performance of TNO$_{-x}$@N-based batteries with previously reported Nb-based electrodes. **d** Long-term cycling stability of the TNO$_{-x}$@N electrode with the mass loading of 1.5 mg cm$^{-2}$ and 10 mg cm$^{-2}$ after an activation process of 1 cycle at 0.2 C. **e** Cross-sectional images of the electrodes with mass loading of 1.5 mg cm$^{-2}$ and 10 mg cm$^{-2}$. **f** Cyclability of the TNO$_{-x}$@N electrode with

the mass loading of 10 mg cm$^{-2}$ operated at 1 C and −30 °C after an activation process of 3 cycles at 0.2 C. (1 C = 250 mA g$^{-1}$). **g** Temperature-dependent areal capacity and discharge capacity of the TNO$_{-x}$@N electrode operated from 25 °C to −40 °C. **h** Rate capability and **i** cyclability of TNO$_{-x}$@N||LiNi$_{0.8}$Co$_{0.1}$Mn$_{0.1}$O$_2$ pouch cell at 25 °C after an activation process of 3 cycles at 0.2 C. (1 C = 250 mA g$^{-1}$) **j** Infrared temperature images of TNO$_{-x}$@N||LiNi$_{0.8}$Co$_{0.1}$Mn$_{0.1}$O$_2$ pouch cell at 0.5 C and 1.0 C.

increased to 20 mg cm$^{-2}$, the areal capacity is up to 4.4 mAh cm$^{-2}$ (Supplementary Fig. 29), which is comparable to or even higher than the areal capacities of reported graphite and Si-based electrode[64–70]. After 80 cycles at 0.2 C, it still remains 3.35 mAh cm$^{-2}$.

Low-temperature (Low-T) performance is a critical criterion for cold-region energy storage Low-T capacity and maintenance of the TNO$_{-x}$@N electrode at −20 °C and −30 °C are tested in Supplementary Fig. 30. At −20 °C, a stable capacity of 229.0 mAh g$^{-1}$ and an high capacity retention of 99.8% are achieved after 70 cycles at 0.5 C, which is preferable to that of TNO flowers (215.9 mAh g$^{-1}$, 98.8%) and solid TNO (141.3 mAh g$^{-1}$, 98.4%). As the temperature drops to −30 °C, the reversible capacity of TNO$_{-x}$@N remains 215.4 mAh g$^{-1}$ after 70

cycles, which surpasses that of TNO (203.5 mAh g$^{-1}$) and solid TNO (116.0 mAh g$^{-1}$). When the loading is increased to 6 mg cm$^{-2}$, a remarkably high charge capacity of 212.5 mAh g$^{-1}$ at 0.2 C is delivered for TNO$_{-x}$@N at −30 °C, corresponding to 75.4% (regular electrodes) of the room-temperature capacity. However, the TNO and solid TNO electrode retain only 68.8% and 72.7% (195.6 mAh g$^{-1}$ and 182.6 mAh g$^{-1}$) of the room-temperature capacity, manifesting the effective promotion of the electron delocalization effect on the Low-T performance (Supplementary Fig. 31). With the rate increased to 5 C and 6 C, it provides high reversible capacities of 154.4 mAh g$^{-1}$ and 147.6 mAh g$^{-1}$ at the high loading of 6 mg cm$^{-2}$, respectively (Supplementary Fig. 32), surpassing previously reported LTO- and

Nb-based electrodes (Supplementary Table 4). Moreover, the thick TNO$_{-x}$@N electrode exhibits a stable capacity of 149.3 mAh g$^{-1}$ and a high capacity retention of 92.9% (the 26$^{th}$ as the initial calculated capacity) after 250 cycles at 1 C. (Fig. 5f). As the temperature further drops to −40 °C, the areal capacity of the TNO$_{-x}$@N electrode can still retain 1.32 mAh cm$^{-2}$ with a high loading of 12 mg cm$^{-2}$, demonstrating the considerable potential for Low-T operation (Fig. 5g). In addition, the cycling performance of TNO$_{-x}$@N electrode with the high mass loading of 20 mg cm$^{-2}$ at −30 °C (Supplementary Fig. 33) and −40 °C (Supplementary Fig. 34). The TNO$_{-x}$@N electrode exhibits the initial areal capacity of 3.14 mAh cm$^{-2}$ and 2.36 mAh cm$^{-2}$ at the current density of 0.1 C. Furthermore, after 50 cycles at 0.2 C, the areal capacity of 2.57 mAh cm$^{-2}$ and 1.86 mAh cm$^{-2}$ are obtained.

To demonstrate its potential commercial application, the TNO$_{-x}$@N-based pouch cell is assembled with commercial LiNi$_{0.8}$Co$_{0.1}$Mn$_{0.1}$O$_2$ as the cathode according to the schematic diagram shown in Fig. 5h inset. In the voltage window (1.0 − 3.0 V), it yields a large discharge capacity of 228.2 mAh g$^{-1}$ at 0.2 C (Fig. 5h). As the current density increases from 0.3 C to 15 C, the capacity preserves 215.1, 205.2, 192.8, 177.4, 163.7, 154.3, 133.4, and 124.0 mAh g$^{-1}$, respectively. Especially, a high reversible capacity of 199.2 mAh g$^{-1}$ is still recovered when the current density is shifted back to 0.5 C. Furthermore, the pronounced cyclability of TNO$_{-x}$@N ||LiNi$_{0.8}$Co$_{0.1}$Mn$_{0.1}$O$_2$ can still be stabilized at 177.6 mAh g$^{-1}$ at 0.5 C over 110 cycles (Fig. 5i). Meanwhile, the 3.5 Ah-level pouch cells with commercial LiNi$_{0.8}$Co$_{0.1}$Mn$_{0.1}$O$_2$ as the cathode is further tested, as shown in Supplementary Fig. 35. The TNO$_{-x}$@N || LiNi$_{0.8}$Co$_{0.1}$Mn$_{0.1}$O$_2$ pouch cell shows remarkable cycling reliability at 0.5 C, with a capacity of 3.32 Ah after 55 cycles and a capacity retention of 97.6% (the 2$^{nd}$ as the initial calculated capacity). The calculated mass energy density and volumetric energy density of TNO$_{-x}$@N electrode are obtained as 271.6 Wh kg$^{-1}$ and 373.4 Wh L$^{-1}$ (details in the **Experimental Section**), respectively, which are comparable to Nb-based materials, Ti-based materials, carbon materials (graphite), and silicon-based materials (Supplementary Table 5). As the temperature further drops to −30 °C, the 3.5 Ah-level TNO$_{-x}$@N ||LiNi$_{0.8}$Co$_{0.1}$Mn$_{0.1}$O$_2$ pouch cell still retains a capacity of 2.05 Ah after 80 cycles (Supplementary Fig. 36). The calculated mass energy density and volumetric energy density of TNO$_{-x}$@N electrode are obtained as 168.4 Wh kg$^{-1}$ and 231.5 Wh L$^{-1}$, respectively, further demonstrating the practical potential of TNO$_{-x}$@N. In addition, the spatial heat distribution of the pouch cells is almost unchanged at various discharging and charging rates and states, indicating a high thermal safety of TNO$_{-x}$@N electrode under extreme conditions (Fig. 5j).

### Tracing the origin of rapid low-temperature dynamics for TNO$_{-x}$@N

To understand the multidimensional ion/electron transmission paths of TNO$_{-x}$@N electrode, synchrotron X-ray tomography microscopy is employed to reveal the 3D spatial distribution. Figure 6a maps the top view of a random 3D slice of the TNO$_{-x}$@N electrode, where the active material (AM) particles have been rendered with different colors. Meanwhile, the morphological characteristics and volume area of the different particles within the microstructure information are investigated using the threshold segmentation method. As highlighted in Fig. 6b, the quantified volume fractions of AM and inactive material particles (carbon/binder domain (CBD) and pore spaces) are 27.4% and 72.6%, respectively. Besides, the pore network illustrates that the carbon/binder completely encapsulates the active particles and possesses a high degree of connectivity, which further ensures efficient transport pathways of ions and electrons at low temperatures.

Interfacial kinetics remains a major challenge in improving Low-T performance. Firstly, electrochemical impedance spectroscopy (EIS) curves of three study electrodes are evaluated, in which the charge-transfer impedance ($R_{ct}$) of the TNO$_{-x}$@N is lower than that of the TNO and solid TNO both at room and low temperatures (Fig. 6c and Supplementary Table 6), which is highly related with the electron transfer

at the solid-liquid interface. In addition, the slope angles of the low-frequency region are larger than 45°, implying the existence of capacitive-like behavior during the Li-storage process.

To further elucidate the kinetic behaviors of TNO$_{-x}$@N, rate-dependent CV measurements are performed at various scan rates from 0.1 to 1.0 mV s$^{-1}$ (Fig. 6d). With increasing the sweep rate, primary oxidation peaks shift to high potential, while the reduction peaks switch to low potential. In principle, the $b$-value can be identified by fitting the log(scan rate ($v$))−log(peak current ($i$)) plot. In Supplementary Fig. 37a, the $b$-values of the anodic peak and cathodic peak are 0.84 and 0.84, respectively, suggesting that the redox reaction dynamic is mainly governed by the fast intercalation pseudo-capacitive process. The total capacitive contribution can be quantified by the equation of $i = k_1 v + k_2 v^{0.5}$ [54]. From the green shaded area in Supplementary Fig. 37b, the integral area ascribed to the capacitive behavior is 84.7% at 0.6 mV s$^{-1}$, higher than the pure TNO (81.6%, Supplementary Fig. 38) and solid TNO (69.2%, Supplementary Fig. 39). When the mass loading is increased to 8 mg cm$^{-2}$, the capacitive contribution remains 78% of the total capacity at 0.6 mV s$^{-1}$, and capacitive contribution gradually improves with the arising scan rate, reaching a maximum value of 82% at 1.0 mV s$^{-1}$ (Fig. 6e and Supplementary Fig. 40). Such high capacitive contribution are ascribed to the fast Li-storage at the high working current induced by the synergistic effect between N-incorporation and O-vacancies. Moreover, the nanosheet-assembled compact structure can provide available extra sites for interfacial storage, which explains why TNO$_{-x}$@N owns superior rate capability even at low temperatures.

To better assess the intrinsic ion-transport capability of TNO$_{-x}$@N, the galvanostatic intermittent titration techniques (GITT) are performed by measuring the macroscopic diffusion coefficients ($D_{Li+}$, details in the **Experimental Section**). During lithiation, the average $D_{Li+}$ value of TNO$_{-x}$@N is $4.0 \times 10^{-13}$ cm$^2$ s$^{-1}$ at 25 °C, and the value during de-lithiation is $6.2 \times 10^{-13}$ cm$^2$ s$^{-1}$ (Supplementary Fig. 41), which is higher than the TNO ($2.9 \times 10^{-13}$ cm$^2$ s$^{-1}$/$4.5 \times 10^{-13}$ cm$^2$ s$^{-1}$) and solid TNO ($1.4 \times 10^{-13}$ cm$^2$ s$^{-1}$/$1.6 \times 10^{-13}$ cm$^2$ s$^{-1}$). As the temperature drops to −40 °C (Fig. 6f), the $D_{Li+}$ values decrease by only one order of magnitude, keeping at $1.6 \times 10^{-14}$ cm$^2$ s$^{-1}$ (lithiation) and $2.6 \times 10^{-14}$ cm$^2$ s$^{-1}$ (delithiation, Fig. 6g), which are significantly larger than those of TNO ($5.5 \times 10^{-15}$ cm$^2$ s$^{-1}$/$8.2 \times 10^{-15}$ cm$^2$ s$^{-1}$), solid TNO ($4.4 \times 10^{-15}$ cm$^2$ s$^{-1}$/$6.4 \times 10^{-15}$ cm$^2$ s$^{-1}$). Excitingly, the apparent Li$^+$ diffusion coefficient of TNO$_{-x}$@N at −40 °C is still comparable to the recently reported some Nb−based materials (Supplementary Table 7) at room temperature. These results demonstrate that anion engineering can expand the lattice spacing to provide wide diffusion paths and lower diffusion energy barriers, thereby achieving fast Li$^+$ transport at room/low temperatures. In addition, the interfacial desolvation/migration reactions of the different electrode materials are also measured by the temperature-dependence EIS technique (Supplementary Fig. 42 and Supplementary Table 8). Based on the Arrhenius relationship, the activation energy ($E_a$) of TNO$_{-x}$@N (24.62 kJ mol$^{-1}$) is inferior to that of TNO (40.57 kJ mol$^{-1}$) and solid TNO (40.87 kJ mol$^{-1}$), suggesting an increased proportion of activated molecules and thus fast Li$^+$ intercalation kinetics (Fig. 6h).

Another crucial factor of Low-T performance is the wettability influences. Figure 6i illustrates the schematic diagram of how TNO$_{-x}$@N realizes superwettability. Specifically, the chemical composition and geometric structure (or surface roughness) of the electrode material have been proven valid to reduce the contact angle (i.e., superwettability). As plotted in Fig. 6j, the contact angles of solid TNO, TNO, and TNO$_{-x}$@N with the electrolyte at room temperature are 24.3°, 17.8°, and 10.6°, respectively, further revealing that the TNO$_{-x}$@N electrode is beneficial for the rapid migration of Li$^+$ between the electrode and the electrolyte, consequently, stable low-temperature maintenance.

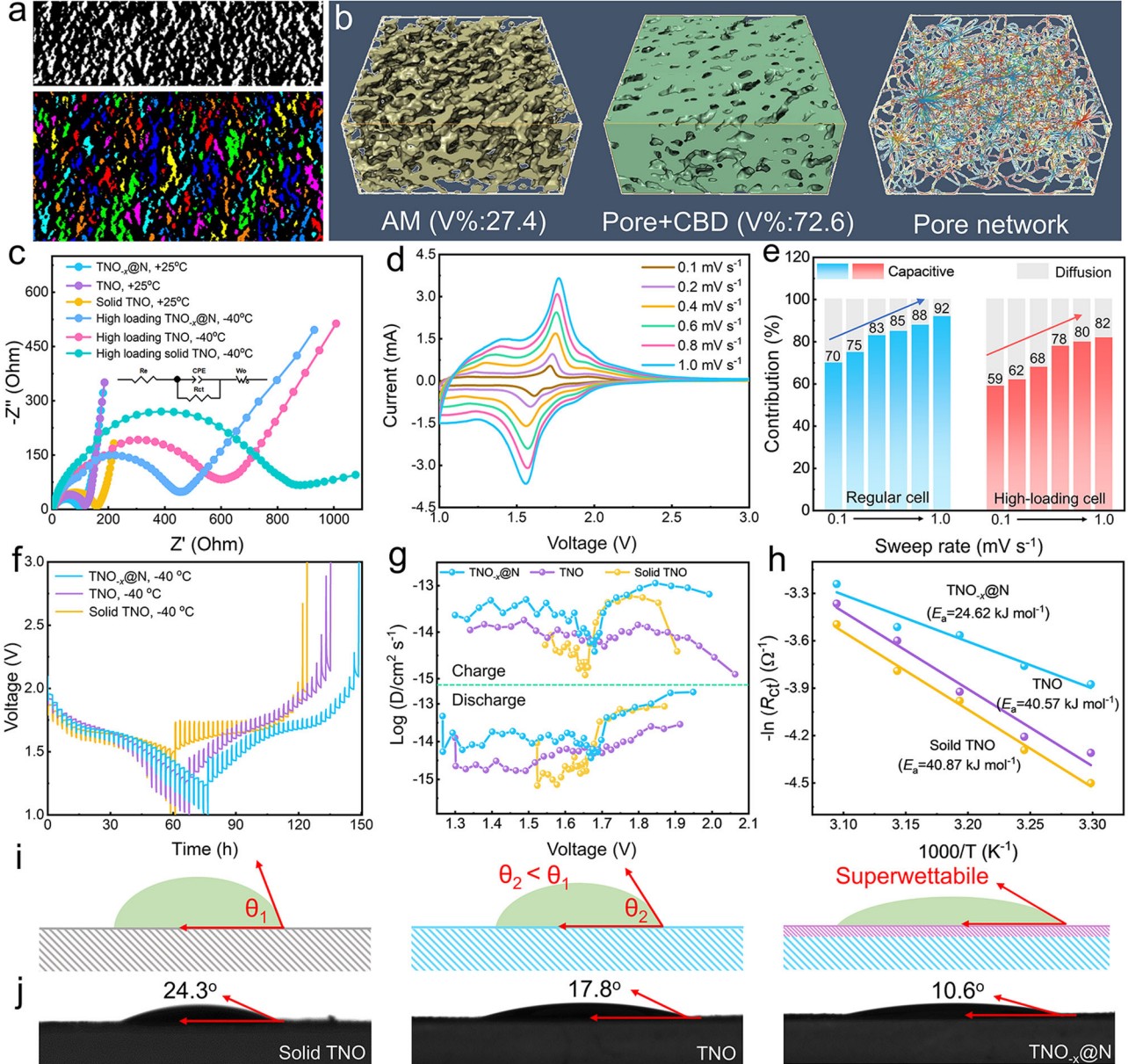

**Fig. 6 | Low-temperature electrochemical behaviors and kinetic analyzes.**
**a** Cross-section view slice of an original TNO$_{-x}$@N electrode. **b** Synchrotron X-ray tomography reconstruction with volume rendering shows the 3D microstructure of the active particles, pores, and pore connectivity network diagram (color indicates local pore center). **c** The EIS of the TNO$_{-x}$@N, TNO and solid TNO at 25 and −40 °C. **d** CV curve of half cells at various sweep rates of TNO$_{-x}$@N at 25 °C. **e** Capacitive contribution to the total capacity of TNO$_{-x}$@N with the different mass loading. **f** GITT curves of the TNO$_{-x}$@N, TNO, and solid TNO at −40 °C. **g** The calculated Li$^+$ diffusion coefficients. **h** Arrhenius curves of TNO$_{-x}$@N, TNO, and solid TNO. **i** Schematic illustration of how TNO$_{-x}$@N realizes superwettability. **j** The contact angle of the TNO$_{-x}$@N, TNO, and solid TNO with the electrolyte at room temperature.

## In situ crystal-structure evolution and Li$^+$ storage mechanism

In situ XRD measurements of TNO$_{-x}$@N and TNO are performed to deeply comprehend the structural evolution under different lithiation/delithiation states. As discharging proceeded, the characteristic peaks at 23.9° and 32.4°, corresponding to the (011) and (21$\bar{5}$) planes of TiNb$_2$O$_7$ (JCDPS No. 72-0116), shift toward a smaller angle with increasing d-spacing for Li$_x$TiNb$_2$O$_7$. Meanwhile, the (300) diffraction peak that appearing at 25.8° displays a zig-zag migration and the peak intensities significantly decrease, indicating the reduced crystal-structural order (Fig. 7a and Supplementary Fig. 43). Subsequently, all characteristic diffraction peaks of TNO$_{-x}$@N return to their original Bragg positions and intensities upon recharging due to a high reversible phase transition. Thus, it can be concluded that the gradient nitriding does not damage the crystalline structure of TiNb$_2$O$_7$ as

observed with the similar behavior of the pure TNO electrode in Fig. 7b. Nevertheless, the lattice anisotropy of TNO$_{-x}$@N is weakened with a smaller variation that can be observed from the selected peak plots in Fig. 7c, d.

To further explicate the evolution of the lattice-parameter variations, the in situ XRD patterns are refined by the Rietveld method. According to the a-value parameters variations, the crystal structure evolution can be divided into three regions during the lithiation process that can be described as follows. M1 region (solid-solution transition) indicates the Li$^+$ insert into the bulk structure of TNO$_{-x}$@N to form Li$_{0.88}$TNO$_{-x}$@N; M2 region (two-phase coexistence) represents the transformation from Li$_{0.88}$TNO$_{-x}$@N to Li$_{2.67}$TNO$_{-x}$@N; M3 region (solid-solution transition) demonstrate the evolution from Li$_{2.67}$TNO$_{-x}$@N to Li$_4$TNO$_{-x}$@N. Note that the unit-cell volume variation

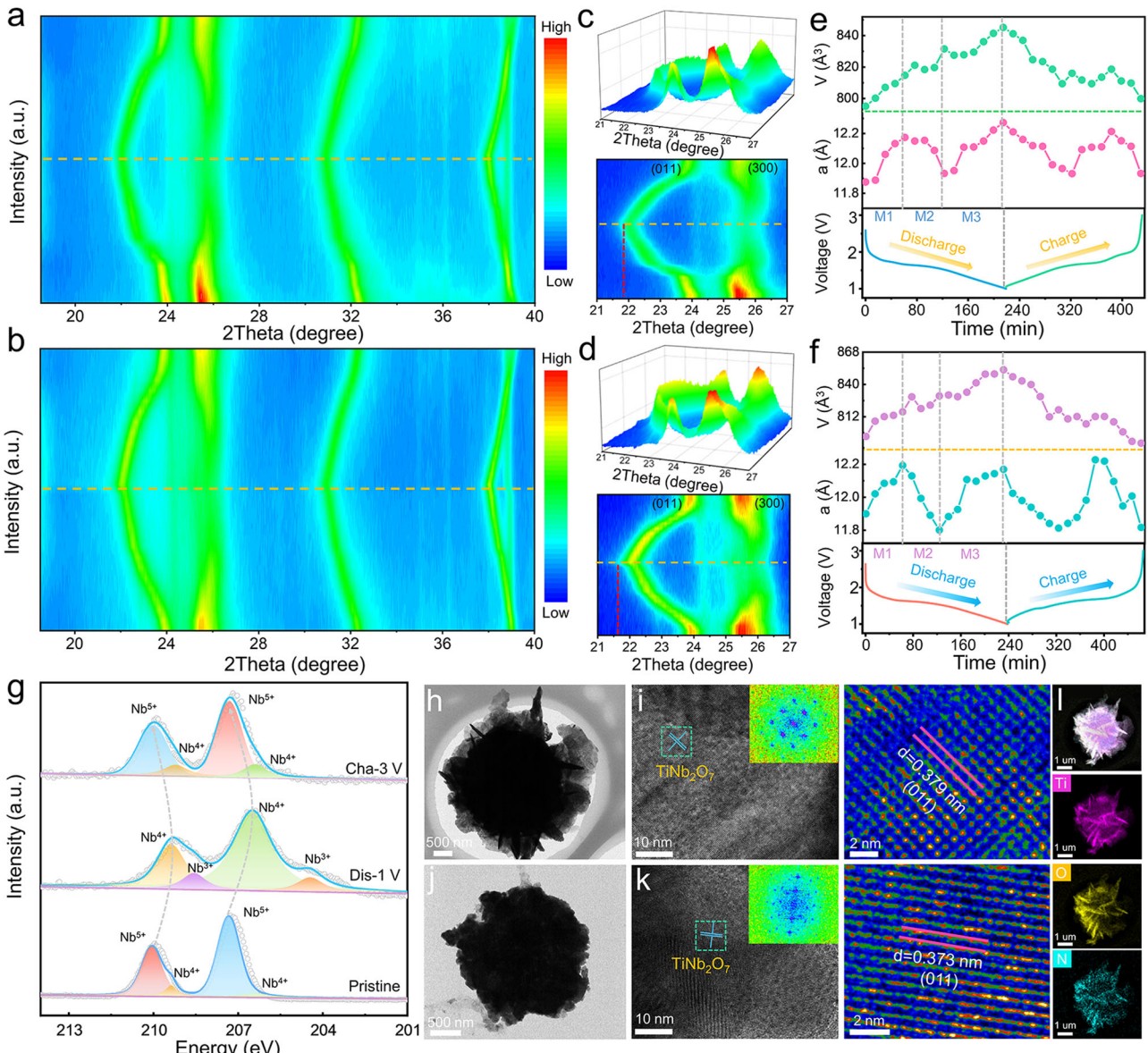

**Fig. 7 | Phase transformation and reaction mechanism. a, b** In situ 2D contour of XRD plots of TNO$_{-x}$@N (**a**) and TNO (**b**). **c, d** Selected 3D surface maps and corresponding contour maps TNO$_{-x}$@N (**c**) and TNO (**d**). **e, f** Lattice-constant variations and corresponding discharge/charge curves of TNO$_{-x}$@N (**e**) and TNO (**f**). **g** Nb 3d XPS spectra recorded at different potentials. **h–k** Ex situ TEM and HRTEM images of the TNO$_{-x}$@N at the discharge to 1.0 V (**h, i**) and the charge to 3.0 V (**j, k**). **l** Elemental mapping images of the TNO$_{-x}$@N at the discharged state of 1.0 V.

of TNO$_{-x}$@N at full lithiation (Fig. 7e) is slightly smaller than that of TNO (Fig. 7f), which is attributed to the fact that anion engineering can effectively increase the interlayer spacings, thereby alleviating the repetitive strains induced by the volumetric expansion and contraction during lithiation and delithiation.

Ex situ XPS spectra of the TNO$_{-x}$@N electrode with various depths of discharge/charge are executed to investigate the evolution of elemental valences (Fig. 7g). After discharged to 1.0 V, the peaks at 209.4 and 206.5 eV are consistent with Nb 3d$_{3/2}$ and Nb 3d$_{5/2}$ of Nb$^{4+}$, whereas the peaks at 208.5 and 204.4 eV belong to Nb$^{3+}$, suggesting that the two-electron transfer per Nb occurs during lithiation. After a full charge, the spectrum can be restored to a great extent, but the content of Nb$^{4+}$ increases, which is ascribed to the partially inserted Li$^+$ cannot be extracted upon the initial charge process (the initial Coulombic efficiency is 98.5%). Further evidence is attested by inspection of the evolution of the (011) lattice spacing of the TNO$_{-x}$@N electrode at cutoff voltages (1.0 V, Fig. 7i; 3.0 V, Fig. 7k), proving the strong

Coulombic repulsion of the residual Li$^+$ in turn stretches the (110) plane. The ex situ TEM images with elemental mapping images of the TNO$_{-x}$@N at a fully discharged (Figs. 7h, l) and charged state (Fig. 7j and Supplementary Fig. 44) further manifest the excellent reversibility of the structural transition during cycling.

## Discussion

In summary, a charge-transfer-enhanced TiNb$_2$O$_{7-x}$@N electrode with a unique delocalized electronic structure has been constructed for high-areal-capacity batteries at low temperatures. Time-resolved optical spectroscopy reveals that the heterogeneous adjustment of the electronic structure of TiNb$_2$O$_7$ induces an impurity energy band at the Fermi energy level, lowering polaron hopping activation energy and increasing the free carrier and polaron concentration for improved electronic conductivity. The results are in strong agreement with the theoretical calculations. Such regulation provides an inclination for Li$^+$ adsorption and decreases the energy barriers for ion

diffusion, which in turn accelerates electrochemical reaction kinetics. As a result, the TNO$_{-x}$@N electrode exhibits large reversible capacity, good rate capability, and high durability. It can even deliver an areal capacity of 1.32 mAh cm$^{-2}$ at −40 °C. Furthermore, a dramatically decreased lattice volume expansion of TNO$_{-x}$@N electrode is achieved during the lithiated/delithiated process. This work sheds light on the design concept of localized structure polarization, inspiring a pathway of structural engineering strategy toward practical applications of low-temperature batteries.

## Methods

### Material synthesis

3 mmol niobium pentachloride and 18 mmol oxalic acid were dissolved in 50 mL distilled water and labeled as Solution A. Meanwhile, 1.5 mmol tetrabutyl titanate and 6 mmol oxalic acid were further dispersed in 20 mL anhydrous ethanol and labeled as Solution B. After which, Solution A was dropped into Solution B to form Solution C. Subsequently, 12 mmol ammonium fluoride was added into Solution C by stirring for 100 min to a transparent solution. Then, Solution C was transferred into a 100 mL Teflon-lined autoclave and heated at 180 °C for 12 h. Finally, the product was washed repeatedly with alcohol before drying it at 80 °C for 10 h to achieve TNO precursor. The TNO precursor was annealed at 750 °C for 4 h with a heating rate of 5 °C min$^{-1}$ in a tube furnace to obtain TNO composite. After that, the obtained TNO powder was heated under a mixed Ar/NH$_3$ atmosphere to 700 °C for 1 h to obtain nitrided TNO microflowers (abbreviated as TNO$_{-x}$@N). The solid TNO material was obtained by homogeneous mixing of Nb$_2$O$_5$ and TiO$_2$ with a certain ratio in an ethanol solution, followed by heating in a muffle furnace at 1225 °C for 20 h.

### Material characterizations

Crystal structures of samples were investigated by X-ray diffraction (XRD, Bruker D8 Advance) and Raman spectra (Jobin-Yvon Lab RAM HR-800). Surface chemistry was analyzed with X-ray photoelectron spectra using the PHI 5700-ESCA system. Scanning electron microscopy (SEM, Helios Nanolab 600i) and transmission electron microscopy (TEM, FEI Talos 200 S) were employed to determine the structure and morphologies of the samples. The high-angle annular-dark-field and annular-bright-field images were performed with a spherical-aberration-corrected scanning transmission electron microscope (Titan Cubed Themis G2 300). Contact angles were acquired on a Cam-plus Micro meter using the sessile-drop technique.

### Electrochemical measurement

The mass ratio of active material, Super P, and polyvinylidene fluoride (PVDF) was 8:1:1 to fabricate the regular working electrode slurry, which was coated on Cu foils and dried in a vacuum at 100 °C for 10 h. The average loading of active materials in the regular anode was about 1.5 mg cm$^{-2}$, and the 1C-rate was defined as 300 mA g$^{-1}$ at the voltage range of 1.0-3.0 V. The electrolyte is homemade, and a purity of 99.99%. The electrolyte was a mixture of ethyl carbonate, diethyl carbonate and dimethyl carbonate with a volume ratio of 1:1:1 containing 1 mol L$^{-1}$ LiPF$_6$. The diameter of the regular electrode was 14 mm, the amount of electrolyte was 70 uL, and the diameter of the separator was 16.5 mm. The separator of all half-cells and pouch cells is the Celgard 2500 polypropylene membrane. The thickness, dimensions and capacity of Li foils were 15.6 mm, 0.45 mm and 20 Ah cm$^{-2}$, respectively. For the high-loading electrode, the active material, Super P, and PVDF were homogeneously mixed with a mass ratio of 9:0.5:0.5. The diameter of the high-loading electrode was 12 or 10 mm, the amount of electrolyte was 100 uL, the separator was polypropylene (PP), and the diameter of the separator was 16.5 mm. For the high-loading electrode, the 1C-rate was defined as 250 mA g$^{-1}$ at the voltage range of 1.0−3.0 V. Lithium metal and LiNi$_{0.8}$Co$_{0.1}$Mn$_{0.1}$O$_2$ were used as electrode materials to assemble half-cells and pouch cell, respectively. For the

cathode electrode, the slurries consisting of LiNi$_{0.8}$Co$_{0.1}$Mn$_{0.1}$O$_2$ as the active materials, Kejten black & carbon nanotube as the conducting agent, and polyvinylidene fluoride (PVDF) binder in a weight ratio of 95.2:3:1.8 were prepared. The cathode electrode had a mass loading of approximately 6.4 mg cm$^{-2}$ and a compaction density of 3.45 g cm$^{-3}$. For the anode electrode, the pastes consisting of TNO$_{-x}$@N as the active materials, Kejten black & carbon nanotube as the conducting agent, and PVDF binder in a weight ratio of 92:5:3 were fabricated. The anode electrode had a mass loading of approximately 4.8 mg cm$^{-2}$ and a compaction density of 2.30 g cm$^{-3}$. The dimensions of the electrode are 4.3 × 4.2 cm. The N/P ratio of the pouch cell was set to 1.0, and the amount of electrolyte was 1.0 mL. For the 3.5 Ah-level pouch cells, the average mass loading of the single-sided anode and cathode electrodes was about 15.4 and 16.1 mg cm$^{-2}$, respectively. The dimensions of the electrode are 4.6 × 8.0 cm, and the cathode and anode electrodes were assembled by stacking the pieces. The amount of electrolyte was 14 mL. The 1C-rate is defined as 3.5 A at the voltage range of 1.0−3.0 V. The mass energy density of TNO$_{-x}$@N-based pouch cell can be quantified by the equation of mass energy density = cell capacity × average voltage/electrode weight. The volumetric energy density of TNO$_{-x}$@N-based pouch cell can be calculated by the equation of volumetric energy density = cell capacity × average voltage/electrode (thickness × width × length). Electrochemical impedance spectroscopy (EIS) and cyclic voltammetry (CV) were performed on a CHI760 electrochemical workstation. The galvanostatic charge/discharge tests were conducted on the Neware-CT3008 system.

According to Fick's second law of diffusion, the $D_{Li+}$ of the TNO$_{-x}$@N, TNO and solid TNO electrodes can be computed by:

$$D_{Li+} = \frac{4}{\tau\pi} \left(\frac{V_m m_B}{S M_B}\right)^2 \left(\frac{\Delta E_S}{\Delta E_\tau}\right)^2 \qquad (1)$$

where $\tau$ is the pulse duration time, $V_m$, $S$, $m_B$ and $M_B$ are the mole volume, electrode area, electrode mass, and molar mass, respectively. $\Delta E_S$ and $\Delta E_\tau$ stand for the change in the steady-state potential and the change in potential after the pulse, respectively.

### Finite element modeling (FEM) calculation

The implement of electrochemical simulation was carried out on the surface of the solid sphere and microflowers electrode using the cubic current distribution physical field interface in the COMSOL Multiphsics software. The model is based on the calculation of currents in the electrolyte and electrodes using the "cubic current distribution" interface. Therefore, the electrolyte current is resolved according to Ohm's law, where one of the electrodes is grounded, and the other is set to the battery potential to meet the total current condition.

Here, the electric field follows the continuity equation using the current density:

$$i = -F \sum -z_i^2 m_i F c_i \nabla \phi_i \qquad (2)$$

where $i$ is the current density vector; $z_i$ is the ion charge; $m_i$ is the mobility, F is the Faraday constant, $\phi_i$ is the ion potential, and $c_i$ represents the concentration of the ion. Meanwhile, it is consistent with the conservation of current density.

$$i = 0 \qquad (3)$$

Expressions of the Butler-Volmer form are utilized in the simulations to describe the electrode kinetics occurring at the electrode surface inserted in the electrolyte, and the exchange current density for the oxidation reaction is considered to be concentration-dependent. The electrode surface current density is referred to as

the Butler-Volmer equation:

$$i_a = i_0 \left[ \frac{c}{c_0} \exp\left(\frac{\eta(1-\beta)F}{RT}\right) - \exp\left(\frac{\eta\beta F}{RT}\right) \right] \quad (4)$$

The initial value of the electrolyte potential is set to be comparable to the potential of the cell at an open circuit (the open circuit voltage). The overpotential is defined as follows:

$$\eta = \phi_s - \phi_l - E_{eq} \quad (5)$$

where $\eta$ is the overpotential, $\phi_s$ is the electrode potential, $\phi_l$ is the electrolyte potential and $E_{eq}$ is the equilibrium potential.

The transport of dissolved Li$^+$ in the electrolyte and electrode during charging and discharging is modeled by transient simulations of the "rare-mass transfer" interface, which assumes that the transport of ions can be described by diffusion according to Fick's law. Meanwhile, mass transfer induced by the diffusion and migration is considered.

$$-D\nabla C - zmFc\nabla\phi_l = N \quad (6)$$

in which $c$ represents the concentration of the ions, $z$, $D$ and $m$ are the valence, diffusivity, and mobility, respectively. F and $\phi_l$ stand for the Faraday's constant and ionic potential.

The ionic concentration in the electrode material meets the depletion reaction process, which is modeled using the PDE module of the software:

$$\frac{\partial c}{\partial t} = -ck_s \quad (7)$$

where $k_s$ is the rate of consumption reaction.

For the boundary setting, we set the bottom reference potential as zero potential, and the top as the average current density boundary 2.5 mA/cm$^2$. In addition, the conductivity of the electrolyte is set as 4.5 S/m. The diffusion coefficient of the ions as $5 \times 10^{-10}$ m$^2$/s, and the environmental concentration of the Li ions boundary as 1 mol/L.

**Transient Absorption Measurements**
The femtosecond transient absorption (fs-TA) setup is based on a regenerative amplified Ti: sapphire laser system from a Coherent and Helios pump-probe system (Ultrafast Systems). The regenerative amplified Ti: sapphire laser system (Legend Elite-1K-HE, 800 nm, 25 fs, 4 mJ, and 1 kHz) was seeded with a mode-locked Ti: sapphire laser system (Vitara) and pumped with a Nd: YLF laser (Evolution 30). Briefly, the 800 nm output pulse from the regenerative amplifier was split into two parts with a 50% beam splitter. The transmitted part was used to pump a TOPAS Optical Parametric Amplifier (OPA) which generates a wavelength-tunable laser pulse of 350 nm as a pump beam and is chopped by a mechanical chopper operating at a frequency of 500 Hz. Another reflected part of the fundamental beam was introduced into the TA spectrometer to generate the probe light. After passing through a motorized optical delay line, the fundamental beam was focused on a sapphire crystal, which was used to generate the white-light continuum (WLC) probe pulses with wavelengths of 430–820 nm. The optical path difference between the pump light and the probe light, which is controlled by the motorized optical delay line, was used to monitor the transient states at different pump-probe delays. A reference beam was split from the WLC to correct the pulse-to-pulse fluctuation of the WLC. The pump was spatially and temporally overlapped with the probe beam on the sample.

For TA measurements, two samples of TNO and TNO$_{-x}$@N were prepared by following steps. First, TNO and TNO$_{-x}$@N were added to ethanol in the weight ratio of 10:1 (ethanol: TNO/TNO$_{-x}$@N), through ultrasonic dispersion in a water bath for 2 hour, to obtain the TNO and TNO$_{-x}$@N mixture suspension. Then, the TNO and TNO$_{-x}$@N films were deposited on cleaned SiO$_2$ substrates by spin coating at 1200 rpm for 30 s. After the films were naturally dried, they were sealed with thin SiO$_2$ substrates and epoxy resin to isolate the air. All these steps were conducted in a nitrogen environment at room temperature.

**In situ XRD and ex situ measurement**
In the in situ XRD measurement, in situ cells were assembled in glovebox with Li metal as counter electrode, TNO$_{-x}$@N or TNO as working electrode, glass fiber as the separator, and 1 mol L$^{-1}$ LiPF$_6$ in a mixture of the ethyl carbonate, diethyl carbonate and dimethyl carbonate (1:1:1 volume ratio) as the electrolyte. Each XRD curve is collected with a time frame of around 15 min, and the total time for the first cycle of charge and discharge at the current density of 0.25 C is approximately 7.2 h.

For the ex situ measurements, the coin cells were transferred to a glove box after a certain number of cycles, and dismantled to remove the test electrodes. The removed electrodes were soaked in dimethyl carbonate solution for 12 h and dried and sealed in a glass bottle for subsequent testing. This process always avoids contact with air.

**Computational methods**
The Vienna Ab Initio Simulation Package (VASP) for all the spin-polarized DFT calculations within the generalized gradient approximation (GGA) using the PBE functional formulation was implemented. Projected augmented wave (PAW) pseudopotentials were employed to describe the electron-ion interaction. The valence electronic states were expanded in plane wave basis sets with the energy cutoff of 450 eV. The electronic energy is considered self-consistent when the energy change is smaller than $10^{-6}$ eV, while a force change smaller than 0.03 eV/Å is used to determine the convergence of the geometry optimization. Oxygen vacancy formation and lithium-ion migration were calculated using the TiNb$_2$O$_7$ supercell bulk model of $1 \times 3 \times 1$, while the adsorption/desorption of EC-Li was calculated using the TiNb$_2$O$_7$ (001) slab model with a $p(1 \times 3)$ supercell. The climbing-image nudged elastic band (CI-NEB) method was applied to obtain the diffusion barriers.

## Data availability
The data that support the plots within this paper and other findings of this study are available from the corresponding author upon request. Source data are provided in this paper.

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

## Acknowledgements

This work was supported by the National Key Research and Development Program of China (2022YFE0138900, S. L.), the National Natural Science Foundation of China (Grant No. 22279026, S. L.), the Shanghai Aerospace Science and Technology Innovation Fund (No. SAST2022-106, S. L.), the Young Elite Scientist Sponsorship Program by CAST (No. YESS20200148, S. L.), and the Fundamental Research Funds for the Central Universities (Grant No. HIT.OCEF.2022017, S. L.). We thank Dr. Guohao Du and Dr. Biao Deng of the beamline BL13W1 and BL18B at Shanghai Synchrotron Radiation Facility (SSRF) for SR-CT measurements.

## Author contributions

Y.Z. and Y.W. contributed equally. Y.Z., G.Y., T.Z., and S.L. conceived the project and designed the experiments. Y.Z., Y.W., and W.Z. contributed to part of the preparation and characterization of the material and the electrode. Y.Z., Y.W., W.Z., P.Z., Y.T., G.Y., T.Z., and S.L. wrote the manuscript with support from all authors.

## Competing interests

The authors declare no competing interests.
