## [Peer Review File · Nature Communications]

REVIEWER COMMENTS

Reviewer #1 (Remarks to the Author):

In this work, the authors have proposed a $\text{TiNb}_2\text{O}_7\text{-x@N}$ anode that has a unique delocalized electronic structure, which enables its high performance in low temperatures of up to -30°C . The authors state their use of time-resolved optical spectroscopy, which is used for the first time and reveals the electronic structure of the material at the Fermi level. While the authors have used vigorous research methods and experimental data to back up their claims, I believe that the current state of the manuscript is significantly lacking in quality and data for the reasons that are listed below. Therefore, I believe that the manuscript is not fit for this journal, and at least a major revision is needed before its acceptance as a publication in this journal.

1. If I am not mistaken, the authors do not specifically mention in the beginning sections of the manuscript whether the material is an anode or a cathode. Although readers of the article are highly likely to easily catch whether the material is an anode or a cathode, the authors should specifically mention the type of the material for researchers outside the battery community reading this article.

2. The authors state that “an obvious blue-shift of about 0.2 eV of the Nb spectra and Ti spectra” can be seen. However, I believe that the shift energy values are too small for the authors to confidently state that the energy shift exists. Are the authors confident enough to say that these values are enough to back up their claims? Data with higher validity should be further given. Furthermore, the raw data of the deconvoluted Nb and Ti data must be present along with the plotted data, in order for the readers to compare the validity of the deconvoluted XPS spectra. In Supplementary Figures S3b, the colored graph is on top of other graphs, making it hard for the readers to observe how the peaks of the graphs are situated. The authors should fix this problem, as their claim on “presence of oxygen deficiencies after the ammonia annealing treatment” cannot be clearly seen. Moreover, the energy of the XPS spectra is normally shown in the order of high to low energy from left to right in the x-axis. The authors should follow this for the figures in the manuscript and the supplementary figures.

3. The authors have used a “Finite element simulation” to capture the real-time Li^+ concentration between the solid sphere and microflowers upon the lithiation process of the synthesized materials. If I am not mistaken, the authors do not fully explain in the Experimental Section on what and how this simulation is carried out. Furthermore, the authors have used this simulation under harsh lithiation (15 C discharge) conditions. What is the reason for this extensively high C-rate? This value must be explained.

4. In Figure 4d, the authors have used electrodes with high mass loading to compare the cycling performances of the different anodes. Cycling performance of the electrodes must be compared also under low mass loadings, as it is known that at low mass loadings the cycling performance of the electrodes decrease.

5. Overall, the figures are very “messy” and very hard to see. The figures are generally all too small to see, and this must be fixed for it to be accepted in this journal. For example, Figure 1h shows elemental mappings of Nb, Ti, O in the synthesized materials, but which picture showing which element is excruciatingly hard to see. This is a crucial factor in deciding the acceptance of this manuscript in the journal, as the readability of a manuscript is a very important factor that must be considered before publishing a journal.

Reviewer #2 (Remarks to the Author):

The manuscript entitled “Delocalized electronic engineering to reduce kinetic barrier enables ultralow temperature capability for high-area-capacity batteries” describes excellent electrochemical performances of a TNO-based anode by surface modification and doping. Although they successfully synthesized well-structured TNO-x@N microflowers and performed in-depth studies of calculations, the concept of this work is not very new including a synthetic method, ammonia gas nitriding, defect engineering, etc. More importantly, the TNO anode material is not practical in the viewpoint of energy density. Therefore, I recommend a rejection of this manuscript. The detailed comments on the manuscript are as follows.

1. There are many papers with the similar concept. Please cite them.

1.1. Material synthesis

[1] Carbon-coated TiNb₂O₇ nanosheet arrays as self-supported high mass-loading anode for flexible Li-ion battery, *Chem. Commun.*, 2021, 57, 18221825

1.2. Gas nitriding

[1] Porosity-Controlled TiNb₂O₇ Microspheres with Partial Nitridation as A Practical Negative Electrode for High-Power Lithium-Ion Batteries. *Adv. Energy Mater.* 2015, 5, 1401945

[2] Surface nitrided and carbon coated TiNb₂O₇ anode material with excellent performance for lithium-ion batteries, *Journal of Alloys and Compounds*, Volume 835, 2020, 155241

[3] Porous TiNb₂O₇ nanofibers decorated with conductive Ti_{1-x}Nb_xN bumps as a high power anode material for Li-ion batteries, *J. Mater. Chem. A* (2015)

1.3. Oxygen defect engineering

[1] Porous oxygen-deficient TiNb₂O₇ spheres wrapped by MXene as high-rate and durable anodes for liquid and all-solid-state lithium-ion batteries, *Chemical Engineering Journal*, 438 (2022), 135328

[2] Defect engineering of TiNb₂O₇ compound for enhanced Li-ion battery anode performances, *Electrochimica Acta*, Volume 404, 2022, 139603

2. The electrochemical performances of the TNO have been reported in many articles, and the reaction mechanism has also been studied by the DFT calculation. Although the nitrogen doping and the oxygen vacancies can improve the electronic property and Li⁺ ion migration, the TNO itself shows a high rate capability and a stable cyclability. To improve the quality of this manuscript, the full cell with a high capacity cathode should be evaluated.

3. In my opinion, the merit of this work is the lithium storage performance at the low temperature. This is very important property of the commercial Li ion batteries. However, the anode with only TNO material does not show a large impact because of the lower capacity. In this regard, it would be a better way to show a low temperature property of a mixture of the graphite and TNO.

Reviewer #3 (Remarks to the Author):

Dear editor,

The article "Delocalized electronic engineering to reduce kinetic barrier enables ultralow temperature capability for high-area-capacity batteries" by Yan Zhang et al. expose delocalized electronic engineering in TiNb₂O₇ @N anodes. The characterization of this material has been carried on using novel techniques (femtosecond laser-based transient absorption spectroscopy, synchrotron X-ray tomography microscopy, et al), good carrier transport performance was confirmed. However, there are several core issues should be solved, I think the major revision is required before we can make a decision.

1. Introduction, paragraph 4, line 1-3

How do active electronic states boost ion-diffusion kinetics? I think there is no direct correlation between electronic conduction and ion conduction since the electronic conductivity is the directional transition of electrons, while Li-ions transport is the ion mass transfer process at the electrolyte/electrode interface.

2. Introduction, paragraph 4

It is recommended not to mention high mass loading or high areal capacity as your advantages in this TNO@N anode. Scheme 1 only shows that an increased specific surface area and a shortened diffusion path can enhance the dynamics of lithium ion transfer. The material you synthesized is a “flower” spherical material with micro size, and I do not believe that this structure has a high packing density because the “flowers” composed of nanosheets cannot be compacted, and there are also large pores between adjacent particles. The manuscript did not directly provide a comparison value between mass loading/areal capacity with commercial materials. Your high mass loading is mainly contributed by high-density metals Ti and Nb, but its thickness is actually not significant. The areal capacity of commercial anode electrode materials ranges from 3.0 to 3.5 mA h cm⁻², while your material is only 1.3 mA h cm⁻².

3. Results and Discussion, Morphology, structure and kinetics of the TNO@N., paragraph 2, line 3-12

(1) Does N 1s exist in the form of free radicals or through bonding? Please provide specific instructions. Your explanation is very vague.

(2) Is the signal of O 1s at 531.9 eV an oxygen hole? Or hydroxyl oxygen? The original text states that this location contains these two types of signal samples, it is not right.

(3) The change in charge distribution can be determined based on the binding energy according to XPS. However, there is no correlation between the charge distribution and the increase in conductivity. The increase in conductivity should be caused by the change in TNO band structure caused by the introduction of N components into the material. We suggest adjusting the conductivity data to the relevant position, such as the band data in Fig. 2.

4. Results and Discussion, Morphology, structure and kinetics of the TNO-x@N., paragraph 4

Providing the specific surface area and pore size distribution data of TNO, otherwise, the only mentions interlayer spacing of 0.372 nm cannot support your view since it remains unchanged before and after doping N.

5. Results and Discussion, Theoretical insight into the electron delocalization effect., paragraph 1

We do not observe any changes in electron delocalization state in Fig. 2c? No significant differences were observed except for the color difference in a small area at the center. Please provide a specific explanation.

6. Results and Discussion, Theoretical insight into the electron delocalization effect., paragraph 1

The explanation of PDOS and band gap results is not clear. You need to directly state that after N doping, the band gap narrows, TNO transforms from an insulator to a semiconductor, hence the carrier diffusion resistance decreases. Though the characterization is done beautifully, the key conclusion is unclear.

7. Results and Discussion, Temperature-dependent electrochemical performance., paragraph 2

Why is there an upward trend in specific capacity under the same rate performance, as shown in Fig. 4b. For instance, at 20 C, the discharge capacity increases during the 35th~40th cycled, while this phenomenon does not occur at a low rate.

8. Results and Discussion, Tracing the origin of rapid low-temperature dynamics for TNO @N. paragraph 3

Complete the DLi+ of TNO in the manuscript, and use less qualitative description for things that can be quantitatively explained. What is the degree when using "larger than" in the manuscript?

9. Results and Discussion, Temperature-dependent electrochemical performance., paragraph 5

The full battery assembled with NCM cathode has not provided key data on mass energy density and volume energy density, nor has it been compared with existing work. Therefore, it is impossible to judge the performance advantages of the battery from the perspective of commercial application from commercial application aspect.

10. There are six spelling errors in the manuscript, for example, "in genral", "infunces", please correct them.

Reviewers' comments:

Reviewer #1 (Remarks to the Author):

In this work, the authors have proposed a $\text{TiNb}_2\text{O}_7\text{-x@N}$ anode that has a unique delocalized electronic structure, which enables its high performance in low temperatures of up to -30°C . The authors state their use of time-resolved optical spectroscopy, which is used for the first time and reveals the electronic structure of the material at the Fermi level. While the authors have used vigorous research methods and experimental data to back up their claims, I believe that the current state of the manuscript is significantly lacking in quality and data for the reasons that are listed below. Therefore, I believe that the manuscript is not fit for this journal, and at least a major revision is needed before its acceptance as a publication in this journal.

1. If I am not mistaken, the authors do not specifically mention in the beginning sections of the manuscript whether the material is an anode or a cathode. Although readers of the article are highly likely to easily catch whether the material is an anode or a cathode, the authors should specifically mention the type of the material for researchers outside the battery community reading this article.
2. The authors state that "an obvious blue-shift of about 0.2 eV of the Nb spectra and Ti spectra" can be seen. However, I believe that the shift energy values are too small for the authors to confidently state that the energy shift exists. Are the authors confident enough to say that these values are enough to back up their claims? Data with higher validity should be further given. Furthermore, the raw data of the deconvoluted Nb and Ti data must be present along with the plotted data, in order for the readers to compare the validity of the deconvoluted XPS spectra. In Supplementary Figures S3b, the colored graph is on top of other graphs, making it hard for the readers to observe how the peaks of the graphs are situated. The authors should fix this problem, as their claim on "presence of oxygen deficiencies after the ammonia annealing treatment" cannot be clearly seen. Moreover, the energy of the XPS spectra is normally shown in the order of high to low energy from left to right in the x-axis. The authors should follow this for the figures in the manuscript and the supplementary figures.
3. The authors have used a "Finite element simulation" to capture the real-time Li^+ concentration between the solid sphere and microflowers upon the lithiation process of the synthesized materials. If I am not mistaken, the authors do not fully explain in the Experimental Section on what and how this simulation is carried out. Furthermore, the authors have used this simulation under harsh lithiation (15 C discharge) conditions. What is the reason for this extensively

high C-rate? This value must be explained.

4. In Figure 4d, the authors have used electrodes with high mass loading to compare the cycling performances of the different anodes. Cycling performance of the electrodes must be compared also under low mass loadings, as it is known that at low mass loadings the cycling performance of the electrodes decrease.

5. Overall, the figures are very “messy” and very hard to see. The figures are generally all too small to see, and this must be fixed for it to be accepted in this journal. For example, Figure 1h shows elemental mappings of Nb, Ti, O in the synthesized materials, but which picture showing which element is excruciatingly hard to see. This is a crucial factor in deciding the acceptance of this manuscript in the journal, as the readability of a manuscript is a very important factor that must be considered before publishing a journal.

Reviewer #2 (Remarks to the Author):

The manuscript entitled “Delocalized electronic engineering to reduce kinetic barrier enables ultralow temperature capability for high-areal-capacity batteries” describes excellent electrochemical performances of a TNO-based anode by surface modification and doping. Although they successfully synthesized well-structured TNO-x@N microflowers and performed in-depth studies of calculations, the concept of this work is not very new including a synthetic method, ammonia gas nitriding, defect engineering, etc. More importantly, the TNO anode material is not practical in the viewpoint of energy density. Therefore, I recommend a rejection of this manuscript. The detailed comments on the manuscript are as follows.

1. There are many papers with the similar concept. Please cite them.

1.1. Material synthesis

[1] Carbon-coated TiNb₂O₇ nanosheet arrays as self-supported high mass-loading anode for flexible Li-ion battery, *Chem. Commun.*, 2021, 57, 18221825

1.2. Gas nitriding

[1] Porosity-Controlled TiNb₂O₇ Microspheres with Partial Nitridation as A Practical Negative Electrode for High-Power Lithium-Ion Batteries. *Adv. Energy Mater.* 2015, 5, 1401945

[2] Surface nitrided and carbon coated TiNb₂O₇ anode material with excellent performance for lithium-ion batteries, *Journal of Alloys and Compounds*, Volume 835, 2020, 155241

[3] Porous TiNb₂O₇ nanofibers decorated with conductive Ti_{1-x}Nb_xN bumps as a high power anode material for Li-ion batteries, *J. Mater. Chem. A* (2015)

1.3. Oxygen defect engineering

[1] Porous oxygen-deficient TiNb₂O₇ spheres wrapped by MXene as high-rate and durable anodes for liquid and all-solid-state lithium-ion batteries, *Chemical Engineering Journal*, 438 (2022), 135328

[2] Defect engineering of TiNb₂O₇ compound for enhanced Li-ion battery anode performances, *Electrochimica Acta*, Volume 404, 2022, 139603

2. The electrochemical performances of the TNO have been reported in many articles, and the reaction mechanism has also been studied by the DFT calculation. Although the nitrogen doping and the oxygen vacancies can improve the electronic property and Li⁺ ion migration, the TNO itself shows a high rate capability and a stable cyclability. To improve the quality of this manuscript, the full cell with a high capacity cathode should be evaluated.

3. In my opinion, the merit of this work is the lithium storage performance at the low temperature. This is very important property of the commercial Li ion batteries. However, the anode with only TNO material does not show a large impact because of the lower capacity. In this regard, it would be a better way to show a low temperature property of a mixture of the graphite and TNO.

Reviewer #3 (Remarks to the Author):

Dear editor,

The article "Delocalized electronic engineering to reduce kinetic barrier enables ultralow temperature capability for high-area-capacity batteries" by Yan Zhang et al. expose delocalized electronic engineering in TiNb₂O₇@N anodes. The characterization of this material has been carried on using novel techniques (femtosecond laser-based transient absorption spectroscopy, synchrotron X-ray tomography microscopy, et al), good carrier transport performance was confirmed. However, there are several core issues should be solved, I think the major revision is required before we can make a decision.

1. Introduction, paragraph 4, line 1-3

How do active electronic states boost ion-diffusion kinetics? I think there is no direct correlation between electronic conduction and ion conduction since the electronic conductivity is the directional transition of electrons, while Li-ions transport is the ion mass transfer process at the electrolyte/electrode interface.

2. Introduction, paragraph 4

It is recommended not to mention high mass loading or high areal capacity as

your advantages in this TNO@N anode. Scheme 1 only shows that an increased specific surface area and a shortened diffusion path can enhance the dynamics of lithium ion transfer. The material you synthesized is a "flower" spherical material with micro size, and I do not believe that this structure has a high packing density because the "flowers" composed of nanosheets cannot be compacted, and there are also large pores between adjacent particles. The manuscript did not directly provide a comparison value between mass loading/areal capacity with commercial materials. Your high mass loading is mainly contributed by high-density metals Ti and Nb, but its thickness is actually not significant. The areal capacity of commercial anode electrode materials ranges from 3.0 to 3.5 mA h cm⁻², while your material is only 1.3 mA h cm⁻².

3. Results and Discussion, Morphology, structure and kinetics of the TNO@N., paragraph 2, line 3-12

(1) Does N 1s exist in the form of free radicals or through bonding? Please provide specific instructions. Your explanation is very vague.

(2) Is the signal of O 1s at 531.9 eV an oxygen hole? Or hydroxyl oxygen? The original text states that this location contains these two types of signal samples, it is not right.

(3) The change in charge distribution can be determined based on the binding energy according to XPS. However, there is no correlation between the charge distribution and the increase in conductivity. The increase in conductivity should be caused by the change in TNO band structure caused by the introduction of N components into the material. We suggest adjusting the conductivity data to the relevant position, such as the band data in Fig. 2.

4. Results and Discussion, Morphology, structure and kinetics of the TNO-x@N., paragraph 4

Providing the specific surface area and pore size distribution data of TNO, otherwise, the only mentions interlayer spacing of 0.372 nm cannot support your view since it remains unchanged before and after doping N.

5. Results and Discussion, Theoretical insight into the electron delocalization effect., paragraph 1

We do not observe any changes in electron delocalization state in Fig. 2c? No significant differences were observed except for the color difference in a small area at the center. Please provide a specific explanation.

6. Results and Discussion, Theoretical insight into the electron delocalization effect., paragraph 1

The explanation of PDOS and band gap results is not clear. You need to directly state that after N doping, the band gap narrows, TNO transforms from an insulator to a semiconductor, hence the carrier diffusion resistance decreases.

Though the characterization is done beautifully, the key conclusion is unclear.

7. Results and Discussion, Temperature-dependent electrochemical performance., paragraph 2

Why is there an upward trend in specific capacity under the same rate performance, as shown in Fig. 4b. For instance, at 20 C, the discharge capacity increases during the 35th~40th cycled, while this phenomenon does not occur at a low rate.

8. Results and Discussion, Tracing the origin of rapid low-temperature dynamics for TNO @N. paragraph 3

Complete the DLi+ of TNO in the manuscript, and use less qualitative description for things that can be quantitatively explained. What is the degree when using "larger than" in the manuscript?

9. Results and Discussion, Temperature-dependent electrochemical performance., paragraph 5

The full battery assembled with NCM cathode has not provided key data on mass energy density and volume energy density, nor has it been compared with existing work. Therefore, it is impossible to judge the performance advantages of the battery from the perspective of commercial application from commercial application aspect.

10. There are six spelling errors in the manuscript, for example, "in genral", "infunces", please correct them.

Response to Reviewer's Comments

(Manuscript ID: NCOMMS-23-54010-T)

Response to Reviewer #1

Overall comment: In this work, the authors have proposed a $\text{TiNb}_2\text{O}_{7-x}\text{@N}$ anode that has a unique delocalized electronic structure, which enables its high performance in low temperatures of up to -30°C . The authors state their use of time-resolved optical spectroscopy, which is used for the first time and reveals the electronic structure of the material at the Fermi level. While the authors have used vigorous research methods and experimental data to back up their claims, I believe that the current state of the manuscript is significantly lacking in quality and data for the reasons that are listed below. Therefore, I believe that the manuscript is not fit for this journal, and at least a major revision is needed before its acceptance as a publication in this journal.

Response: Thank you very much for your positive and constructive comments concerning our manuscript. Those comments are valuable and helpful for revising and improving our paper. We highly agree with the comments, and the corresponding revisions in the paper and the point-by-point responses to the reviewer's comments are listed as follows:

Comment 1: If I am not mistaken, the authors do not specifically mention in the beginning sections of the manuscript whether the material is an anode or a cathode. Although readers of the article are highly likely to easily catch whether the material is an anode or a cathode, the authors should specifically mention the type of the material

for researchers outside the battery community reading this article.

Response 1: Thank you for this valuable comment. We apologize for not being able to specifically state whether the material is an anode or a cathode in the beginning sections in the original manuscript. According to the reviewer's suggestion, we emphasized that the niobium-based oxide (TiNb_2O_7) is a potential anode material with the potential of 1.64 V vs. lithium anodes, and have received extensive attention owing to the safe lithiation potential and suitable channels for Li^+ migration related to the defective ReO_3 -like structure.[1-3]

References:

R1. Cui, P., Li, G. T., Zhang, P. P., Wan T., Li, M. Q., Chen, X. L., Zhou, Y., Guo, R. Q., Su, M. R., Liu, Y. J., Chu, D. W. Arranging cation mixing and charge compensation of TiNb_2O_7 with W^{6+} doping for high lithium storage performance. *Rare Met.* **42**, 3364–3377 (2023).

R2. Han, J.-T.; Goodenough, J. B. 3-V full cell performance of anode framework TiNb_2O_7 /Spinel $\text{LiNi}_{0.5}\text{Mn}_{1.5}\text{O}_4$. *Chem. Mater.* **23**, 3404-3407 (2011).

R3. Liang, H., Liu, L., Wang, N., Zhang, W., Hung, C.-T., Zhang, X., Zhang, Z., Duan, L., Chao, D., Wang, F., Xia, Y., Li, W., Zhao, D. Unusual mesoporous titanium niobium oxides realizing sodium-ion batteries operated at -40°C . *Adv. Mater.* **34**, 2202873 (2022).

The detail has been added to the revised manuscript, which is highlighted in yellow as shown below,

Revision in the manuscript

Page 3:

Niobium-based oxides (TiNb_2O_7), as a potential anode material with the potential of 1.64 V vs. lithium anodes, have received extensive attention owing to the safe lithiation potential and suitable channels for Li^+ migration related to the defective ReO_3 -like structure.[11-13]

References:

Ref 11. Cui, P., Li, G. T., Zhang, P. P., Wan T., Li, M. Q., Chen, X. L., Zhou, Y., Guo, R. Q., Su, M. R., Liu, Y. J., Chu, D. W. Arranging cation mixing and charge compensation of TiNb_2O_7 with W^{6+} doping for high lithium storage performance. *Rare Met.* **42**, 3364–3377 (2023).

Ref 12. Han, J.-T.; Goodenough, J. B. 3-V full cell performance of anode framework TiNb_2O_7 /Spinel $\text{LiNi}_{0.5}\text{Mn}_{1.5}\text{O}_4$. *Chem. Mater.* **23**, 3404-3407 (2011).

Ref 13. Liang, H., Liu, L., Wang, N., Zhang, W., Hung, C.-T., Zhang, X., Zhang, Z., Duan, L., Chao, D., Wang, F., Xia, Y., Li, W., Zhao, D. Unusual mesoporous titanium niobium oxides realizing sodium-ion batteries operated at $-40\text{ }^\circ\text{C}$. *Adv. Mater.* **34**, 2202873 (2022).

Comment 2: The authors state that “an obvious blue-shift of about 0.2 eV of the Nb spectra and Ti spectra” can be seen. However, I believe that the shift energy values are too small for the authors to confidently state that the energy shift exists. Are the authors confident enough to say that these values are enough to back up their claims? Data with higher validity should be further given. Furthermore, the raw data of the deconvoluted Nb and Ti data must be present along with the plotted data, in order for the readers to compare the validity of the deconvoluted XPS spectra. In Supplementary Figures S3b, the colored graph is on top of other graphs, making it hard for the readers to observe how the peaks of the graphs are situated. The authors should fix this problem, as their claim on “presence of oxygen deficiencies after the ammonia annealing treatment” cannot be clearly seen. Moreover, the energy of the XPS spectra is normally shown in the order of high to low energy from left to right in the x-axis. The authors should follow this for the figures in the manuscript and the supplementary figures.

Response 2: We thank the reviewer very much for this valuable comment. We agree with the reviewer that the shift energy values of the Nb spectra and Ti spectra are too small to confidently state that the energy shift exists. In the original manuscript, the

sample was exposed to air for a long time before being measured and therefore caused a partially polluted sample surface, rendering an inaccurate signal peak due to that the XPS analysis depth is merely a few nanometers (≤ 10 nm).

To objectively prove the existence of energy transfer, we supplemented the etching-assisted XPS depth measurement (etch the surface of TNO and TNO- x @N samples for 60 s) and X-ray absorption near-edge structure (XANES). For the high-resolution Nb 3d spectrum of TNO (the bottom half of **Figure R1a**), two peaks at binding energies of 209.8 and 207.1 eV are ascribed to Nb⁵⁺.^[4,5] For the high-resolution Nb 3d spectrum of TNO- x @N (the top half of **Figure R1a**), two additional small peaks attributed to the Nb⁴⁺ can be also observed at 209.1 and 206.0 eV.^[6] In **Figure R1b**, two additional peaks at 463.4 and 457.3 eV are detected in the Ti 2p spectra of TNO- x @N in comparison to TNO, corresponding to the characteristic positions of Ti³⁺ cations.^[7] Compared to the TNO material, the Nb spectra and Ti spectra in the TNO- x @N show an obvious blue shift of about 0.2 eV, which can be ascribed to the presence of charge redistribution.^[8,9] Besides, the x -axis of the XPS spectra is revised to show from high energy (left) to low energy (right).

Figure R1. High-resolution XPS spectra of (a) Nb 3d and (b) Ti 2p.

To further confirm the existence of energy shift, X-ray absorption near-edge structure of TNO and TNO_x@N are investigated. The XANES profile of TNO_x@N (Figure R2) lies between Nb foil and TNO, indicating an average oxidation state of Nb that is intermediate between the two referenced samples. This analysis is consistent with the XPS results, further suggesting that an enhanced electron redistribution occurs after the ammonia annealing treatment.

Figure R2. Nb K-edge XANES of Nb foil, TNO, and TNO_x@N.

To better observe the position of the peaks, according to the reviewer's suggestion, we revised the **Supplementary Figure S3b** to make the colored graphs below the other graphs. In the original manuscript, the sample was exposed to air for an extended time before being measured, causing an inaccurate signal peak for the O 1s. To objectively present the test results, XPS depth measurements are employed to etch the surface of TNO and TNO_x@N samples for 60 s and tested two times. As displayed in **Figure R3**, the O 1s peaks in TNO_x@N are fitted with three peaks at 529.6/530.0, 531.2/531.6, and 532.7/533.1 eV, which are attributed to the lattice O (metal-oxygen bands), oxygen vacancies, and oxygen species in hydroxyl oxygen, respectively.[10-12] The peak area

ratio of O vacancy to lattice O for $\text{TNO}_{-x}\text{@N}$ (0.24/0.36) is much higher than that of TNO (0.12/0.31), confirming the plenty of oxygen vacancy introduced in $\text{TNO}_{-x}\text{@N}$. This can be also verified by the appearance of low-valent Nb^{4+} and Ti^{3+} XPS peaks in Nb 3d (**Figure R1a**) and Ti 2p (**Figure R1b**) spectrum and the reduction of the Nb valence state for $\text{TNO}_{-x}\text{@N}$ in the XANES profile (**Figure R2**).

Figure R3. High-resolution XPS O 1s spectra.

In addition, electron paramagnetic resonance (EPR) measurements are also supplemented to further detect the oxygen defects. As shown in **Figure R4**, $\text{TNO}_{-x}\text{@N}$ shows a distinct EPR signal peaks at $g=2.003$, whereas TNO exhibits a slight resonance. This result further proves that the thermal reduction process due to the ammonia annealing treatment leads to the appearance of bulk-phase oxygen vacancies in $\text{TNO}_{-x}\text{@N}$ in addition to surface vacancies.

Figure R4. EPR spectra of TNO and TNO_x@N.

References:

R4. Chen, J., Meng, J., Han, K., Liu, F., Wang, W., An, Q., Mai, L. Crystal structure regulation boosts the conductivity and redox chemistry of T-Nb₂O₅ anode material. *Nano Energy* **110**, 108377 (2023).

R5. Wu, D., Wang, F., Yang, H., Xu, Y., Zhuang, Y., Zeng, J., Yang, Y., Zhao, J. Realizing rapid electrochemical kinetics of Mg²⁺ in Ti-Nb oxides through a Li⁺ intercalation activated strategy toward extremely fast charge/discharge dual-ion batteries. *Energy Storage Mater.* **52**, 94-103 (2022).

R6. Gao, J. Z., Yang, L. T., Huang, C. H., Liang, G. S., Lei, Y., Li, S. J., Wang, W. A., Ou, Y. J., Gao, S. F., Liu, X. H., Cheng, Y. F., Zhang, J. C., Liu, Z. Z., Guo, A., Monteiro, Parreira, R., Ribas, L. R., Lin, C. F., Wu, L. M., Che, R. C. Sodium niobate with a large interlayer spacing: a fast-charging, long-life, and low-temperature friendly lithium-storage material. *Adv. Sci.* **10**, 2300583 (2023).

R7. Wang, H., Wei, Y., Wang, G., Pu, Y., Yuan, L., Liu, C., Wang, Q., Zhang, Y., Wu, H. Selective nitridation crafted a high-density, carbon-free heterostructure host with built-in electric field for enhanced energy density Li-S batteries. *Adv. Sci.* **9**, 2201823 (2022).

R8. Deng, S., Zhang, Y., Xie, D., Yang, L., Wang, G., Zheng, X., Zhu, J., Wang, X., Yu, Y., Pan, G., Xia, X., Tu, J. Oxygen vacancy modulated Ti₂Nb₁₀O_{29-x} embedded onto porous bacterial cellulose carbon for highly efficient lithium ion storage. *Nano Energy* **58**, 355-364 (2019).

R9. Wang, M., Zhao, G., Bai, X., Yu, W., Zhao, C., Gao, Z., Lyu, P., Chen, Z., Zhang, N. Gradient concentration refilling of N stabilizes oxygen vacancies for enhanced Zn²⁺ storage. *Adv. Energy Mater.* **13**, 2301730 (2023).

R10. Xin, S., Liu, T., Li, J., Cui, H., Liu, Y., Liu, K., Yang, Y., Wang, M. Coupling of oxygen vacancies and heterostructure on Fe₃O₄ via an anion doping strategy to boost catalytic activity for lithium-sulfur batteries. *Small* **19**, 2207924 (2023).

R11. Koo, B.-R., Ahn, H.-J., Fast-switching electrochromic properties of mesoporous WO₃ films with oxygen vacancy defects. *Nanoscale* **9**, 17788-17793 (2017).

R12. Zhang, Y., Zhang, M., Liu, Y., Zhu, H., Wang, L., Liu, Y., Xue, M., Li, B., Tao, X. Oxygen vacancy regulated TiNb₂O₇ compound with enhanced electrochemical performance used as anode material in Li-ion batteries. *Electrochim. Acta* **330**, 135299 (2020).

The detail has been added to the revised manuscript and Supplementary Information (**Supplementary Fig. 4** and **Fig. 5**), which is highlighted in yellow as shown below,

Revision in the manuscript

Page 6:

For the high-resolution Nb 3d spectrum of TNO_{-x}@N, two peaks at binding energies of 209.7 and 206.9 eV are assigned to Nb⁵⁺, while two additional small peaks attributed to the Nb⁴⁺ can be also observed at 209.1 and 206.0 eV (**Supplementary Fig. 4a**).^[24] In **Supplementary Fig. 4b**, two additional peak at 463.4 and 457.3 eV are detected in the Ti 2p spectra of TNO_{-x}@N in comparison to TNO, corresponding to the characteristic positions of Ti³⁺ cations.^[25] Compared to the TNO material, the Nb spectra in the TNO_{-x}@N shows an obvious blue shift of about 0.2 eV, which may be the presence of charge redistribution.^[26,27] Similar shift trend is also presented in the Ti 2p XPS spectra. X-ray absorption near-edge structure of TNO and TNO_{-x}@N are examined to confirm the existence of energy shift. In **Fig. 1c**, the XANES profile of TNO_{-x}@N lies between Nb foil and TNO, which indicates an average oxidation state of Nb that is intermediate between the two reference samples, further suggesting that an enhanced electron redistribution occurs after the ammonia annealing treatment. The O 1s spectrum in TNO_{-x}@N (**Supplementary Fig. 4c**) can be divided into three peaks located at 529.6, 531.2, and 532.7 eV, belonging to lattice O (metal-oxygen bands),

oxygen vacancies, and oxygen species in hydroxyl oxygen, respectively.^[28] The peak area ratio of O vacancy to lattice O for TNO_{-x}@N (0.24) is much higher than that of TNO (0.12), confirming the introduction of oxygen vacancies in TNO_{-x}@N. Electron paramagnetic resonance (EPR) measurements are carried out to further detect the oxygen defects. As shown in **Supplementary Fig. 5**, TNO_{-x}@N shows a distinct EPR signal peaks at $g=2.003$, whereas TNO exhibits a slight resonance. This result strongly proves that the thermal reduction process due to the ammonia annealing treatment leads to the appearance of bulk-phase oxygen vacancies in TNO_{-x}@N in addition to surface vacancies.

Fig. 1 ...c, K-edge XANES of Nb foil, TNO, and TNO_{-x}@N...

References:

Ref. 24 Chen, J., Meng, J., Han, K., Liu, F., Wang, W., An, Q., Mai, L. Crystal structure regulation boosts the conductivity and redox chemistry of T-Nb₂O₅ anode material. *Nano Energy* **110**, 108377 (2023).

Ref. 25 Wang, H., Wei, Y., Wang, G., Pu, Y., Yuan, L., Liu, C., Wang, Q., Zhang, Y., Wu, H. Selective

nitridation crafted a high-density, carbon-free heterostructure host with built-in electric field for enhanced energy density Li-S batteries. *Adv. Sci.* **9**, 2201823 (2022).

Ref. 26 Deng, S., Zhang, Y., Xie, D., Yang, L., Wang, G., Zheng, X., Zhu, J., Wang, X., Yu, Y., Pan, G., Xia, X., Tu, J. Oxygen vacancy modulated $\text{Ti}_2\text{Nb}_{10}\text{O}_{29-x}$ embedded onto porous bacterial cellulose carbon for highly efficient lithium ion storage. *Nano Energy* **58**, 355-364 (2019).

Ref. 27 Wang, M., Zhao, G., Bai, X., Yu, W., Zhao, C., Gao, Z., Lyu, P., Chen, Z., Zhang, N. Gradient concentration refilling of N stabilizes oxygen vacancies for enhanced Zn^{2+} storage. *Adv. Energy Mater.* **13**, 2301730 (2023).

Ref. 28 Xin, S., Liu, T., Li, J., Cui, H., Liu, Y., Liu, K., Yang, Y., Wang, M. Coupling of oxygen vacancies and heterostructure on Fe_3O_4 via an anion doping strategy to boost catalytic activity for lithium-sulfur batteries. *Small* **19**, 2207924 (2023).

Revision in the Supplementary Information

Page 7:

Fig. S4. The deconvoluted Nb 3d, Ti 2p, and O 1s XPS spectra. XPS analysis spectra of (a) Nb 3d, (b) Ti 2p, and (c) O 1s of TNO and $\text{TNO}_x@\text{N}$.

Page 8:

Fig. S5. EPR spectra of TNO and $\text{TNO}_x@\text{N}$.

3. The authors have used a “Finite element simulation” to capture the real-time Li⁺ concentration between the solid sphere and microflowers upon the lithiation process of the synthesized materials. If I am not mistaken, the authors do not fully explain in the Experimental Section on what and how this simulation is carried out. Furthermore, the authors have used this simulation under harsh lithiation (15 C discharge) conditions. What is the reason for this extensively high C-rate? This value must be explained.

Response 3: Thanks for the professional and valuable suggestion. We apologize for the absence of the specific simulation procedure in the Experimental Section of the original manuscript. The implement of electrochemical lithiation simulation was carried out on the surface of the solid sphere and microflowers electrode using the cubic current distribution physical field interface in the COMSOL Multiphysics software. The model is based on the calculation of currents in the electrolyte and electrodes using the "cubic current distribution" interface. Therefore, the electrolyte current is resolved according to Ohm's law, where one of the electrodes is grounded and the other is set to the battery potential to meet the total current condition.

Here, the electric field follows the continuity equation using the current density:

$$i = -F \sum -z_i^2 m_i F c_i \nabla \phi_i$$

where i is the current density vector; z_i is the ion charge; m_i is the mobility, F is the Faraday constant, ϕ_i is the ion potential, and c_i represents the concentration of the ion. Meanwhile, it is consistent with the conservation of current density.

$$\nabla \cdot i = 0$$

Expressions of the Butler-Volmer form are utilized in the simulations to describe the electrode kinetics occurring at the electrode surface inserted in the electrolyte, and the exchange current density for the oxidation reaction is considered to be concentration-dependent. The electrode surface current density is referred to the Butler-Volmer equation:

$$i_a = i_0 \left[\frac{c}{c_0} \exp\left(\frac{\eta(1-\beta)F}{RT}\right) - \exp\left(\frac{\eta\beta F}{RT}\right) \right]$$

The initial value of the electrolyte potential is set to be comparable to the potential of the cell at open circuit (the open circuit voltage). The overpotential is defined as follows:

$$\eta = \phi_s - \phi_l - E_{eq}$$

where η is the overpotential, ϕ_s is the electrode potential, ϕ_l is the electrolyte potential and E_{eq} is the equilibrium potential.

The transport of dissolved Li^+ in the electrolyte and electrode during charging and discharging is modeled by transient simulations of the "rare-mass transfer" interface, which assumes that the transport of ions can be described by diffusion according to Fick's law. Meanwhile, mass transfer induced by the diffusion and migration is considered.

$$-D\nabla C - zmFc\nabla\phi_l = N$$

in which c represents the concentration of the ions, z , D and m are the valence, diffusivity, and mobility, respectively. F and ϕ_l stand for the Faraday's constant and ionic potential.

The ionic concentration in the electrode material meets the depletion reaction process, which is modelled using the PDE module of the software:

$$\frac{\partial c}{\partial t} = -ck_s$$

where k_s is the rate of consumption reaction.

For the boundary setting, we set the bottom reference potential as zero potential, and the top as the average current density boundary 2.5 mA/cm^2 . In addition, and the conductivity of the electrolyte is set as 4.5 S/m . The diffusion coefficient of the ions as $5 \times 10^{-10} \text{ m}^2/\text{s}$, and the environmental concentration of the Li ions boundary as 1 mol/L .

In the beginning, we selected a current density of 15 C for the finite element simulations to capture the real-time Li^+ concentration distribution in the solid spheres and microflowers, due to the fact that we aimed to highlight the evolution of the Li^+ concentration distributions of the two models during the lithiation state in the case of fast charging. Generally, reactions at high C rates is relatively fast so that the sluggish mass transport that lithium ions cannot transport into the particle interior at a sufficient rate.

To further capture the real-time Li^+ concentration distribution between the solid spheres and microflowers, we supplemented the finite element simulations at the current density of 5 C . As shown in **Figure R5**, the Li^+ concentration in the solid sphere increases steadily from the outside in during lithiation. In contrast, the solid spheres still suffer from underutilization at up to the 100% depth-of-discharge (DOD) state on account of the low active area and restricted migration channels of the solid spheres,

which in agreement with the results simulated at the current density of 15 C. On the contrary, the Li^+ concentration of the microflowers model is more uniform compared to the solid sphere model during the discharge process. These results are further verified by the variation of Li^+ concentration distribution along the radial direction (**Figure R6**). Therefore, under the fast charging conditions (either at the current densities of 5 C or 15 C) in the manuscript, the microflowers possesses the ultrafast ion diffusion and storage capability, resulting in a good electrochemical performance.

Figure R5. Finite element simulation models of Li^+ concentration of solid sphere and microflowers with the different lithiation states at the current density of 5 C.

Figure R6. Evolution of Li^+ concentration in the (a) solid sphere and (b) microflowers at the current density of 5 C.

The detail has been added to the revised manuscript and Supplementary

Information (Supplementary Fig. 13 and Fig. 14), which is highlighted in yellow as shown below,

Revision in the manuscript

Page 9:

Inspired by this, the Finite element simulations are conducted to capture the real-time Li^+ concentration distribution between the solid sphere ($R=3 \text{ }\mu\text{m}$) and microflowers ($R=3 \text{ }\mu\text{m}$, $L=300 \text{ nm}$, $W=400 \text{ nm}$, $\delta=10 \text{ nm}$) upon the lithiation (5 C and 15 C discharge) process, where R , L , W , δ are the radius of a model particle, the length, the width, and the thickness of each nanosheet, respectively. As shown in Fig. 1i and Supplementary Fig. 13, the Li^+ concentration in the solid spheres steadily increases from the outside to the inside during lithiation on account of the lithium diffusion inwards from the entire periphery... Meanwhile, the inhomogeneity of the particles is more pronounced at the higher current densities. Such results are further verified by the evolution of the Li^+ concentration distribution achieved along the radial direction (Fig. 1k,l and Supplementary Fig. 14).

Fig. 1...i,j, Finite element simulation models of Li⁺ concentration of solid sphere (i) and microflowers (j) for different lithiation states at the current density of 5 C. **k,l**, Evolution of Li⁺ concentration in the solid sphere (k) and microflowers (l).

Revision in the Supplementary Information

Page 2:

Experiment Section

Finite element modeling (FEM) calculation: The implement of electrochemical lithiation simulation was carried out on the surface of the solid sphere and microflowers electrode using the cubic current distribution physical field interface in the COMSOL Multiphysics software. The model is based on the calculation of currents in the electrolyte and electrodes using the "cubic current distribution" interface. Therefore, the electrolyte current is resolved according to Ohm's law, where one of the electrodes is grounded and the other is set to the battery potential to meet the total current condition.

Here, the electric field follows the continuity equation using the current density:

$$i = -F \sum -z_i^2 m_i F c_i \nabla \phi_i$$

where i is the current density vector; z_i is the ion charge; m_i is the mobility, F is the Faraday constant, ϕ_i is the ion potential, and c_i represents the concentration of the ion. Meanwhile, it is consistent with the conservation of current density.

$$\nabla \cdot i = 0$$

Expressions of the Butler-Volmer form are utilized in the simulations to describe the electrode kinetics occurring at the electrode surface inserted in the electrolyte, and the exchange current density for the oxidation reaction is considered to be

concentration-dependent. The electrode surface current density is referred to the Butler-Volmer equation:

$$i_a = i_0 \left[\frac{c}{c_0} \exp\left(\frac{\eta(1-\beta)F}{RT}\right) - \exp\left(\frac{\eta\beta F}{RT}\right) \right]$$

The initial value of the electrolyte potential is set to be comparable to the potential of the cell at open circuit (the open circuit voltage). The overpotential is defined as follows:

$$\eta = \phi_s - \phi_l - E_{eq}$$

where η is the overpotential, ϕ_s is the electrode potential, ϕ_l is the electrolyte potential and E_{eq} is the equilibrium potential.

The transport of dissolved Li^+ in the electrolyte and electrode during charging and discharging is modeled by transient simulations of the "rare-mass transfer" interface, which assumes that the transport of ions can be described by diffusion according to Fick's law. Meanwhile, mass transfer induced by the diffusion and migration is considered.

$$-D\nabla C - zmFc\nabla\phi_l = N$$

in which c represents the concentration of the ions, z , D and m are the valence, diffusivity, and mobility, respectively. F and ϕ_l stand for the Faraday's constant and ionic potential.

The ionic concentration in the electrode material meets the depletion reaction process, which is modelled using the PDE module of the software:

$$\frac{\partial c}{\partial t} = -ck_s$$

where k_s is the rate of consumption reaction.

For the boundary setting, we set the bottom reference potential as zero potential, and the top as the average current density boundary 2.5 mA/cm^2 . In addition, the conductivity of the electrolyte is set as 4.5 S/m . The diffusion coefficient of the ions as $5 \times 10^{-10} \text{ m}^2/\text{s}$, and the environmental concentration of the Li ions boundary as 1 mol/L .

Page 16:

Fig. S13. Finite element simulation models of Li^+ concentration of solid sphere and microflowers with the different lithiation states at the current density of 15 C.

Page 17:

Fig. S14. Evolution of Li^+ concentration in the (a) solid sphere and (b) microflowers at the current density of 15 C.

4. In Figure 4d, the authors have used electrodes with high mass loading to compare the cycling performances of the different anodes. Cycling performance of the electrodes must be compared also under low mass loadings, as it is known that at low mass loadings the cycling performance of the electrodes decrease.

Response 4: Many thanks reviewer for this helpful suggestion. We agree with the reviewer that the cycling performance of the electrodes must also be compared under low mass loadings. In the original Supplementary Information, **Figure S18** compared the cycling performances of the three electrodes under the mass loadings of 1.5 mg cm^{-2} , where the $\text{TNO}_x@\text{N}$ electrode delivered the reversible capacity of $259.1 \text{ mA h g}^{-1}$ after 250 cycles, and the capacity retention rate was 97.5% at the current density of 1.0 C, which was obviously larger than those of the pure TNO flower ($252.2 \text{ mA h g}^{-1}$, 95.4%) and solid TNO ($220.3 \text{ mA h g}^{-1}$, 93.3%).

Fig. S18. Cyclic performance of the $\text{TNO}_x@\text{N}$, TNO and solid TNO under the mass loadings of 1.5 mg cm^{-2} .

According to the reviewer's suggestion, we also supplemented the cycling performances of the electrodes under the mass loadings of 1.0 mg cm^{-2} . **Figure R7** shows that the reversible specific capacity of the $\text{TNO}_x@\text{N}$ electrode preserves up to

258.5 mA h g⁻¹ with 97.2% capacity retention after 250 cycles, which is superior to that of the pure TNO flower (250.3 mA h g⁻¹, 93.6%) and solid TNO (224.7 mA h g⁻¹, 94.7%). This can be ascribed to the synergistic effect of N-incorporation and O-vacancies in TNO_{-x}@N microflowers can expand the interlayer distance for Li⁺ diffusion. In **Figure 4d** of the original manuscript, the thick TNO_{-x}@N electrode with the mass loadings of 10 mg cm⁻² exhibited a capacity retention of 82.3% with a discharge capacity of 185.9 mA h g⁻¹ at 1 C after 250 cycles. Notably, the cycling performance at low mass loadings is higher than that of thick electrodes, which is mainly attributed to the small volume expansion of Nb-based oxides (<10%), whereas thicker electrodes usually cause a proportional increase in ion/electron transport distance and resistance, accompanied by high polarization and low utilization, leading to decreased cycling stability.[13-15] Meanwhile, the thick electrode can lead to pulverisation of the lithium metal on the surface under the high current density of 1 C, thereby decreasing cycling performance.

Figure R7. Cyclic performance of the TNO_{-x}@N, TNO and solid TNO under the mass loadings of 1.0 mg cm⁻².

References:

R13. Yang, K., Yang, L., Wang, Z., Guo, B., Song, Z., Fu, Y., Ji, Y., Liu, M., Zhao, W., Liu, X., Yang, S., Pan, F. Constructing a highly efficient aligned conductive network to facilitate depolarized high-area-capacity electrodes in Li-ion batteries. *Adv. Energy Mater.* **11**, 2100601 (2021).

R14. Zhao, Z., Sun, M., Chen, W., Liu, Y., Zhang, L., Dong, N., Ruan, Y., Zhang, J., Wang, P., Dong, L., Xia, Y., Lu, H. *Adv. Funct. Mater.* **29**, 1809196 (2019).

R15. Zhang, M., Chouchane, M., Shojaee, S. A., Winiarski, B., Liu, Z., Li, L., Pelapur, R., Shodiev, A., Yao, W., Doux, J.-M., Wang, S., Li, Y., Liu, C., Lemmens, H., Franco, A. A., Meng, Y. S. Coupling of multiscale imaging analysis and computational modeling for understanding thick cathode degradation mechanisms. *Joule* **7**, 201-220 (2023).

The detail has been added to the revised manuscript and Supplementary Information (**Supplementary Fig. 27**), which is highlighted in yellow as shown below,

Revision in the manuscript

Page 18:

The thick TNO_x@N electrode exhibits a high capacity retention of 82.3% with an ultrastable capacity of 185.9 mA h g⁻¹ at 1 C after 250 cycles, which is much larger than that of the TNO (166.5 mA h g⁻¹, 75.3%, **Supplementary Fig. 27**) and solid TNO electrode (114.7 mA h g⁻¹, 56.8%).

Fig. 4... **d**, Long-term cycling stability of the TNO_x@N electrode with the mass loading of 1.5 mg cm⁻² and 10 mg cm⁻²...

Revision in the Supplementary Information

Page 30:

Fig. S27. Long-term cycling stability of the TNO and solid TNO electrode with the mass loading of 10 mg cm⁻².

5. Overall, the figures are very “messy” and very hard to see. The figures are generally all too small to see, and this must be fixed for it to be accepted in this journal. For example, Figure 1h shows elemental mappings of Nb, Ti, O in the synthesized materials, but which picture showing which element is excruciatingly hard to see. This is a crucial factor in deciding the acceptance of this manuscript in the journal, as the readability of a manuscript is a very important factor that must be considered before publishing a journal.

Response 5: We highly appreciate the reviewer for the kind reminder. Referring to your suggestion, we have carefully checked and proofed the entire manuscript, and the corresponding figures have been adjusted. The corresponding description has been revised in the manuscript and highlighted as yellow fonts, such as:

Revision in the manuscript

Page 8:

Fig. 1 ...**d**, TEM image of $\text{TNO}_x@\text{N}$. **e**, HR-TEM image of $\text{TNO}_x@\text{N}$. **f**, Aberration-corrected STEM-HAADF image of the $\text{TNO}_x@\text{N}$. **g**, Oxygen defect analysis with ABF image of the $\text{TNO}_x@\text{N}$. **h**, Atomic-resolution EDS mappings of the $\text{TNO}_x@\text{N}$...

Page 15:

Fig. 3 ...**a**, Schematic illustration of principles for femtosecond transient absorption spectroscopy. **b,c**, The pseudo color TA spectra plots of TNO (**b**) and $\text{TNO}_x@\text{N}$ (**c**) measured after excitation at 3.55 eV , $\sim 1.7 \text{ mJ cm}^{-2}$. The black lines represent the position of positive and negative boundaries of the TA signal. **d,e**, TA spectra at different time delays between pump and probe pulse of TNO (**d**)

and TNO_x@N (e). f, Comparison of extracted dynamics at 625 nm probe and bi-exponential function fit curve. g, The energy-band structure diagram and recombination processes of excited-state carriers in TNO_x@N...

Page 21:

Fig. 5 ...b, Synchrotron X-ray tomography reconstruction with volume rendering shows the 3D microstructure of the active particles, pores, and pore connectivity network diagram (color indicates local pore center). c, The EIS of the TNO_x@N, TNO and solid TNO at 25 and -40 °C. d, CV curve of half cells at various sweep rates of TNO_x@N at 25 °C. e, Capacitive contribution to the total capacity of TNO_x@N with the different mass loading.

Page 24:

Fig. 6 ...h-k Ex situ TEM and HRTEM images of the TNO_x@N at the discharge to 1.0 V (h,i) and the charge to 3.0 V (j,k). l, Elemental mapping images of the TNO_x@N at the discharged state of 1.0 V...

Revision in the Supplementary Information

Page 10:

Fig. S7... (b) TEM image, and (c) HRTEM image of TNO. (d) HAADF image and corresponding elemental mapping of TNO.

Page 46:

Fig. S43. HAADF and corresponding elemental mapping images of TNO_x@N at the charged state of 3.0 V.

Response to Reviewer #2

Overall comment: The manuscript entitled “Delocalized electronic engineering to reduce kinetic barrier enables ultralow temperature capability for high-areal-capacity batteries” describes excellent electrochemical performances of a TNO-based anode by surface modification and doping. Although they successfully synthesized well-structured TNO_x@N microflowers and performed in-depth studies of calculations, the concept of this work is not very new including a synthetic method, ammonia gas nitriding, defect engineering, etc. More importantly, the TNO anode material is not practical in the viewpoint of energy density. Therefore, I recommend a rejection of this manuscript. The detailed comments on the manuscript are as follows.

General response: Thank you very much for your constructive comments on improving the quality of our manuscript. We have studied the comments carefully and made corresponding corrections in the revised manuscript, and we sincerely hope the revised version is qualified for publication in Nature Communications. The main corrections in the paper and the responses to the reviewer’s comments are as follows:

Comment 1: There are many papers with the similar concept. Please cite them.

1.1. Material synthesis

[1] Carbon-coated TiNb₂O₇ nanosheet arrays as self-supported high mass-loading anode for flexible Li-ion battery, Chem. Commun., 2021, 57, 18221825

1.2. Gas nitriding

[1] Porosity-Controlled TiNb₂O₇ Microspheres with Partial Nitridation as A Practical Negative Electrode for High-Power Lithium-Ion Batteries. Adv. Energy Mater. 2015, 5,

1401945

[2] Surface nitrated and carbon coated TiNb_2O_7 anode material with excellent performance for lithium-ion batteries, *Journal of Alloys and Compounds*, Volume 835, 2020, 155241

[3] Porous TiNb_2O_7 nanofibers decorated with conductive $\text{Ti}_{1-x}\text{Nb}_x\text{N}$ bumps as a high power anode material for Li-ion batteries, *J. Mater. Chem. A* (2015)

1.3. Oxygen defect engineering

[1] Porous oxygen-deficient TiNb_2O_7 spheres wrapped by MXene as high-rate and durable anodes for liquid and all-solid-state lithium-ion batteries, *Chemical Engineering Journal*, 438 (2022), 135328

[2] Defect engineering of TiNb_2O_7 compound for enhanced Li-ion battery anode performances, *Electrochimica Acta*, Volume 404, 2022, 139603

Response 1: We sincerely thank the reviewer for the suggestions on adding important references. Actually, our work focuses on emphasizing the in-depth investigation of the mechanism for improvement of the low-temperature performance, which has been barely reported in the previous literature. Meanwhile, this work firstly employs femtosecond laser-based transient absorption spectroscopy to analyze carrier dynamics and electronic structure of TiNb_2O_7 and directly verify the DFT theoretical calculations, which establishes a direct graph of relations among the energy band structure, carrier dynamics, and intrinsic electronic conductivity. Indeed, as the reviewer suggested, the material synthesis, ammonia gas nitriding, and oxygen defect engineering in the manuscript are insufficiently novel. Meanwhile, we think it is equally important to

elucidate the relationship between the intrinsic structure of the material and its low-temperature performance. The synergistic effect of N-incorporation and O-vacancies in TNO- x @N microflowers can induce localized structure polarization, improving electron transfer, reducing Li⁺ diffusion energy barriers, and boosting the excellent fast-charging capability, surpassing the above-mentioned reported Nb-based anodes (details are shown in **Table R1**).^[1-6] In view of the demand for high-energy-density lithium-ion batteries, the cycling performance of the TNO- x @N electrode with the high mass loading are examined at low-temperature condition, which is more relevant to industrialization and provides guidance for the design of anode materials for commercial lithium-ion batteries in cold-region regions. Furthermore, the TNO- x @N-based pouch cell is assembled with commercial LiNi_{0.8}Co_{0.1}Mn_{0.1}O₂ as the cathode, delivering better electrochemical performance and safety, which makes this work more systematic.

We believe that this work not only provides a fundamental understanding of the intrinsic structure-function, but also puts forward a rational viewpoint for designing high-areal-capacity batteries in cold regions.

Table R1. Comparisons of electrochemical performance of the above-mentioned reported Nb-based anodes for LIBs.

Materials	Rate capability	Mass loading	Reference
TNO- x @N	50/60 C (193/187 mA h g ⁻¹)	1.5 mg cm ⁻² (25 °C)	This work
TNO- x @N	5/6 C (154/148 mA h g ⁻¹)	8 mg cm ⁻² (-30 °C)	This work
TNO nanosheets	30 C (96 mA h g ⁻¹)	/ (25 °C)	[1]
PTNO MS-2	50 C (~170 mA h g ⁻¹)	/ (25 °C)	[2]

N-TiNb ₂ O ₇ @C	50 C (135 mA h g ⁻¹)	/ (25 °C)	[3]
TNON NFs-2	50 C (196 mA h g ⁻¹) Discharge current fixed at 1C	0.2 mg cm ⁻² (25 °C)	[4]
TNO _{-x} @MXene	30 C (143 mA h g ⁻¹)	1 mg cm ⁻² (25 °C)	[5]
TNO-800-A	5 C (160 mA h g ⁻¹)	5 mg cm ⁻² (25 °C)	[6]

References:

- R1. Zhou, J., Dong, H., Chen, Y., Ye, Y., Xiao, L., Deng, B., Liu, J. Carbon-coated TiNb₂O₇ nanosheet arrays as self-supported high mass-loading anodes for flexible Li-ion batteries. *Chem. Commun.* **57**, 1822-1825 (2021).
- R2. Park, H., Wu, H. B., Song, T., Lou, X. W., Paik, U. Porosity-controlled TiNb₂O₇ microspheres with partial nitridation as a practical negative electrode for high-power lithium-ion batteries. *Adv. Energy Mater.* **5**, 1401945 (2015).
- R3. Gao, J., Cheng, X., Lou, S., Wu, X., Ding, F., Zuo, P., Ma, Y., Du, C., Gao, Y., Yin, G. Surface nitrated and carbon coated TiNb₂O₇ anode material with excellent performance for lithium-ion batteries. *J. Alloys Compd.* **835**, 155241 (2020).
- R4. Park, H., Song, T., Paik, U. Porous TiNb₂O₇ nanofibers decorated with conductive Ti_{1-x}Nb_xN bumps as a high power anode material for Li-ion batteries. *J. Mater. Chem. A* **3**, 8590-8596 (2015).
- R5. Wu, Y., Liu, D., Qu, D., Li, J., Xie, Z., Zhang, X., Chen, H., Tang, H. Porous oxygen-deficient TiNb₂O₇ spheres wrapped by MXene as high-rate and durable anodes for liquid and all-solid-state lithium-ion batteries. *Chem. Eng. J.* **438**, 135328 (2022).
- R6. Choi, H., Kim, T., Park, H. Defect engineering of TiNb₂O₇ compound for enhanced Li-ion battery anode performances. *Electrochim. Acta* **404**, 139603 (2022).

According to the reviewer's suggestion, we have cited the above-mentioned reported Nb-based anodes in the revised manuscript, which is highlighted in yellow as shown below,

Revision in the manuscript

Page 18:

A stable reversible capacity of 186.7 mA h g⁻¹ can still be achieved even at an extremely high current density of 60 C (60 s), which surpasses that of TNO, solid TNO, and the

majority of the recently reported Nb-based electrodes (**Fig. 4c**).^[29, 30, 48-60]

Fig. 4 ...c, Comparison of rate performance of TNO_x@N-based batteries with previously reported Nb-based electrodes...

References:

Ref.30 Park, H., Wu, H. B., Song, T., Lou, X. W., Paik, U. Porosity-controlled TiNb₂O₇ microspheres with partial nitridation as a practical negative electrode for high-power lithium-ion batteries. *Adv. Energy Mater.* **5**, 1401945 (2015).

Ref.56 Zhou, J., Dong, H., Chen, Y., Ye, Y., Xiao, L., Deng, B., Liu, J. Carbon-coated TiNb₂O₇ nanosheet arrays as self-supported high mass-loading anodes for flexible Li-ion batteries. *Chem. Commun.* **57**, 1822-1825 (2021).

Ref.57 Gao, J., Cheng, X., Lou, S., Wu, X., Ding, F., Zuo, P., Ma, Y., Du, C., Gao, Y., Yin, G. Surface nitrided and carbon coated TiNb₂O₇ anode material with excellent performance for lithium-ion batteries. *J. Alloys Compd.* **835**, 155241 (2020).

Ref.58 Park, H., Song, T., Paik, U. Porous TiNb₂O₇ nanofibers decorated with conductive Ti_{1-x}Nb_xN bumps as a high power anode material for Li-ion batteries. *J. Mater. Chem. A* **3**, 8590-8596 (2015).

Ref.59 Wu, Y., Liu, D., Qu, D., Li, J., Xie, Z., Zhang, X., Chen, H., Tang, H. Porous oxygen-deficient TiNb₂O₇ spheres wrapped by MXene as high-rate and durable anodes for liquid and all-solid-state lithium-ion batteries. *Chem. Eng. J.* **438**, 135328 (2022).

Ref.60 Choi, H., Kim, T., Park, H. Defect engineering of TiNb₂O₇ compound for enhanced Li-ion battery anode performances. *Electrochim. Acta* **404**, 139603 (2022).

Comment 2: The electrochemical performances of the TNO have been reported in many articles, and the reaction mechanism has also been studied by the DFT calculation.

Although the nitrogen doping and the oxygen vacancies can improve the electronic property and Li⁺ ion migration, the TNO itself shows a high rate capability and a stable cyclability. To improve the quality of this manuscript, the full cell with a high capacity cathode should be evaluated.

Response 2: We are delighted to receive this constructive comment. We agree with the reviewer that the full cell with a high capacity cathode should be evaluated. According to the reviewer's suggestion, we supplemented the 3.5 Ah-level pouch cells with the high mass loading of LiNi_{0.8}Co_{0.1}Mn_{0.1}O₂ as the cathode, as shown in **Figure R1a**. For the cathode electrode, the slurries consisting of LiNi_{0.8}Co_{0.1}Mn_{0.1}O₂ as the active materials, Kejten black and carbon nanotube as the conducting agent, and polyvinylidene fluoride (PVDF) binder in a weight ratio of 95.2:3:1.8 are prepared. For the anode electrode, the pastes consisting of LiNi_{0.8}Co_{0.1}Mn_{0.1}O₂ as the active materials, Kejten black and carbon nanotube as the conducting agent, and PVDF binder in a weight ratio of 92:5:3 are fabricated. The average mass loading of the single-sided anode and cathode electrode about are 15.4 and 16.1 mg cm⁻², respectively. In the voltage window (1.0–3.0 V, **Figure R1b**), the initial charge capacity of the TNO_{-x}@N-based pouch cell is 3.47 Ah at the current density of 0.2 C. Meanwhile, the pouch cell exhibits remarkable stability during cycling at 0.5 C rate, maintaining a capacity of 3.32 Ah after 55 cycles, with an impressive capacity retention ratio of 97.6% (**Figure R1c**), which demonstrates great potential for commercialization.

Figure R1. (a) Schematic illustration of the TNO- x @N|LiNi $_{0.8}$ Co $_{0.1}$ Mn $_{0.1}$ O $_2$ pouch cell; (b) Charge/discharge profiles of the TNO- x @N|LiNi $_{0.8}$ Co $_{0.1}$ Mn $_{0.1}$ O $_2$ pouch cell; (c) Cycling performance the TNO- x @N|LiNi $_{0.8}$ Co $_{0.1}$ Mn $_{0.1}$ O $_2$ pouch cell at 0.5 C after an activation process of 1 cycle at 0.2 C.

The detail has been added into the revised manuscript and Supplementary Information (**Supplementary Fig. 35**), which is highlighted in yellow as shown below,

Revision in the manuscript

Page 20:

Meanwhile, the 3.5 Ah-level pouch cells with commercial LiNi $_{0.8}$ Co $_{0.1}$ Mn $_{0.1}$ O $_2$ as the cathode is further tested, as shown in **Supplementary Fig. 35**. The TNO- x @N|LiNi $_{0.8}$ Co $_{0.1}$ Mn $_{0.1}$ O $_2$ pouch cell shows remarkable cycling reliability at 0.5 C, with a capacity of 3.32 Ah after 55 cycles and an impressive capacity retention of 97.6%. The calculated mass energy density and volumetric energy density of TNO- x @N anode are obtained as 271.6 Wh kg $^{-1}$ and 373.4 Wh L $^{-1}$, respectively, which are comparable to Nb-based materials, Ti-based materials, carbon materials (graphite), and silicon-based materials (**Supplementary Table 5**), showing certain potential for commercialization.

Revision in the Supplementary Information

Page 38:

Fig. S35. Electrochemical properties of TNO- x @N|LiNi $_{0.8}$ Co $_{0.1}$ Mn $_{0.1}$ O $_2$ Ah-level pouch cell. (a) Schematic illustration of the TNO- x @N|LiNi $_{0.8}$ Co $_{0.1}$ Mn $_{0.1}$ O $_2$ pouch cell; (b) Charge/discharge profiles of the TNO- x @N|LiNi $_{0.8}$ Co $_{0.1}$ Mn $_{0.1}$ O $_2$ pouch cell; (c) Cycling performance the TNO- x @N|LiNi $_{0.8}$ Co $_{0.1}$ Mn $_{0.1}$ O $_2$ pouch cell at 0.5 C after an activation process of 1 cycle at 0.2 C.

Comment 3: In my opinion, the merit of this work is the lithium storage performance at the low temperature. This is very important property of the commercial Li ion batteries. However, the anode with only TNO material does not shows a large impact because of the lower capacity. In this regard, it would be a better way to show a low temperature property of a mixture of the graphite and TNO.

Response 3: Thank the reviewer for raising such a detailed and suggestive comment.

We agree with the reviewer that a mixture of the graphite and TNO is interesting for improving reversible capacity at low temperatures. Meanwhile, the mixture of graphite and TNO can reduce the platform voltage, thereby improving the energy density of the battery, making Nb-based materials more commercialization prospects. Following the reviewer's helpful suggestion, we supplemented the electrochemical properties of graphite and TNO- x @N in a mass ratio of 1:1.

Firstly, the cycling performances of commercial graphite are evaluated. As shown in **Figure R2a**, the first-cycle Coulombic efficiency and reversible capacity of graphite are 89.4% and 369.1 mAh g $^{-1}$ at 0.2 C. When the current rate is increased to 0.5 C, the reversible capacities of graphite retain 355.2 mAh g $^{-1}$ after 30 cycles (**Figure R2b**).

Furthermore, graphite shows a discharge capacity of 207.2 mAh g⁻¹ after 200 cycles at 3 C (**Figure R2c**), showing better long-term cycling stability.

Figure R2. Electrochemical properties of commercial graphite at 25 °C. (a) Galvanostatic charge/discharge of commercial graphite at 0.2 C. Cyclic performance of commercial graphite at (b) 0.5 C and (c) 3.0 C.

As the temperature drops to -20 °C and -30 °C, graphite anodes undergo severe capacity degradation at 0.5 C (20 mAh g⁻¹, **Figure R3a,b**), which is attributed to the enlargement of electrode polarization, leading to the uncontrolled Li-metal plating and rapid capacity deterioration at low-temperature.[7,8] Until temperature further drops to -40 °C (**Figure R3c**), the graphite anode electrode basically cannot work.

Figure R3. Cyclic performance of commercial graphite at (a) -20 °C, (b) -30 °C and (c) -40 °C.

Subsequently, the electrochemical properties of graphite and TNO- x @N (denoted as G/TNO) in a mass ratio of 1:1 are investigated. The cyclic voltammetry (CV) experiments are performed on the Li|G/TNO cell within 0.01–3.0 V. As shown in **Figure R4a**, the G/TNO electrode shows the reduction/oxidation peak pairs of 1.63V/1.74 V, corresponding to the Nb $^{5+}$ /Nb $^{4+}$ redox, as well as the broad bump peaks at 1.0–1.4 V and 1.95–2.13 V, related to the Nb $^{4+}$ /Nb $^{3+}$ redox and the partial Ti $^{4+}$ /Ti $^{3+}$. [9,10] Meanwhile, the G/TNO anode shows a short potential plateau at around 0.7 V corresponding to the formation of a solid–electrolyte interphase (SEI) film. The subsequent appearance of several reduction/oxidation peak pairs at about 0.2 V are ascribed to the typical stagewise Li $^{+}$ insertion/extraction for graphite. [11] **Figure R4b** exhibits the galvanostatic charge curves of G/TNO at 0.2 C and 25 °C, where the discharge capacity of graphite is 358.8 mAh g $^{-1}$. At a rate of 0.5 C, G/TNO possesses a high initial discharge capacity of 271.1 mAh g $^{-1}$, and maintains an excellent reversible capacity of 273.4 mAh g $^{-1}$ after 50 cycles (**Figure R4c**). These results indicate that the mixture of graphite and TNO exhibits the high reversible capacity and excellent cycling stability at 25 °C.

Figure R4. Electrochemical properties of the G/TNO electrode at 25 °C. (a) CV profiles of the G/TNO electrode at 0.1 mVs $^{-1}$ within the potential range of 0.01–3 V during the initial 3 cycles. (b) Galvanostatic charge/discharge of the G/TNO electrode at 0.2 C. (c) Cyclic performance of the G/TNO electrode at 0.5 C.

When the temperature drops to $-20\text{ }^{\circ}\text{C}$ and $-30\text{ }^{\circ}\text{C}$, the reversible capacities of the G/TNO electrodes (about $40\text{-}50\text{ mAh g}^{-1}$, **Figure R5a, b**) at the current density of 0.5 C is higher than that of the commercial graphite electrodes (about 20 mAh g^{-1}), but also much lower than that of the $\text{TNO}_x\text{@N}$ electrodes (about 200 mAh g^{-1}). As the temperature further drops to $-40\text{ }^{\circ}\text{C}$, the G/TNO electrode still cannot deliver available capacity (**Figure R5c**). This suggests that the low-temperature performances of a mixture of the graphite and TNO are still unsatisfactory due to the large gap between the voltage plateau of TNO and graphite, which cannot avoid the uncontrolled Li plating of graphite at low temperatures. However, this problem is expected to be addressed by constructing an electrolyte of high conductivity electrolyte and an SEI of fast ionic diffusion ability at low temperatures, so that the mixture of the graphite and TNO graphite exhibits high reversible capacity and cycling stability at room temperatures, but the low-temperature performance is anticipated to be modified.

Figure R5. Cyclic performance of the G/TNO electrode at (a) $-20\text{ }^{\circ}\text{C}$, (b) $-30\text{ }^{\circ}\text{C}$ and (c) $-40\text{ }^{\circ}\text{C}$.

Therefore, we believe that the proposed strategy by the reviewer is a promising way to develop the high-rate anode material, providing a new mean for the design of low-temperature fast-charging lithium-ion batteries. We are also very grateful for the reviewer's constructive comments and insightful suggestions for further improving the manuscript's quality.

References:

- R7. Huang, Y., Wang, C., Lv, H., Xie, Y., Zhou, S., Ye, Y., Zhou, E., Zhu, T., Xie, H., Jiang, W., Wu, X., Kong, X., Jin, H., Ji, H. Bifunctional interphase promotes Li^+ de-solvation and transportation enabling fast-charging graphite anode at low temperature. *Adv. Mater.* 2308675 (2023).
- R8. Mo, Y., Liu, G., Chen, J., Zhu, X., Peng, Y., Wang, Y., Wang, C., Dong, X., Xia, Y. Unraveling the temperature-responsive solvation structure and interfacial chemistry for graphite anodes. *Energy Environ. Sci.* **17**, 227-237 (2024).
- R9. Aravindan, V., Sundaramurthy, J., Jain, A., Kumar, P. S., Ling, W. C., Ramakrishna, S., Srinivasan, M. P., Madhavi, S. Unveiling TiNb_2O_7 as an insertion anode for lithium ion capacitors with high energy and power density. *ChemSusChem* **7**, 1858-1863 (2014).
- R10. Gong, S., Wang, Y., Zhu, Q., Li, M., Wen, Y., Wang, H., Qiu, J., Xu, B. High-rate lithium storage performance of TiNb_2O_7 anode due to single-crystal structure coupling with Cr^{3+} -doping. *J. Power Sources* **564**, 232672 (2023).
- R11. Wang, M., Wang, J., Xiao, J., Ren, N., Pan, B., Chen, C.-s., Chen, C.-h. Introducing a pseudocapacitive lithium storage mechanism into graphite by defect engineering for fast-charging lithium-ion batteries. *ACS Appl. Mater. Interfaces* **14**, 16279-16288 (2022).

The detail has been added to the revised manuscript and Supplementary Information (**Supplementary Fig. 24**), which is highlighted in yellow as shown below,

Revision in the manuscript

Page 17:

Meanwhile, the mixture of graphite and $\text{TNO}_{-x}\text{@N}$ (denoted as G/TNO) according to a mass ratio of 1:1 demonstrates excellent cycling reliability at 0.5 C, with a high reversible capability of $273.4 \text{ mA h g}^{-1}$ and capacity retention of 100% over 50 cycles (**Supplementary Fig. 24**), further demonstrating that $\text{TNO}_{-x}\text{@N}$ holds a positive commercial prospect.

Revision in the Supplementary Information

Page 27:

Fig. S24. Electrochemical properties of the G/TNO electrode at 25 °C. (a) CV profiles of the G/TNO electrode at 0.1 mVs⁻¹ within the potential range of 0.01–3 V during the initial 3 cycles. (b) Galvanostatic charge/discharge of the G/TNO electrode at 0.2 C. (c) Cyclic performance of the G/TNO electrode at 0.5 C.

Overall, we are very grateful for the reviewer's constructive comments and suggestions for improving the quality of this manuscript. We have revised the manuscript with our greatest effort and best knowledge. We sincerely hope the reviewer can give us a chance to publish the paper on Nature Communications. If you have any further questions or concerns, please feel free to give the comments, and we will try our best to address them and improve the quality of the paper. Thank you!

Response to Reviewer #3

Overall comment: The article “Delocalized electronic engineering to reduce kinetic barrier enables ultralow temperature capability for high-area-capacity batteries” by Yan Zhang et al. expose delocalized electronic engineering in $\text{TiNb}_2\text{O}_7@\text{N}$ anodes. The characterization of this material has been carried on using novel techniques (femtosecond laser-based transient absorption spectroscopy, synchrotron X-ray tomography microscopy, et al), good carrier transport performance was confirmed. However, there are several core issues should be solved, I think the major revision is required before we can make a decision.

General response: Thank you very much for your positive and constructive comments concerning our manuscript. Those comments are all valuable and very helpful for improving the paper quality, as well as the important guiding significance to our research. Point-by-point response can be found below, and we hope our responses can well resolve your concerns and questions. Thank you again for your professional comments and precious time.

Comment 1: Introduction, paragraph 4, line 1-3

How do active electronic states boost ion-diffusion kinetics? I think there is no direct correlation between electronic conduction and ion conduction since the electronic conductivity is the directional transition of electrons, while Li-ions transport is the ion mass transfer process at the electrolyte/electrode interface.

Response 1: We deeply appreciate this constructive comment. We apologize for the absence of the discussion and explanation on the relationship between active electronic

states and ion-diffusion kinetics in the submitted manuscript. We agree with the reviewer that there is no direct correlation between electronic conduction and ion conduction. Actually, there is an indirect correlation between electronic conduction and ion conduction. The poor electronic conductivity of the active material can cause the electrode potential to deviate from the equilibrium potential during charging and discharging, generating a larger voltage polarization (especially at high current densities), which will decrease the effective potential and reduce the available Li ions.[1] Meanwhile, the Li-ion migration barrier can be lowered by the increment of polarons concentration due to the strong screening experienced by Li-ion during migration through polarons.[2,3] In addition, the poor intrinsic conductivity (almost an insulator) of the active material gives rise to large kinetic barriers during the initial insertion of lithium ions, resulting in an insufficient ability to accommodate the fast-passing Li⁺ during the solid-solution transformation process, especially under low temperatures and fast-charging conditions.[4,5] Therefore, improving the intrinsic conductivity of the TNO material is conducive to reducing the initial reaction barrier, thereby promoting the low-temperature dynamics for Li⁺ diffusion during solid-solution transformation.

References:

- R1. Xu, S., Tan, X., Ding, W., Ren, W., Zhao, Q., Huang, W., Liu, J., Qi, R., Zhang, Y., Yang, J., Zuo, C., Ji, H., Ren, H., Cao, B., Xue, H., Gao, Z., Yi, H., Zhao, W., Xiao, Y., Zhao, Q., Zhang, M., Pan, F. Promoting surface electric conductivity for high-rate LiCoO₂. *Angew. Chem. Int. Ed.* **62**, e202218595 (2023).
- R2. Kick, M., Scheurer, C., Oberhofer, H. Polaron-assisted charge transport in Li-ion battery anode materials. *ACS Appl. Mater. Interfaces* **4**, 8583-8591 (2021).
- R3. Nasara, R. N., Tsai, P.-c., Lin, S.-k. One-step synthesis of highly oxygen-deficient lithium titanate oxide with conformal amorphous carbon coating as anode material for lithium ion batteries. *Adv. Mater. Interfaces* **4**, 1700329 (2017).

R4. Zhang, J., Yan, Y., Wang, X., Cui, Y., Zhang, Z., Wang, S., Xie, Z., Yan, P., Chen, W. Bridging multiscale interfaces for developing ionically conductive high-voltage iron sulfate-containing sodium-based battery positive electrodes. *Nat Commun.* **14**, 3701 (2023).

R5. Liu, Y., Qiu, M., Hu, X., Yuan, J., Liao, W., Sheng, L., Chen, Y., Wu, Y., Zhan, H., Wen, Z. Anion defects engineering of ternary Nb-based chalcogenide anodes toward high-performance sodium-based dual-ion batteries. *Nano-Micro Lett.* **15**, 104 (2023).

Based on the above discussion, the detail has been added to the revised manuscript, which is highlighted in yellow as shown below,

Revision in the manuscript

Page 3:

The large bandgap of TiNb_2O_7 gives rise to large kinetic barriers during the first insertion of lithium ions, resulting in an insufficient ability to accommodate the fast-passing Li^+ during the solid solution transformation process, especially under low temperatures.^[14,15] Meanwhile, the poor electronic conductivity can cause the electrode potential to deviate from the equilibrium potential during cycling, generating a larger voltage polarization (especially at fast-charging conditions), which will decrease the effective potential and reduce the available Li ions.

References:

Ref 14. Xu, S., Tan, X., Ding, W., Ren, W., Zhao, Q., Huang, W., Liu, J., Qi, R., Zhang, Y., Yang, J., Zuo, C., Ji, H., Ren, H., Cao, B., Xue, H., Gao, Z., Yi, H., Zhao, W., Xiao, Y., Zhao, Q., Zhang, M., Pan, F. Promoting surface electric conductivity for high-rate LiCoO_2 . *Angew. Chem. Int. Ed.* **62**, e202218595 (2023).

Ref 15. Zhang, J., Yan, Y., Wang, X., Cui, Y., Zhang, Z., Wang, S., Xie, Z., Yan, P., Chen, W. Bridging multiscale interfaces for developing ionically conductive high-voltage iron sulfate-containing sodium-based battery positive electrodes. *Nat Commun.* **14**, 3701 (2023).

Comment 2: Introduction, paragraph 4

It is recommended not to mention high mass loading or high areal capacity as your

advantages in this TNO@N anode. Scheme 1 only shows that an increased specific surface area and a shortened diffusion path can enhance the dynamics of lithium ion transfer. The material you synthesized is a “flower” spherical material with micro size, and I do not believe that this structure has a high packing density because the “flowers” composed of nanosheets cannot be compacted, and there are also large pores between adjacent particles. The manuscript did not directly provide a comparison value between mass loading/areal capacity with commercial materials. Your high mass loading is mainly contributed by high-density metals Ti and Nb, but its thickness is actually not significant. The areal capacity of commercial anode electrode materials ranges from 3.0 to 3.5 mA h cm⁻², while your material is only 1.3 mA h cm⁻².

Response 2: Thank you for your valuable comment. We agree with the reviewer that the flowers-like structure composed of nanosheets has a low packing density due to the large pores between adjacent particles. We are so sorry that it is hard to compare the areal capacity of TNO@N anode to commercial TNO anode because no commercial material for this system, so it is only comparable to commercial graphite. The areal capacity of commercial graphite anode electrode materials ranges from 3.0 to 3.5 mAh cm⁻². In **Figure S22** of the original Supplementary Information, the cycling performance of the TNO_x@N electrode with the mass loading of 20 mg cm⁻² were evaluated. During the first cycle at 0.1 C, the areal capacity retained 4.4 mAh cm⁻², which was comparable to or even higher than the areal capacities of reported graphite anodes.[6-13] After 80 cycles at 0.2 C, it still remains 3.35 mAh cm⁻² with the capacity retention of 79.4%.

Fig. S22. Electrochemical properties of TNO- x @N with high active-material loadings of 20 mg cm^{-2} at $25 \text{ }^\circ\text{C}$ after an activation process of 2 cycles at 0.1 C . (a) Charge–discharge curves and (b) cyclability at 0.2 C of TNO- x @N half coin cell.

According to the reviewer's suggestion, we supplemented the cycling performances of the electrodes with the mass loadings of 20 mg cm^{-2} at $-30 \text{ }^\circ\text{C}$ and $-40 \text{ }^\circ\text{C}$. As displayed in **Figure R1a**, the TNO- x @N electrode exhibits the initial areal capacity of 3.14 mAh cm^{-2} at the current density of 0.1 C and $-30 \text{ }^\circ\text{C}$. **Figure R1b** shows the areal capacity of 3.0 mAh cm^{-2} after 50 cycles. Furthermore, the thick TNO- x @N electrode exhibits an areal capacity of 2.57 mAh cm^{-2} at the current density of 0.2 C and $-30 \text{ }^\circ\text{C}$ after 50 cycles, corresponding to a capacity retention rate of 84.9% (**Figure R1c**).

Figure R1. Electrochemical properties of TNO- x @N with the high active-material loadings of 20 mg cm^{-2} at $-30 \text{ }^\circ\text{C}$. (a) Charge–discharge curve of TNO- x @N, Cycling performance of TNO- x @N half coin cell at (b) 0.1 C and (c) 0.2 C .

When the temperature further drops to $-40 \text{ }^\circ\text{C}$, the areal capacity of the Li|TNO- x @N cell can still retain 2.36 mAh cm^{-2} with the high loading of 20 mg cm^{-2} at 0.1 C (**Figure R2a**). As shown in **Figure R2b**, the areal capacity of 2.23 mAh cm^{-2} is

obtained after 50 cycles. Moreover, TNO- x @N electrode can maintain a considerable areal capacity of 1.86 mAh cm $^{-2}$ after 50 cycles at 0.2 C with the mass loading of 20 mg cm $^{-2}$ (Figure R2c).

Figure R2. Electrochemical properties of TNO- x @N with the high active-material loadings of 20 mg cm $^{-2}$ at -40 °C. (a) Charge–discharge curve of TNO- x @N, Cycling performance of TNO- x @N half coin cell at (b) 0.1 C and (c) 0.2 C.

References:

- R6. Xia, D., Kamphaus, E. P., Hu, A., Hwang, S., Tao, L., Sainio, S., Nordlund, D., Fu, Y., Huang, H., Cheng, L., Lin, F. Design criteria of dilute ether electrolytes toward reversible and fast intercalation chemistry of graphite anode in Li-ion batteries. *ACS Energy Lett.* **8**, 1379-1389 (2023).
- R7. Sarkar, A., Shrotriya, P., Nlebedim, I. C. Parametric analysis of anodic degradation mechanisms for fast charging lithium batteries with graphite anode. *Comp. Mater. Sci.* **202**, 110979 (2022).
- R8. Han, D.-Y., Han, I. K., Son, H. B., Kim, Y. S., Ryu, J., Park, S. Layering charged polymers enable highly integrated high-capacity battery anodes. *Adv. Funct. Mater.* **33**, 2213458 (2023).
- R9. Xu, C., Shen, L., Zhang, W., Huang, Y., Sun, Z., Zhao, G., Lin, Y., Zhang, Q., Huang, Z., Li, J. Efficient implementation of kilogram-scale, high-capacity and long-life Si-C/TiO $_2$ anodes. *Energy Storage Mater.* **56**, 319-330 (2023).
- R10. Li, Z., Wu, G., Yang, Y., Wan, Z., Zeng, X., Yan, L., Wu, S., Ling, M., Liang, C., Hui, K. N., Lin, Z. An ion-conductive grafted polymeric binder with practical loading for silicon anode with high interfacial stability in lithium-ion batteries. *Adv. Energy Mater.* **12**, 2201197 (2022).
- R11. Li, Y., Jin, B., Wang, K., Song, L., Ren, L., Hou, Y., Gao, X., Zhan, X., Zhang, Q. Coordinatively-intertwined dual anionic polysaccharides as binder with 3D network conducive for stable SEI formation in advanced silicon-based anodes. *Chem. Eng. J.* **429**, 132235 (2022).
- R12. Zeng, W., Wang, L., Peng, X., Liu, T., Jiang, Y., Qin, F., Hu, L., Chu, P. K., Huo, K., Zhou, Y. Enhanced ion conductivity in conducting polymer binder for high-performance silicon anodes in advanced lithium-ion batteries. *Adv. Energy Mater.* **8**, 1702314 (2018).
- R13. Song, J., Zhou, M., Yi, R., Xu, T., Gordin, M. L., Tang, D., Yu, Z., Regula, M., Wang, D. Interpenetrated gel polymer binder for high-performance silicon anodes in lithium-ion batteries. *Adv. Funct. Mater.* **24**, 5904-5910 (2014).

The detail has been added to the revised manuscript and Supplementary Information (**Supplementary Fig. 32** and **Fig. 33**), which is highlighted in yellow as shown below,

Revision in the manuscript

Page 19:

Even if the mass loading is increased to 20 mg cm^{-2} , the areal capacity is up to 4.4 mAh cm^{-2} (**Supplementary Fig. 22**), which is comparable to or even higher than the areal capacities of reported graphite and Si-based anode.^[64-70] After 80 cycles at 0.2 C, it still remains 3.35 mAh cm^{-2} with the capacity retention of 79.4%.

Page 19:

In addition, the cycling performance of TNO_x@N electrode with the high mass loading of 20 mg cm^{-2} at $-30 \text{ }^\circ\text{C}$ (**Supplementary Fig. 32**) and $-40 \text{ }^\circ\text{C}$ (**Supplementary Fig. 33**). The TNO_x@N electrode exhibits the initial areal capacity of 3.14 mAh cm^{-2} and 2.36 mAh cm^{-2} at the current density of 0.1 C. Furthermore, after 50 cycles at 0.2 C, the areal capacity of 2.57 mAh cm^{-2} and 1.86 mAh cm^{-2} are obtained.

References:

R64. Xia, D., Kamphaus, E. P., Hu, A., Hwang, S., Tao, L., Sainio, S., Nordlund, D., Fu, Y., Huang, H., Cheng, L., Lin, F. Design criteria of dilute ether electrolytes toward reversible and fast intercalation chemistry of graphite anode in Li-ion batteries. *ACS Energy Lett.* **8**, 1379-1389 (2023).

R65. Sarkar, A., Shrotriya, P., Nlebedim, I. C. Parametric analysis of anodic degradation mechanisms for fast charging lithium batteries with graphite anode. *Comp. Mater. Sci.* **202**, 110979 (2022).

R66. Han, D.-Y., Han, I. K., Son, H. B., Kim, Y. S., Ryu, J., Park, S. Layering charged polymers enable highly integrated high-capacity battery anodes. *Adv. Funct. Mater.* **33**, 2213458 (2023).

R67. Xu, C., Shen, L., Zhang, W., Huang, Y., Sun, Z., Zhao, G., Lin, Y., Zhang, Q., Huang, Z., Li, J. Efficient implementation of kilogram-scale, high-capacity and long-life Si-C/TiO₂ anodes. *Energy Storage Mater.* **56**, 319-330 (2023).

R68. Li, Z., Wu, G., Yang, Y., Wan, Z., Zeng, X., Yan, L., Wu, S., Ling, M., Liang, C., Hui, K. N., Lin, Z. An ion-conductive grafted polymeric binder with practical loading for silicon anode with high interfacial stability in lithium-ion batteries. *Adv. Energy Mater.* **12**, 2201197 (2022).

R69. Li, Y., Jin, B., Wang, K., Song, L., Ren, L., Hou, Y., Gao, X., Zhan, X., Zhang, Q. Coordinatively-intertwined dual anionic polysaccharides as binder with 3D network conducive for stable SEI formation in advanced silicon-based anodes. *Chem. Eng. J.* **429**, 132235 (2022).

R70. Zeng, W., Wang, L., Peng, X., Liu, T., Jiang, Y., Qin, F., Hu, L., Chu, P. K., Huo, K., Zhou, Y. Enhanced ion conductivity in conducting polymer binder for high-performance silicon anodes in advanced lithium-ion batteries. *Adv. Energy Mater.* **8**, 1702314 (2018).

Revision in the Supplementary Information

Page 35:

Fig. S32. Electrochemical properties of TNO_x@N with the high active-material loadings of 20 mg cm⁻² at -30 °C. (a) Charge–discharge curve of TNO_x@N, Cycling performance of TNO_x@N half coin cell at (b) 0.1 C and (c) 0.2 C.

Page 36:

Fig. S33. Electrochemical properties of TNO_x@N with high active-material loadings of 20 mg cm⁻² at -40 °C. (a) Charge–discharge curve of TNO_x@N, Cycling performance of TNO_x@N half coin cell at (b) 0.1 C and (c) 0.2 C.

Comment 3: Results and Discussion, Morphology, structure and kinetics of the TNO@N., paragraph 2, line 3-12

(1) Does N 1s exist in the form of free radicals or through bonding? Please provide

specific instructions. Your explanation is very vague.

Response 3 (1): We thank the reviewer for this constructive comment. It's our negligence that the discussion of the existence of N 1s in the form of free radicals or through bonding is deficient in the original manuscript. From the HRTEM image of the original manuscript in **Fig. 1e**, it can be seen that there is an amorphous layer with a thickness of ~ 8 nm in the edge region, which may be attributed to the partial nitriding of the surface TNO material due to the outward-inward expansion of the NH_3 heat treatment.[14, 15] Meanwhile, there are no obvious characteristic peaks in the XRD pattern in **Fig. 1b**, which may be attributed to the low nitride content and the generation of amorphous nitrides. Therefore, we believe that N 1s exist in the form of through bonding.

Fig. 1 ...b, Powder XRD and Refined XRD... e, HR-TEM image of TNO-x@N...

References:

R14. Yang, Y., Zhu, H., Yang, F., Yang, F., Chen, D., Wen, Z., Wu, D., Ye, M., Zhang, Y., Zhao, J., Liu, Q., Lu, X., Gu, M., Li, C. C., He, W. Ten thousand-cycle ultrafast energy storage of Wadsley-

Roth phase Fe–Nb oxides with a desolvation promoting interfacial layer. *Nano Lett.* **21**, 9675-968 (2021).

R15. Park, H., Wu, H. B., Song, T., Lou, X. W., Paik, U. Porosity-controlled TiNb₂O₇ microspheres with partial nitridation as a practical negative electrode for high-power lithium-ion batteries. *Adv. Energy Mater.* **5**, 1401945 (2015).

The detail has been added to the revised manuscript, which is highlighted in yellow as shown below,

Revision in the manuscript

Page 9:

Note that an amorphous layer with a thickness of ~8 nm can be observed in the edge region, which may be attributed to the partial nitriding of the surface TNO material due to the outward-inward expansion of the NH₃ heat treatment.^[29,30] Meanwhile, there are no obvious characteristic peaks in the XRD pattern in **Fig. 1b**, which may be attributed to the low nitride content and the generation of amorphous nitrides.

References:

Ref 29. Yang, Y., Zhu, H., Yang, F., Yang, F., Chen, D., Wen, Z., Wu, D., Ye, M., Zhang, Y., Zhao, J., Liu, Q., Lu, X., Gu, M., Li, C. C., He, W. Ten thousand-cycle ultrafast energy storage of Wadsley–Roth phase Fe–Nb oxides with a desolvation promoting interfacial layer. *Nano Lett.* **21**, 9675-968 (2021).

Ref 30. Park, H., Wu, H. B., Song, T., Lou, X. W., Paik, U. Porosity-controlled TiNb₂O₇ microspheres with partial nitridation as a practical negative electrode for high-power lithium-ion batteries. *Adv. Energy Mater.* **5**, 1401945 (2015).

(2) Is the signal of O 1s at 531.9 eV an oxygen hole? Or hydroxyl oxygen? The original text states that this location contains these two types of signal samples, it is not right.

Response 3 (2): Many thanks reviewer for this suggestion. We are so sorry for the absence of the detailed explanation for the signal of O 1s. In the original manuscript, the sample was exposed to air for an extended period of time before being measured,

causing a partially polluted sample surface, allowing for an inaccurate signal peak for the O 1s due to that XPS analysis depth is merely a few nanometers. To objectively present the test results, XPS depth measurements are employed to etch the surface of TNO and TNO- x @N samples for 60 s and tested two times. As displayed in **Figure R3**, the O1s peaks in TNO- x @N are fitted with three peaks at 529.6/530.0, 531.2/531.6, and 532.7/533.1 eV, which are attributed to the lattice O (metal–oxygen bands), oxygen vacancies, and oxygen species in hydroxyl oxygen, respectively.[16-18] The peak area ratio of O vacancy to lattice O for TNO- x @N (0.24/0.36) is much higher than that of TNO (0.12/0.31), confirming the plenty of oxygen vacancy introduced in TNO- x @N.

Figure R3. High-resolution XPS O1s spectra.

In addition, electron paramagnetic resonance (EPR) measurements are also supplemented to further detect the oxygen defects. As shown in **Figure R4**, TNO- x @N shows a distinct EPR signal peak at $g=2.003$, whereas TNO exhibits a slight resonance. This result further proves that the thermal reduction process in the ammonia annealing treatment leads to the appearance of bulk-phase oxygen vacancies in TNO- x @N in addition to surface vacancies.

Figure R4. EPR spectra of TNO and TNO_x@N.

References:

- R16. Xin, S., Liu, T., Li, J., Cui, H., Liu, Y., Liu, K., Yang, Y., Wang, M. Coupling of oxygen vacancies and heterostructure on Fe₃O₄ via an anion doping strategy to boost catalytic activity for lithium-sulfur batteries. *Small* **19**, 2207924 (2023).
- R17. Koo, B.-R., Ahn, H.-J., Fast-switching electrochromic properties of mesoporous WO₃ films with oxygen vacancy defects. *Nanoscale* **9**, 17788-17793 (2017).
- R18. Zhang, Y., Zhang, M., Liu, Y., Zhu, H., Wang, L., Liu, Y., Xue, M., Li, B., Tao, X. Oxygen vacancy regulated TiNb₂O₇ compound with enhanced electrochemical performance used as anode material in Li-ion batteries. *Electrochim. Acta* **330**, 135299 (2020).

The detail has been added to the revised manuscript and Supplementary Information (**Supplementary Fig. 4** and **Fig. 5**), which is highlighted in yellow as shown below,

Revision in the manuscript

Page 7:

The O 1s spectrum (**Supplementary Fig. 4c**) in TNO_x@N can be divided into three peaks located at 529.6, 531.2, and 532.7 eV, belonging to lattice O (metal–oxygen bands), oxygen vacancies, and oxygen species in hydroxyl oxygen, respectively.^[28] The peak area ratio of O vacancy to lattice O for TNO_x@N (0.24) is much higher than that of TNO (0.12), confirming the introduction of oxygen vacancies in TNO_x@N. Electron

paramagnetic resonance (EPR) measurements are carried out to further detect the oxygen defects. As shown in **Supplementary Fig. 5**, TNO- x @N shows a distinct EPR signal peak at $g=2.003$, whereas TNO exhibits a slight resonance. This result strongly proves that the thermal reduction process in the ammonia annealing treatment leads to the appearance of bulk-phase oxygen vacancies in TNO- x @N in addition to surface vacancies.

References:

Ref 28. Xin, S., Liu, T., Li, J., Cui, H., Liu, Y., Liu, K., Yang, Y., Wang, M. Coupling of oxygen vacancies and heterostructure on Fe₃O₄ via an anion doping strategy to boost catalytic activity for lithium-sulfur batteries. *Small* **19**, 2207924 (2023).

Revision in the Supplementary Information

Page 7:

Fig. S4. ... (c) O 1s of TNO and TNO- x @N...

Revision in the Supplementary Information

Page 8:

Fig. S5. EPR spectra of TNO and TNO_x@N.

(3) The change in charge distribution can be determined based on the binding energy according to XPS. However, there is no correlation between the charge distribution and the increase in conductivity. The increase in conductivity should be caused by the change in TNO band structure caused by the introduction of N components into the material. We suggest adjusting the conductivity data to the relevant position, such as the band data in Fig. 2.

Response 3 (3): We are delighted to receive this constructive comment. We agree with the reviewer's comment to adjust the conductivity data to the band data in Fig. 2. The detail has been modified to the revised manuscript (**Fig. 2e**), which is highlighted in yellow as shown below,

Revision in the manuscript

Page 12:

The electronic conductivities of the TNO and TNO_x@N powders are evaluated by the direct polarization test. As shown in **Fig. 2e**, the electrical conductivity of TNO_x@N is $1.47 \times 10^{-3} \text{ S cm}^{-1}$, nearly 5 orders of magnitude higher than that of TNO ($4.29 \times 10^{-8} \text{ S cm}^{-1}$).

Fig. 2 ... e, The electronic conductivities of TNO and $TNO_x@N$...

Comment 4: Results and Discussion, Morphology, structure and kinetics of the $TNO_x@N$, paragraph 4

Providing the specific surface area and pore size distribution data of TNO, otherwise, the only mentions interlayer spacing of 0.372 nm cannot support your view since it remains unchanged before and after doping N.

Response 4: Thank you for your valuable comment. We agree with the reviewer that the specific surface area and pore size distribution data of TNO should be provided in the manuscript. In the original **Supplementary Fig. S9**, the Brunauer–Emmett–Teller (BET) results exhibit that the specific surface area of $TNO_x@N$ microflowers reaches $26.42\ m^2\ g^{-1}$, which is larger than the TNO ($20.76\ m^2\ g^{-1}$) and solid TNO ($1.20\ m^2\ g^{-1}$).

In addition, the pore size distributions of TNO- x @N are mainly concentrated at around 20-40 nm according to the Barrett–Joyner–Halenda (BJH) results, showing a mesoporous structure. Such a striking feature provides a large surface-to-volume ratio compared to TNO and bulk samples, enabling abundant Li⁺ storage sites and excellent transport ability in the TNO- x @N sample.

Fig. S9. N₂ adsorption/desorption isotherms and pore size distribution. (a, b) TNO- x @N, (c, d) TNO, and (e, f) solid TNO.

The detail has been added to the revised manuscript, which was highlighted in yellow as shown below,

Revision in the manuscript

Page 7:

According to the Barrett–Joyner–Halenda (BJH) results the pore size distributions of TNO- x @N are mainly concentrated at around 20-40 nm, further proving a mesoporous structure (**Supplementary Fig. 10**). In addition, the Brunauer–Emmett–Teller (BET) results exhibit that the specific surface area of TNO- x @N microflowers reaches 26.42 m² g⁻¹, which is larger than the TNO (20.76 m² g⁻¹) and solid TNO (1.20 m² g⁻¹).

Comment 5: Results and Discussion, Theoretical insight into the electron

delocalization effect., paragraph 1

We do not observe any changes in electron delocalization state in Fig. 2c? No significant differences were observed except for the color difference in a small area at the center. Please provide a specific explanation.

Response 5: Many thanks to the reviewer for the constructive comment. Indeed, we acknowledge that there was an oversight in the discussion and explanation of the electron delocalization state in the original manuscript. In **Fig. 2c**, the area surrounded by the ELF's equivalent surface of N in TNO- x @N is larger than that of O in TNO, indicating that the electrons are delocalized inside and there are more free electrons in TNO- x @N.[19,20]

Considering the reviewer's comment, we supplemented the ELF diagram of the (30 $\bar{1}$) crystal plane of TNO- x @N and TNO to further prove the electron delocalization state. From ELF plots in **Figure R5**, the ELF's equivalent surface of N in TNO- x @N is also larger than that of O in TNO, which confirms the fewer localized electrons in TNO- x @N. In addition, we also supplemented the Bader charge number of active sites for TNO- x @N and TNO to further explain the delocalization effect. As shown in **Figure R6**, the Nb atoms neighboring the N atoms in the TNO- x @N have a lower Bader charge number than the Nb atoms neighboring the O atoms in the TNO, indicating that N atoms have a weak electron absorption ability, and electrons can be excited more easily to form free electrons and holes, making the electrons more delocalized. This may be conducive to the improvement of the electrical conductivity of TNO materials and reduce the initial reaction barrier.

Figure R5. The corresponding ELF plots of (a) TNO and (b) TNO_{-x}@N.

Figure R6. The Bader charge numbers of atoms of (a) TNO and (b) TNO_{-x}@N. The red, green, blue, and gray balls represent O, Nb, Ti, and N atoms, respectively.

References:

R19. Han, X., Zhao, L., Wang, J., Liang, Y., Zhang, J. Delocalized electronic engineering of Ni₅P₄ nanoroses for durable Li–O₂ batteries. *Adv. Mater.* **35**, 2301897 (2023).

R20. Masana, J. J., Xiao, J., Zhang, H., Lu, X., Qiu, M., Yu, Y. Nitrogen-rich carbon nitride inducing electron delocalization of Co-N₄ site to enhance electrocatalytic carbon dioxide reduction. *Appl. Catal. B-Environ.* **323**, 122199 (2023).

The detail has been added to the revised manuscript and Supplementary Information (**Supplementary Fig. 18**), which is highlighted in yellow as shown below,

Revision in the manuscript

Page 11:

The corresponding electron localization functions (ELF) of TNO_{-x}@N and TNO (**Fig. 2c** and **Supplementary Fig. 18**) show that the electron delocalization is caused by the weak interactions between the TiNb₂O₇ and N-terminal with low electronegativity, indicating that N atoms have a weak electron absorption ability, and the electron localization is weaker and more free electrons are generated in TNO_{-x}@N, which may

be conducive to the improvement of the electrical conductivity of TNO materials and reduce the initial reaction barrier.^[32,33]

References:

Ref 32. Han, X., Zhao, L., Wang, J., Liang, Y., Zhang, J. Delocalized electronic engineering of Ni₅P₄ nanoroses for durable Li–O₂ batteries. *Adv. Mater.* **35**, 2301897 (2023).

Ref 33. Masana, J. J., Xiao, J., Zhang, H., Lu, X., Qiu, M., Yu, Y. Nitrogen-rich carbon nitride inducing electron delocalization of Co-N₄ site to enhance electrocatalytic carbon dioxide reduction. *Appl. Catal. B-Environ.* **323**, 122199 (2023).

Revision in the Supplementary Information

Page 21:

Fig. S18. The corresponding ELF plots of (a) TNO and (b) TNO_x@N.

Comment 6: Results and Discussion, Theoretical insight into the electron delocalization effect., paragraph 1

The explanation of PDOS and band gap results is not clear. You need to directly state that after N doping, the band gap narrows, TNO transforms from an insulator to a semiconductor, hence the carrier diffusion resistance decreases. Though the characterization is done beautifully, the key conclusion is unclear.

Response 6: Many thanks reviewer for this helpful suggestion. In the original manuscript, the calculated TNO bandgap was around 2.2 eV through the DFT calculation (Fig. 2d), indicating an insulating nature. By contrast, the electrical

conductivity of TNO_{-x}@N can be significantly improved because of the appearance of the impurity bands between the valence and conduction band, which is beneficial for the formation of free electrons and holes, leading to an increase in carrier concentration. This result is consistent with the ELF plot results.

According to the reviewer's suggestion, we supplemented the Ultraviolet–visible diffuse reflectance spectra (UV–vis DRS) to directly test the bandgap values of the TNO and TNO_{-x}@N. Based on the Tauc plots based on the Kubelka–Munk equation, as shown in **Figure R7**, the bandgap values of TNO and TNO_{-x}@N are 2.95 and 2.62 eV, respectively, which directly proves the band gap narrows and the carrier diffusion resistance is decreased after N doping. This agrees well with the DFT calculation and experiment results, but this value is larger than the bandgap calculated by the DFT in **Fig. 2d**, which is mainly attributed to the fact that the DFT theory itself tends to underestimate the energy gap.[21]

Figure R7. (a) UV–vis. Absorption spectra of TNO and TNO_{-x}@N, (b) Relationship between $(\alpha h\nu)^2$ and photon energy demonstrating the band gap of TNO and TNO_{-x}@N.

References:

R21. Lu, X., Jian, Z., Fang, Z., Gu, L., Hu, Y.-S., Chen, W., Wang, Z., Chen, L. Atomic-scale investigation on lithium storage mechanism in TiNb_2O_7 . *Energy Environ. Sci.* **2011**, 4 (8), 2638-2644.

The detail has been added to the revised manuscript and Supplementary Information (**Supplementary Fig. 19**), which is highlighted in yellow as shown below,

Revision in the manuscript

Page 11:

In addition, the Ultraviolet–visible diffuse reflectance spectra (UV–vis DRS) are carried out to further estimate the bandgap values of the TNO and $\text{TNO}_{-x}\text{@N}$. Based on the Tauc plots based on the Kubelka–Munk equation, as shown in **Supplementary Fig. 19**, the bandgap values of TNO and $\text{TNO}_{-x}\text{@N}$ are 2.95 and 2.62 eV, respectively, which directly proves the band gap narrows and the carrier diffusion resistance is decreased after N doping. This agrees well with the DFT calculation, but this value is larger than the bandgap calculated by the DFT, which is mainly attributed to the fact that the DFT theory itself tends to underestimate the energy gap.^[34]

References:

R34. Lu, X., Jian, Z., Fang, Z., Gu, L., Hu, Y.-S., Chen, W., Wang, Z., Chen, L. Atomic-scale investigation on lithium storage mechanism in TiNb_2O_7 . *Energy Environ. Sci.* **4**, 2638-2644 (2011).

Revision in the Supplementary Information

Page 22:

Fig. S19. Ultraviolet–visible diffuse reflectance spectra (UV–vis DRS) of TNO and TNO_x@N. (a) UV–vis. Absorption spectra of TNO and TNO_x@N, (b) Relationship between $(\alpha h\nu)^2$ and photon energy demonstrating the band gap of TNO and TNO_x@N.

Comment 7: Results and Discussion, Temperature-dependent electrochemical performance., paragraph 2

Why is there an upward trend in specific capacity under the same rate performance, as shown in Fig. 4b. For instance, at 20 C, the discharge capacity increases during the 35th~40th cycled, while this phenomenon does not occur at a low rate.

Response 7: We thank the reviewer for the in-depth comments. It's our negligence that the discussion and explanation for the discharge capacity increases during the 35th~40th cycled at 20 C is deficient in **Fig. 4b** of the original manuscript. The capacity increases can be ascribed to the initial SEI layer breakup, followed by the re-formation of a suitable film for high rates.[22,23] Meanwhile, the increase in capacity can be due to the concentration polarization at the high current density, resulting in the inability to deintercalation active lithium during the initial process. In addition, we found that the majority of the Nb-based oxides possess a capacity increase trend at high current densities (details are shown in **Table R8**)[24-32], which may also be related to the intrinsic properties and pseudo-capacitive behavior of the Nb-based oxides.

Figure R8. Previously reported rate performance of Nb-based oxides.

References:

- R22. Su, H., Ma, Y., Zhao, Z., Yang, D., Wang, H., Zhang, J., Li, D. Anchoring ternary CoNiSn alloys nanoparticles on hollow architected SnO₂ for exceptional lithium storage performance. *J. Power Sources* **450**, 227626 (2020).
- R23. He, Y., Li, A., Dong, C., Li, C., Xu, L. Mesoporous tin-based oxide nanospheres/reduced graphene composites as advanced anodes for lithium-ion half/full cells and sodium-ion batteries. *Chem. Eur. J.* **23**, 13724-13733 (2017).
- R24. Park, H., Shin, D. H., Song, T., Park, W. I., Paik, U. Synthesis of hierarchical porous TiNb₂O₇ nanotubes with controllable porosity and their application in high power Li-ion batteries. *J. Mater. Chem. A* **5**, 6958-6965 (2017).
- R25. Yang, C., Ma, D., Yang, J., Manawan, M., Zhao, T., Feng, Y., Li, J., Liu, Z., Zhang, Y.-W., Von Dreele, R. B., Toby, B. H., Albarrán, C. P. d. L., Pan, J. H. Crystallographic insight of reduced lattice volume expansion in mesoporous Cu²⁺-doped TiNb₂O₇ microspheres during Li⁺ insertion. *Adv. Funct. Mater.* **33**, 2212854 (2023).
- R26. Li, H., Zhang, Y., Tang, Y., Zhao, F., Zhao, B., Hu, Y., Murat, H., Gao, S., Liu, L. TiNb₂O₇ nanowires with high electrochemical performances as anodes for lithium ion batteries. *Appl. Surf. Sci.* **475**, 942-946 (2019).
- R27. Ji, X., Yang, Y., Ding, Y., Lu, Z., Liu, G., Liu, Y., Song, J., Yang, Z., Liu, X. Fluorine-doped

carbon-coated mesoporous $\text{Ti}_2\text{Nb}_{10}\text{O}_{29}$ microspheres as a high-performance anode for lithium-ion batteries. *J. Phys. Chem. C* **126**, 7799-7808 (2022).

R28. Zhao, L., Wang, S., Dong, Y., Quan, W., Han, F., Huang, Y., Li, Y., Liu, X., Li, M., Zhang, Z., Zhang, J., Tang, Z., Li, J. Coarse-grained reduced $\text{Mo}_x\text{Ti}_{1-x}\text{Nb}_2\text{O}_{7+y}$ anodes for high-rate lithium-ion batteries. *Energy Storage Mater.* **34**, 574–581 (2021).

R29. Park, H., Song, T., Paik, U. Porous TiNb_2O_7 nanofibers decorated with conductive $\text{Ti}_{1-x}\text{Nb}_x\text{N}$ bumps as a high power anode material for Li-ion batteries. *J. Mater. Chem. A* **3**, 8590-8596 (2015).

R30. Cui, P., Zhang, P., Chen, X., Chen, X., Wan, T., Zhou, Y., Su, M., Liu, Y., Xu, H., Chu, D. Oxygen defect and Cl^- -doped modulated TiNb_2O_7 compound with high rate performance in lithium-ion batteries. *ACS Appl. Mater. Interfaces* **15**, 43745-43755 (2023).

R31. Wu, Z., Guo, M., Yan, Y., Dou, H., Zhao, W., Zhang, Y., Li, S., Wu, J., Bin, X., Zhao, X., Yang, X., Ruan, D. Reducing crystallinity of micrometer-sized titanium–niobium oxide through cation substitution for high-rate lithium storage. *ACS Sustainable Chem. Eng.* **9**, 7422-7430 (2021).

R32. Chen, H., Cheng, H., Liu, H., Hu, Y., Yuan, T., Dai, S., Liu, M., Hu, H. Design of phase interface and defect in niobium-nickel oxide for ultrafast Li-ion storage. *J. Mater. Sci. Technol.* **147**, 145-152 (2023).

Based on the above discussion, the detail has been added to the revised manuscript, which is highlighted in yellow as shown below,

Revision in the manuscript

Page 18:

At the high current density, the increase in capacity can be due to the concentration polarization, resulting in the inability to deintercalation active lithium during the initial process. In addition, we found that the majority of the Nb-based oxides possess a capacity increase trend at high current densities^[53, 61-63], which may also be related to the intrinsic properties and pseudo-capacitive behavior of the Nb-based oxides.

References:

R53. Zhao, L., Wang, S., Dong, Y., Quan, W., Han, F., Huang, Y., Li, Y., Liu, X., Li, M., Zhang, Z., Zhang, J., Tang, Z., Li, J. Coarse-grained reduced $\text{Mo}_x\text{Ti}_{1-x}\text{Nb}_2\text{O}_{7+y}$ anodes for high-rate lithium-ion batteries. *Energy Storage Mater.* **34**, 574–581 (2021).

R61. Park, H., Shin, D. H., Song, T., Park, W. I., Paik, U. Synthesis of hierarchical porous TiNb_2O_7 nanotubes with controllable porosity and their application in high power Li-ion batteries. *J. Mater.*

Chem. A **5**, 6958-6965 (2017).

R62. Yang, C., Ma, D., Yang, J., Manawan, M., Zhao, T., Feng, Y., Li, J., Liu, Z., Zhang, Y.-W., Von Dreele, R. B., Toby, B. H., Albarrán, C. P. d. L., Pan, J. H. Crystallographic insight of reduced lattice volume expansion in mesoporous Cu²⁺-doped TiNb₂O₇ microspheres during Li⁺ insertion. *Adv. Funct. Mater.* **33**, 2212854 (2023).

R63. Li, H., Zhang, Y., Tang, Y., Zhao, F., Zhao, B., Hu, Y., Murat, H., Gao, S., Liu, L. TiNb₂O₇ nanowires with high electrochemical performances as anodes for lithium ion batteries. *Appl. Surf. Sci.* **475**, 942-946 (2019).

Comment 8: Results and Discussion, Tracing the origin of rapid low-temperature dynamics for TNO @N. paragraph 3

Complete the D_{Li^+} of TNO in the manuscript, and use less qualitative description for things that can be quantitatively explained. What is the degree when using “larger than” in the manuscript?

Response 8: Thank the reviewer for the valuable comment. We have added the qualitative description for D_{Li^+} of TNO in the revised manuscript. During lithiation, the average D_{Li^+} value of TNO-_x@N is $4.0 \times 10^{-13} \text{ cm}^2 \text{ s}^{-1}$ at 25 °C, and the value during delithiation is $6.2 \times 10^{-13} \text{ cm}^2 \text{ s}^{-1}$, which is higher than the TNO ($2.9 \times 10^{-13} \text{ cm}^2 \text{ s}^{-1} / 4.5 \times 10^{-13} \text{ cm}^2 \text{ s}^{-1}$) and solid TNO ($1.4 \times 10^{-13} \text{ cm}^2 \text{ s}^{-1} / 1.6 \times 10^{-13} \text{ cm}^2 \text{ s}^{-1}$). As the temperature drops to -40 °C, the D_{Li^+} values decrease by only one order of magnitude, keeping at $1.6 \times 10^{-14} \text{ cm}^2 \text{ s}^{-1}$ (lithiation) and $2.6 \times 10^{-14} \text{ cm}^2 \text{ s}^{-1}$ (delithiation), which are significantly larger than those of TNO ($5.5 \times 10^{-15} \text{ cm}^2 \text{ s}^{-1} / 8.2 \times 10^{-15} \text{ cm}^2 \text{ s}^{-1}$), solid TNO ($4.4 \times 10^{-15} \text{ cm}^2 \text{ s}^{-1} / 6.4 \times 10^{-15} \text{ cm}^2 \text{ s}^{-1}$). Excitingly, the apparent Li⁺ diffusion coefficient of TNO-_x@N at -40 °C is still comparable to the recently reported some Nb-based anode materials (**Table R1**) at room temperature.

Table R1. Comparisons of apparent Li^+ diffusion coefficient (D_{Li^+}) of $\text{TNO}_x@\text{N}$ at $-40\text{ }^\circ\text{C}$ during lithiation with previously reported Nb-based anode materials at $25\text{ }^\circ\text{C}$.

Material	D_{Li^+} ($\text{cm}^2\text{ s}^{-1}$)	Test technique	Reference
$\text{TNO}_x@\text{N}$ microflowers	4.0×10^{-13} ($25\text{ }^\circ\text{C}$)	GITT	This work
$\text{TNO}_x@\text{N}$ microflowers	1.6×10^{-14} ($-40\text{ }^\circ\text{C}$)	GITT	This work
TNO microflowers	5.5×10^{-15} ($-40\text{ }^\circ\text{C}$)	GITT	This work
Solid TNO	4.4×10^{-15} ($-40\text{ }^\circ\text{C}$)	GITT	This work
$\text{Mo}_{1.5}\text{W}_{1.5}\text{Nb}_{14}\text{O}_{44}$ micron-sized particles	7.7×10^{-18} ($25\text{ }^\circ\text{C}$)	GITT	[33]
N- Nb_2O_5 microflowers	2.4×10^{-16} ($25\text{ }^\circ\text{C}$)	EIS	[34]
$\text{Cr}_{0.6}\text{Ti}_{0.8}\text{Nb}_{10.6}\text{O}_{29}$ micron-sized	1.4×10^{-14} ($25\text{ }^\circ\text{C}$)	EIS	[35]
$\text{VNb}_9\text{O}_{25}$ nanoribbons	5.2×10^{-15} ($25\text{ }^\circ\text{C}$)	EIS	[36]
L-NbO sphere-like	7.3×10^{-15} ($25\text{ }^\circ\text{C}$)	GITT	[37]
$\text{GeNb}_{18}\text{O}_{47}$ nanowires	1.6×10^{-14} ($25\text{ }^\circ\text{C}$)	CV	[38]
m-TNO@C nanoparticles	5.7×10^{-17} ($25\text{ }^\circ\text{C}$)	CV	[39]
$\text{TiCr}_{0.5}\text{Nb}_{10.5}\text{O}_{29}$ nanoparticles	2.0×10^{-14} ($25\text{ }^\circ\text{C}$)	CV	[40]
$\text{Cr}_{0.5}\text{Nb}_{24.5}\text{O}_{62}$ nanowires	4.6×10^{-14} ($25\text{ }^\circ\text{C}$)	EIS	[41]
Nb_2O_5 nanorods	3.6×10^{-17} ($25\text{ }^\circ\text{C}$)	CV	[42]

References:

R33. Tao, R.; Zhang, T.; Tan, S.; Jafta, C. J.; Liang, J.; Sun, X.-G.; Wang, T.; Fan, J.; Lu, Z.; Bridges, C. A.; Dai, S. Insight into the fast-rechargeability of a novel $\text{Mo}_{1.5}\text{W}_{1.5}\text{Nb}_{14}\text{O}_{44}$ anode material for high-performance lithium-ion batteries. *Adv. Energy Mater.* **12**, 2200519 (2022).

- R34. Liu, G., Liu, S., Chen, H., Liu, X., Luo, X., Li, X., Ma, J. Highly [001]-oriented N-doped orthorhombic Nb₂O₅ microflowers with intercalation pseudocapacitance for lithium-ion storage. *Nanoscale* **14**, 11710-11718 (2022).
- R35. Yang, C., Yu, S., Ma, Y., Lin, C., Xu, Z., Zhao, H., Wu, S., Zheng, P., Zhu, Z.-Z., Li, J., Wang, N. Cr³⁺ and Nb⁵⁺ co-doped Ti₂Nb₁₀O₂₉ materials for high-performance lithium-ion storage. *J. Power Sources* **360**, 470-479 (2017).
- R36. Qian, S., Yu, H., Yan, L., Zhu, H., Cheng, X., Xie, Y., Long, N., Shui, M., Shu, J. High-rate long-life pored nanoribbon VNb₉O₂₅ built by interconnected ultrafine nanoparticles as anode for lithium-ion batteries. *ACS Appl. Mater. Interfaces* **9**, 30608-30616 (2017).
- R37. Zheng, Y., Yao, Z., Shadike, Z., Lei, M., Liu, J., Li, C. Defect-concentration-mediated T-Nb₂O₅ anodes for durable and fast-charging Li-ion batteries. *Adv. Funct. Mater.* **32**, 2107060 (2022).
- R38. Ran, F., Cheng, X., Yu, H., Zheng, R., Liu, T., Li, X., Ren, N., Shui, M., Shu, J. Nano-structured GeNb₁₈O₄₇ as novel anode host with superior lithium storage performance. *Electrochim. Acta* **282**, 634-641 (2018).
- R39. Qian, R., Yang, C., Ma, D., Li, K., Feng, T., Feng, J., Pan, J. H. Robust lithium storage of block copolymer-templated mesoporous TiNb₂O₇ and TiNb₂O₇@C anodes evaluated in half-cell and full-battery configurations. *Electrochim. Acta* **379**, 138179 (2021).
- R40. Hu, L., Lu, R., Tang, L., Xia, R., Lin, C., Luo, Z., Chen, Y., Li, J. TiCr_{0.5}Nb_{10.5}O₂₉/CNTs nanocomposite as an advanced anode material for high-performance Li⁺-ion storage. *J. Alloys Compd.* **732**, 116-123 (2018).
- R41. Yang, C., Yu, S., Lin, C., Lv, F., Wu, S., Yang, Y., Wang, W., Zhu, Z.-Z., Li, J., Wang, N., Guo, S. Cr_{0.5}Nb_{24.5}O₆₂ nanowires with high electronic conductivity for high-rate and long-life lithium-ion storage. *ACS Nano* **11**, 4217-4224 (2017).
- R42. Shi, C., Xiang, K., Zhu, Y., Zhou, W., Chen, X., Chen, H. Box-implanted Nb₂O₅ nanorods as superior anode materials in lithium ion batteries. *Ceram. Inter.* **43**, 12388-12395 (2017).

The detail has been added to the revised manuscript and Supplementary Information (**Supplementary Table 7**), which is highlighted in yellow as shown below,

Revision in the manuscript

Page 22:

To better assess the intrinsic ion-transport capability of TNO_{-x}@N, the galvanostatic intermittent titration techniques (GITT) are performed by measuring the macroscopic diffusion coefficients (D_{Li^+} , details in the **Experimental Section**). During lithiation, the

average D_{Li^+} value of TNO- x @N is $4.0 \times 10^{-13} \text{ cm}^2 \text{ s}^{-1}$ at 25 °C, and the value during delithiation is $6.2 \times 10^{-13} \text{ cm}^2 \text{ s}^{-1}$ (**Supplementary Fig. 41**), which is higher than the TNO ($2.9 \times 10^{-13} \text{ cm}^2 \text{ s}^{-1}/4.5 \times 10^{-13} \text{ cm}^2 \text{ s}^{-1}$) and solid TNO ($1.4 \times 10^{-13} \text{ cm}^2 \text{ s}^{-1}/1.6 \times 10^{-13} \text{ cm}^2 \text{ s}^{-1}$). As the temperature drops to -40 °C, the D_{Li^+} values decrease by only one order of magnitude, keeping at $1.6 \times 10^{-14} \text{ cm}^2 \text{ s}^{-1}$ (lithiation) and $2.6 \times 10^{-14} \text{ cm}^2 \text{ s}^{-1}$ (delithiation, **Fig. 5f**), which are significantly larger than those of TNO ($5.5 \times 10^{-15} \text{ cm}^2 \text{ s}^{-1}/8.2 \times 10^{-15} \text{ cm}^2 \text{ s}^{-1}$), solid TNO ($4.4 \times 10^{-15} \text{ cm}^2 \text{ s}^{-1}/6.4 \times 10^{-15} \text{ cm}^2 \text{ s}^{-1}$). Excitingly, the apparent Li^+ diffusion coefficient of TNO- x @N at -40 °C is still comparable to the recently reported some Nb-based anode materials (**Supplementary Table 7**) at room temperature.

Revision in the Supplementary Information

Page 35:

Table S7. Comparisons of apparent Li^+ diffusion coefficient (D_{Li^+}) of TNO- x @N at -40 °C during lithiation with previously-reported M-Nb-O anode materials at 25 °C.

Material	D_{Li^+} ($\text{cm}^2 \text{ s}^{-1}$)	Test technique	Reference
TNO- x @N microflowers	4.0×10^{-13} (25 °C)	GITT	This work
TNO- x @N microflowers	1.6×10^{-14} (-40 °C)	GITT	This work
TNO microflowers	5.5×10^{-15} (-40 °C)	GITT	This work
Solid TNO	4.5×10^{-15} (-40 °C)	GITT	This work
Mo _{1.5} W _{1.5} Nb ₁₄ O ₄₄ micron-sized particles	7.7×10^{-18} (25 °C)	GITT	[14]
N-Nb ₂ O ₅ microflowers	2.4×10^{-16} (25 °C)	EIS	[15]
Cr _{0.6} Ti _{0.8} Nb _{10.6} O ₂₉ micron-sized	1.4×10^{-14} (25 °C)	EIS	[16]

VNb ₉ O ₂₅ nanoribbons	5.2×10 ⁻¹⁵ (25 °C)	EIS	[17]
L-NbO sphere-like	7.3×10 ⁻¹⁵ (25 °C)	GITT	[18]
GeNb ₁₈ O ₄₇ nanowires	1.6×10 ⁻¹⁴ (25 °C)	CV	[19]
m-TNO@C nanoparticles	5.7×10 ⁻¹⁷ (25 °C)	CV	[20]
TiCr _{0.5} Nb _{10.5} O ₂₉ nanoparticles	2.0×10 ⁻¹⁴ (25 °C)	CV	[21]
Cr _{0.5} Nb _{24.5} O ₆₂ nanowires	4.6×10 ⁻¹⁴ (25 °C)	EIS	[22]
Nb ₂ O ₅ nanorods	3.6×10 ⁻¹⁷ (25 °C)	CV	[23]

References:

S14. Tao, R.; Zhang, T.; Tan, S.; Jafta, C. J.; Liang, J.; Sun, X.-G.; Wang, T.; Fan, J.; Lu, Z.; Bridges, C. A.; Dai, S. Insight into the fast-rechargeability of a novel Mo_{1.5}W_{1.5}Nb₁₄O₄₄ anode material for high-performance lithium-ion batteries. *Adv. Energy Mater.* **12**, 2200519 (2022).

S15. Liu, G., Liu, S., Chen, H., Liu, X., Luo, X., Li, X., Ma, J. Highly [001]-oriented N-doped orthorhombic Nb₂O₅ microflowers with intercalation pseudocapacitance for lithium-ion storage. *Nanoscale* **14**, 11710-11718 (2022).

S16. Yang, C., Yu, S., Ma, Y., Lin, C., Xu, Z., Zhao, H., Wu, S., Zheng, P., Zhu, Z.-Z., Li, J., Wang, N. Cr³⁺ and Nb⁵⁺ co-doped Ti₂Nb₁₀O₂₉ materials for high-performance lithium-ion storage. *J. Power Sources* **360**, 470-479 (2017).

S17. Qian, S., Yu, H., Yan, L., Zhu, H., Cheng, X., Xie, Y., Long, N., Shui, M., Shu, J. High-rate long-life pored nanoribbon VNb₉O₂₅ built by interconnected ultrafine nanoparticles as anode for lithium-ion batteries. *ACS Appl. Mater. Interfaces* **9**, 30608-30616 (2017).

S18. Zheng, Y., Yao, Z., Shadik, Z., Lei, M., Liu, J., Li, C. Defect-concentration-mediated T-Nb₂O₅ anodes for durable and fast-charging Li-ion batteries. *Adv. Funct. Mater.* **32**, 2107060 (2022).

S19. Ran, F., Cheng, X., Yu, H., Zheng, R., Liu, T., Li, X., Ren, N., Shui, M., Shu, J. Nano-structured GeNb₁₈O₄₇ as novel anode host with superior lithium storage performance. *Electrochim. Acta* **282**, 634-641 (2018).

S20. Qian, R., Yang, C., Ma, D., Li, K., Feng, T., Feng, J., Pan, J. H. Robust lithium storage of block copolymer-templated mesoporous TiNb₂O₇ and TiNb₂O₇@C anodes evaluated in half-cell and full-battery configurations. *Electrochim. Acta* **379**, 138179 (2021).

S21. Hu, L., Lu, R., Tang, L., Xia, R., Lin, C., Luo, Z., Chen, Y., Li, J. TiCr_{0.5}Nb_{10.5}O₂₉/CNTs nanocomposite as an advanced anode material for high-performance Li⁺-ion storage. *J. Alloys*

Compd. **732**, 116-123 (2018).

S22. Yang, C., Yu, S., Lin, C., Lv, F., Wu, S., Yang, Y., Wang, W., Zhu, Z.-Z., Li, J., Wang, N., Guo, S. Cr_{0.5}Nb_{24.5}O₆₂ nanowires with high electronic conductivity for high-rate and long-life lithium-ion storage. *ACS Nano* **11**, 4217-4224 (2017).

S23. Shi, C., Xiang, K., Zhu, Y., Zhou, W., Chen, X., Chen, H. Box-implanted Nb₂O₅ nanorods as superior anode materials in lithium ion batteries. *Ceram. Inter.* **43**, 12388-12395 (2017).

Comment 9: Results and Discussion, Temperature-dependent electrochemical performance., paragraph 5

The full battery assembled with NCM cathode has not provided key data on mass energy density and volume energy density, nor has it been compared with existing work. Therefore, it is impossible to judge the performance advantages of the battery from the perspective of commercial application from commercial application aspect.

Response 9: We're grateful to the reviewer for the valuable and professional suggestions, which really helps us improve the quality and value of this work. As the reviewer mentioned, the mass energy density and volume energy density of a full battery assembled with NCM cathode are key indicators for judging the commercial application aspect of the battery. To further confirm the commercial application potential of TNO_{-x}@N electrode, we supplemented the fabrication of the 3.5 Ah-level pouch cells under commercial conditions, pairing a commercial LiNi_{0.8}Co_{0.1}Mn_{0.1}O₂ as the cathode, as shown in **Figure R9a**. For the cathode electrode, the slurries consisting of LiNi_{0.8}Co_{0.1}Mn_{0.1}O₂ as the active materials, Kejten black and carbon nanotube as the conducting agent, and polyvinylidene fluoride (PVDF) binder in a weight ratio of 95.2:3:1.8 are prepared. For the anode electrode, the pastes consisting of LiNi_{0.8}Co_{0.1}Mn_{0.1}O₂ as the active materials, Kejten black and carbon nanotube as the

conducting agent, and PVDF binder in a weight ratio of 92:5:3 are prepared. The average mass loading of the single-sided anode and cathode electrode about are 15.4 and 16.1 mg cm⁻², respectively. In the voltage window (1.0–3.0 V, **Figure R9b**), the initial charge capacity of the TNO_{-x}@N-based pouch cell is 3.47 Ah at the current density of 0.2 C. Meanwhile, the pouch cell exhibits remarkable stability during cycling at 0.5 C, maintaining a capacity of 3.32 Ah after 55 cycles, with an impressive capacity retention ratio of 97.6% (**Figure R9c**), which demonstrates great potential for commercialization.

Figure R9. (a) Schematic illustration of the TNO_{-x}@N|LiNi_{0.8}Co_{0.1}Mn_{0.1}O₂ pouch cell; (b) Charge/discharge profiles of the TNO_{-x}@N|LiNi_{0.8}Co_{0.1}Mn_{0.1}O₂ pouch cell; (c) Cycling performance the TNO_{-x}@N|LiNi_{0.8}Co_{0.1}Mn_{0.1}O₂ pouch cell at 0.5 C after an activation process of 1 cycle at 0.2 C.

According to the reviewer's opinion, we compare our work and the representative literary/commercial Nb-based materials, Ti-based materials, carbon materials (graphite), and silicon-based materials in terms of the mass energy density and volumetric energy density (details are shown in **Table R2**). The mass energy density of TNO_{-x}@N-based pouch cell can be calculated by the formula: Mass energy density = Cell capacity × Average voltage/ Electrode weight, giving 271.6 Wh kg⁻¹. The volumetric energy density of TNO_{-x}@N-based pouch cell can be calculated by the equation: Volumetric energy density = Cell capacity × Average

voltage/(Thickness*Width*Length), yielding 373.4 Wh L⁻¹. It can be clearly seen from **Table R2** that the mass energy density and volumetric energy density of TNO_x@N anode are comparable to Nb-based materials, Ti-based materials, carbon materials (graphite), and silicon-based materials, showing certain potential for commercialization.

Table R2 Comparison of the mass energy density and volumetric energy density of the TNO_x@N electrode with recently reported anodes.

Materials	Particle size	Mass energy density (Wh kg ⁻¹)	Volumetric energy density (Wh L ⁻¹)	test types of Pouch or Coin	Refs
TNO _x @N	2~3 um	271.6	373.4	pouch	This work
XTO	/	84	176	pouch	Toshiba
Si-C/TiO ₂	5~15 μm	288.4	350.6	pouch	[43]
Graphite	/	270	/	pouch	[44]
Nb ₁ Mo _{0.1} O _{2.8}	1 um	129	146	Coin	[45]
Li ₄ Ti ₅ O ₁₂	/	110	100	Coin	[45]
Nb ₁₆ /Nb ₁₈	1 um	158	/	Coin	[46]
Graphite/SiN	2~3 um	/	412.2	pouch	[47]

References:

- R43. Xu, C., Shen, L., Zhang, W., Huang, Y., Sun, Z., Zhao, G., Lin, Y., Zhang, Q., Huang, Z., Li, J. Efficient implementation of kilogram-scale, high-capacity and long-life Si-C/TiO₂ anodes. *Energy Storage Mater.* **56**, 319-330 (2023).
- R44. Zheng, X., Cao, Z., Luo, W., Weng, S., Zhang, X., Wang, D., Zhu, Z., Du, H., Wang, X., Qie, L., Zheng, H., Huang, Y. Solvation and interfacial engineering enable -40 °C operation of graphite/NCM batteries at energy density over 270 Wh kg⁻¹. *Adv. Mater.* **35**, 2210115 (2023).
- R45. Shen, F., Sun, Z., Zhao, L., Xia, Y., Shao, Y., Cai, J., Li, S., Lu, C., Tong, X., Zhao, Y., Sun, J., Shao, Y. Triggering the phase transition and capacity enhancement of Nb₂O₅ for fast-charging

lithium-ion storage. *J. Mater. Chem. A* **9**, 14534-14544 (2021).

R46. Ma, J., Zhang, H., Yu, X., Xiang, Y., Qiu, J., Liu, S., Lin, H., Cao, G., Zhang, W. Regulating the local coordination model of homologous and heterogeneous niobium tungsten oxides toward ultrafast lithium storage. *Energy Storage Mater.* **63**, 102979 (2023).

R47. Chae, S., Park, S., Ahn, K., Nam, G., Lee, T., Sung, J., Kim, N., Cho, J. Gas phase synthesis of amorphous silicon nitride nanoparticles for high-energy LIBs. *Energy Environ. Sci.* **13**, 1212-1221 (2020).

The detail has been added to the revised manuscript and Supplementary Information (**Supplementary Fig. 35** and **Table 5**), which is highlighted in yellow as shown below,

Revision in the manuscript

Page 20:

Meanwhile, the 3.5 Ah-level pouch cells with commercial $\text{LiNi}_{0.8}\text{Co}_{0.1}\text{Mn}_{0.1}\text{O}_2$ as the cathode is further tested, as shown in **Supplementary Fig. 35**. The $\text{TNO}_{-x}\text{@N}|\text{LiNi}_{0.8}\text{Co}_{0.1}\text{Mn}_{0.1}\text{O}_2$ pouch cell shows remarkable cycling reliability at 0.5 C, with a capacity of 3.32 Ah after 55 cycles and an impressive capacity retention of 97.6%. The calculated mass energy density and volumetric energy density of $\text{TNO}_{-x}\text{@N}$ anode are obtained as 271.6 Wh kg^{-1} and 373.4 Wh L^{-1} , respectively, which are comparable to Nb-based materials, Ti-based materials, carbon materials (graphite), and silicon-based materials (**Supplementary Table 5**), showing certain potential for commercialization.

Revision in the Supplementary Information

Page 38:

Fig. S35. Electrochemical properties of TNO- x @N|LiNi $_{0.8}$ Co $_{0.1}$ Mn $_{0.1}$ O $_2$ Ah-level pouch cell. (a) Schematic illustration of the TNO- x @N|LiNi $_{0.8}$ Co $_{0.1}$ Mn $_{0.1}$ O $_2$ pouch cell; (b) Charge/discharge profiles of the TNO- x @N|LiNi $_{0.8}$ Co $_{0.1}$ Mn $_{0.1}$ O $_2$ pouch cell; (c) Cycling performance the TNO- x @N|LiNi $_{0.8}$ Co $_{0.1}$ Mn $_{0.1}$ O $_2$ pouch cell at 0.5 C after an activation process of 1 cycle at 0.2 C.

Page 52:

Table S5. Comparison of the mass energy density and volumetric energy density of the TNO- x @N electrode with recently reported some anodes.

Materials	Particle size	Mass energy density (Wh kg $^{-1}$)	Volumetric energy density (Wh L $^{-1}$)	test types of Pouch or Coin	Refs
TNO- x @N	2~3 μ m	271.6	373.4	pouch	This work
XTO	/	84	176	pouch	Toshiba
Si-C/TiO $_2$	5~15 μ m	288.4	350.6	pouch	[9]
Graphite	/	270	/	pouch	[10]
Nb $_1$ Mo $_{0.1}$ O $_{2.8}$	1 μ m	129	146	Coin	[11]
Li $_4$ Ti $_5$ O $_{12}$	/	110	100	Coin	[11]
Nb $_{16}$ /Nb $_{18}$	1 μ m	158	/	Coin	[12]
Graphite/SiN	2~3 μ m	/	412.2	pouch	[13]

References:

S9. Xu, C., Shen, L., Zhang, W., Huang, Y., Sun, Z., Zhao, G., Lin, Y., Zhang, Q., Huang, Z., Li, J. Efficient implementation of kilogram-scale, high-capacity and long-life Si-C/TiO $_2$ anodes. *Energy Storage Mater.* **56**, 319-330 (2023).

S10. Zheng, X., Cao, Z., Luo, W., Weng, S., Zhang, X., Wang, D., Zhu, Z., Du, H., Wang, X., Qie, L., Zheng, H., Huang, Y. Solvation and interfacial engineering enable -40 $^{\circ}$ C operation of

graphite/NCM batteries at energy density over 270 Wh kg⁻¹. *Adv. Mater.* **35**, 2210115 (2023).

S11. Shen, F., Sun, Z., Zhao, L., Xia, Y., Shao, Y., Cai, J., Li, S., Lu, C., Tong, X., Zhao, Y., Sun, J., Shao, Y. Triggering the phase transition and capacity enhancement of Nb₂O₅ for fast-charging lithium-ion storage. *J. Mater. Chem. A* **9**, 14534-14544 (2021).

S12. Ma, J., Zhang, H., Yu, X., Xiang, Y., Qiu, J., Liu, S., Lin, H., Cao, G., Zhang, W. Regulating the local coordination model of homologous and heterogeneous niobium tungsten oxides toward ultrafast lithium storage. *Energy Storage Mater.* **63**, 102979 (2023).

S13. Chae, S., Park, S., Ahn, K., Nam, G., Lee, T., Sung, J., Kim, N., Cho, J. Gas phase synthesis of amorphous silicon nitride nanoparticles for high-energy LIBs. *Energy Environ. Sci.* **13**, 1212-1221 (2020).

Comment 10: There are six spelling errors in the manuscript, for example, “in genral”, “infunces”, please correct them.

Response 10: We sincerely apologize for confusion caused by the spelling errors in the manuscript. The entire manuscript has been carefully checked and proofread, and the corresponding six spelling errors in the original manuscript have been corrected. The corresponding descriptions has been added in the revised manuscript and highlighted in yellow as shown below, such as:

Revision in the manuscript

Page 3:

In general, the apparent Li⁺ diffusion rate depends on the intrinsic structure and particle size.

Page 6:

The high-resolution N 1s spectra located at 399.1 eV (**Supplementary Fig. 3a**) confirms the valid nitrogen doping.

Page 13:

It can be seen that the relatively low diffusion barriers of TNO@N manifests the positive role of N-incorporation for Li^+ migration.

Page 17:

Lithium storage performance of $\text{TNO}_{-x}\text{@N}$, TNO, and solid TNO with high mass loading under ultralow temperature.

Page 23:

Another crucial factor of Low-T performance is the wettability influences.

Page 26:

...and reveals that the heterogeneous adjustment of the electronic structure of TiNb_2O_7 induces an impurity energy band at the Fermi energy level, lowering polaron hopping activation energy and increasing the free carrier and polaron concentration for improved electronic conductivity.

REVIEWER COMMENTS

Reviewer #1 (Remarks to the Author):

In the revised version, the authors addressed specific issues. The reviewer acknowledged that the prepared TNO-x@N outperforms both TNO and solid TNO. However, the reviewer still feels that novelty is insufficient for this journal. The author highlighted the novelty of the work, emphasizing its focus on low-temperature performance. Nevertheless, it was noted that only the results of the TNO-x@N half-cell test were presented. The reviewer recommends including a full-cell test and a TNO half-cell test at low temperatures to enhance the manuscript.

Reviewer #2 (Remarks to the Author):

The authors carefully addressed reviewer's comments. Therefore, I recommend an acceptance of this revised manuscript without a further revision.

Reviewer #3 (Remarks to the Author):

The authors have addressed some issues mentioned earlier, however, there are still some issues present that should be solved before we can make a decision.

(1) In the introduction part, the authors mentioned "conventional graphite anode suffers from the dramatic decline of ion diffusion...at conditions of both high rate...". Actually, the lithium ions diffusion efficiency quite high for graphite bulk, which is not the critical limit factor for fast charging.

(2) According to the XPS, both the N doping and O defects present in this TNO-138 x@N anode material. Which factor is the main contribution for the superior electrochemical performance? Please provide the electrochemical performance of the benchmark sample with just containing N doping feature or just O defects feature.

(3) What is the N doping position for TNO @N sample? Besides the theoretical calculations, is there some experiment characterization to conform the N-doping position?

(4) In Figure 4b, the TNO and TNO@N show different rate performance. Please clarify if the loading mass of TNO and TNO@N electrode are similar.

(5) Figure 4f demonstrates a cycling performance in low temperature. We suggestion the loading mass and the discharging rate should be same with Figure 4d.

(6) We notice the authors prepared TNO@N|LiNi_{0.8}Co_{0.1}Mn_{0.1}O₂ punch cell, please the energy density and low temperature cycling performance of this punch cell.

Reviewers' comments:

Reviewer #1 (Remarks to the Author):

In the revised version, the authors addressed specific issues. The reviewer acknowledged that the prepared TNO-x@N outperforms both TNO and solid TNO. However, the reviewer still feels that novelty is insufficient for this journal. The author highlighted the novelty of the work, emphasizing its focus on low-temperature performance. Nevertheless, it was noted that only the results of the TNO-x@N half-cell test were presented. The reviewer recommends including a full-cell test and a TNO half-cell test at low temperatures to enhance the manuscript.

Reviewer #2 (Remarks to the Author):

The authors carefully addressed reviewer's comments. Therefore, I recommend an acceptance of this revised manuscript without a further revision.

Reviewer #3 (Remarks to the Author):

The authors have addressed some issues mentioned earlier, however, there are still some issues present that should be solved before we can make a decision.

(1) In the introduction part, the authors mentioned "conventional graphite anode suffers from the dramatic decline of ion diffusion...at conditions of both high rate...". Actually, the lithium ions diffusion efficiency quite high for graphite bulk, which is not the critical limit factor for fast charging.

(2) According to the XPS, both the N doping and O defects present in this TNO-138 x@N anode material. Which factor is the main contribution for the superior electrochemical performance? Please provide the electrochemical performance of the benchmark sample with just containing N doping feature or just O defects feature.

(3) What is the N doping position for TNO @N sample? Besides the theoretical calculations, is there some experiment characterization to conform the N-doping position?

(4) In Figure 4b, the TNO and TNO@N show different rate performance. Please clarify if the loading mass of TNO and TNO@N electrode are similar.

(5) Figure 4f demonstrates a cycling performance in low temperature. We suggestion the loading mass and the discharging rate should be same with Figure 4d.

(6) We notice the authors prepared TNO@N|LiNi_{0.8}Co_{0.1}Mn_{0.1}O₂ punch cell,

please the energy density and low temperature cycling performance of this punch cell.

Response to Reviewer's Comments

(Manuscript ID: NCOMMS-23-54010A)

Response to Reviewer #1

Comment: In the revised version, the authors addressed specific issues. The reviewer acknowledged that the prepared TNO-x@N outperforms both TNO and solid TNO. However, the reviewer still feels that novelty is insufficient for this journal. The author highlighted the novelty of the work, emphasizing its focus on low-temperature performance. Nevertheless, it was noted that only the results of the TNO-x@N half-cell test were presented. The reviewer recommends including a full-cell test and a TNO half-cell test at low temperatures to enhance the manuscript.

Response: Thank you very much for your constructive comments improving our manuscript. We have studied the comments carefully and made corresponding corrections in the revised manuscript, and we sincerely hope the revised version is qualified for publication in the Nature Communications. Point-by-point response can be found below, and we hope our responses can well resolve your concerns and questions.

As the reviewer mentioned, a full-cell test and a TNO half-cell test at low temperatures are key indicators for improving the quality of the manuscript. According to the reviewer's suggestion, we supplemented the electrochemical properties of the 3.5 Ah-level pouch cells, TNO half-cell and solid TNO half-cell at low temperatures. In the pouch cell, the TNO-x@N is used as the anode, and the $\text{LiNi}_{0.8}\text{Co}_{0.1}\text{Mn}_{0.1}\text{O}_2$ is used as the cathode. For the cathode electrode, the slurries consisting of $\text{LiNi}_{0.8}\text{Co}_{0.1}\text{Mn}_{0.1}\text{O}_2$

as the active materials, Ketjen black & carbon nanotube as the conducting agent, and polyvinylidene fluoride (PVDF) binder in a weight ratio of 95.2:3:1.8 are prepared. The single-sided cathode electrode has a mass loading of approximately 16.1 mg cm^{-2} and a compaction density of 3.45 g cm^{-3} . For the anode electrode, the pastes consisting of $\text{TNO}_x@\text{N}$ as the active materials, Ketjen black & carbon nanotube as the conducting agent, and PVDF binder in a weight ratio of 92:5:3 are fabricated. The single-sided anode electrode has a mass loading of approximately 15.4 mg cm^{-2} and a compaction density of 2.30 g cm^{-3} . The N/P ratio of the pouch cell is set to 1.0, and the dimensions of the electrode are $4.6 \text{ cm} \times 8.0 \text{ cm}$. The amount of electrolyte was 14 mL. The 1C-rate is defined as 3.5 A at the voltage range of 1.0-3.0 V. **Figure R1a** shows the charge/discharge curves of the 3.5 Ah-level pouch cells at $-30 \text{ }^\circ\text{C}$. The initial charge capacity of the $\text{TNO}_x@\text{N}$ -based pouch cell is 2.15 Ah at the current density of 0.2 C. Meanwhile, the pouch cell exhibits remarkable stability during cycling at 0.5 C rate, maintaining a charge capacity of 2.05 Ah after 80 cycles, with a high capacity retention ratio of 95.7% (**Figure R1b**), further demonstrating the practical potential of $\text{TNO}_x@\text{N}$.

Figure R1. (a) Charge/discharge profiles of the $\text{TNO}_x@\text{N}|\text{LiNi}_{0.8}\text{Co}_{0.1}\text{Mn}_{0.1}\text{O}_2$ pouch cell at $-30 \text{ }^\circ\text{C}$; (b) Cycling performance the $\text{TNO}_x@\text{N}|\text{LiNi}_{0.8}\text{Co}_{0.1}\text{Mn}_{0.1}\text{O}_2$ pouch cell at 0.5 C and $-30 \text{ }^\circ\text{C}$ after an activation process of 1 cycle at 0.2 C. (1 C= 3.5 A)

As shown in **Figure R2a**, the initial discharge capacities of the $\text{TNO}_x@\text{N}$, TNO,

and solid TNO electrode are 236.2 mAh g⁻¹, 225.7 mAh g⁻¹ and 175.4 mAh g⁻¹ at 0.2 C and -20 °C. When the current rate is increased to 0.5 C, the reversible capacity of TNO_x@N retains 229.0 mAh g⁻¹ after 70 cycles with the capacity retention of 99.8%, which is preferable to that of the TNO (215.9 mAh g⁻¹, 98.8%) and solid TNO (141.3 mAh g⁻¹, 98.4%).

As the temperature drops to -30 °C, the 1st discharge capacities of the TNO_x@N, TNO, and solid TNO electrode are 225.2 mAh g⁻¹, 202.3 mAh g⁻¹ and 153.9 mAh g⁻¹ at 0.2 C (**Figure R2b**). As the current rate is increased to 0.5 C, the reversible capacity of TNO_x@N remains 215.4 mAh g⁻¹ after 70 cycles, which surpasses that of the TNO (203.5 mAh g⁻¹) and solid TNO (116.0 mAh g⁻¹).

Therefore, we believe that the synergistic effect between N-incorporation and O-vacancies can significantly improve the electrochemical performance of the TNO material at low temperatures, and providing a new direction for the construction of functional materials for fast-charging batteries for cold-region energy storage.

Figure R2. Cycling performance of the TNO_x@N, TNO, and solid TNO electrode at (a) -20 °C and (b) -30 °C after an activation process of 3 cycles at 0.2 C. The average mass loading of the electrodes is about 1.5 mg cm⁻². (1 C=300 mA g⁻¹)

The detail has been added to the revised manuscript and Supplementary

Information (**Supplementary Fig. 30** and **Fig. 37**), which is highlighted in yellow as shown below,

Revision in the manuscript

Page 20:

Low-T capacity and maintenance of the TNO_{-x}@N electrode at -20 °C and -30 °C are tested in **Supplementary Fig. 30**. At -20 °C, a stable capacity of 229.0 mAh g⁻¹ and an ultrahigh capacity retention of 99.8% are achieved after 70 cycles at 0.5 C, which is preferable to that of TNO flowers (215.9 mAh g⁻¹, 98.8%) and solid TNO (141.3 mAh g⁻¹, 98.4%). As the temperature drops to -30 °C, the reversible capacity of TNO_{-x}@N remains 215.4 mAh g⁻¹ after 70 cycles, which surpasses that of TNO (203.5 mAh g⁻¹) and solid TNO (116.0 mAh g⁻¹).

Revision in the manuscript

Page 21:

As the temperature further drops to -30 °C, the 3.5 Ah-level TNO_{-x}@N|LiNi_{0.8}Co_{0.1}Mn_{0.1}O₂ pouch cell still retains a capacity of 2.05 Ah after 80 cycles, with a capacity retention of 95.7% (**Supplementary Fig. 37**). The calculated mass energy density and volumetric energy density of TNO_{-x}@N anode are obtained as 168.4 Wh kg⁻¹ and 231.5 Wh L⁻¹, respectively, further demonstrating the practical potential of TNO_{-x}@N.

Revision in the Supplementary Information

Page 34:

Fig. S30. Electrochemical properties of the three electrodes at $-20\text{ }^{\circ}\text{C}$ and $-30\text{ }^{\circ}\text{C}$. Cycling performance of the $\text{TNO}_x\text{@N}$, TNO, and solid TNO electrode at (a) $-20\text{ }^{\circ}\text{C}$ and (b) $-30\text{ }^{\circ}\text{C}$ after an activation process of 3 cycles at 0.2 C . The average mass loading of the electrodes is about 1.5 mg cm^{-2} . ($1\text{ C}=300\text{ mA g}^{-1}$)

Revision in the Supplementary Information

Page 41:

Fig. S37. Electrochemical properties of $\text{TNO}_x\text{@N}|\text{LiNi}_{0.8}\text{Co}_{0.1}\text{Mn}_{0.1}\text{O}_2$ Ah-level pouch cell at $-30\text{ }^{\circ}\text{C}$. (a) Charge/discharge profiles; (b) Cycling performance at 0.5 C after an activation process of 1 cycle at 0.2 C . ($1\text{ C}=3.5\text{ A}$)

Overall, we are very grateful for the reviewer's constructive comments. Those comments are all valuable and very helpful for improving the paper's quality, as well as the important guiding significance to our research. We have revised the manuscript with our greatest effort. We sincerely hope the reviewer can give us a chance to publish the paper on Nature Communications. If you have any further questions or concerns, please feel free to give comments, and we will try our best to address them and improve the quality of the paper. Thank you!

Response to Reviewer #2

Comment: The authors carefully addressed reviewer's comments. Therefore, I recommend an acceptance of this revised manuscript without a further revision.

Response: We are grateful for the reviewer's positive recommendation in acceptance of our manuscript and insightful comments that help greatly improve the quality of the manuscript, as well as the important guiding significance to our research. Thank you again for your precious time and recommendations.

Response to Reviewer #3

Overall comment: The authors have addressed some issues mentioned earlier, however, there are still some issues present that should be solved before we can make a decision.

General response: We would like to express our gratitude for your meticulous review and valuable feedback on our work. Those comments are valuable and very meaningful for improving the paper's quality, as well as the important guiding significance to our research. We highly agree with the comments, and the corresponding revisions in the paper and the point-by-point responses to the reviewer's comments are listed as follows:

Comment 1: In the introduction part, the authors mentioned “conventional graphite anode suffers from the dramatic decline of ion diffusion...at conditions of both high rate...”. Actually, the lithium ions diffusion efficiency quite high for graphite bulk, which is not the critical limit factor for fast charging.

Response 1: We appreciate your sincere suggestions and acknowledge that there may be differences between our understanding of the critical limit factor of fast charging for graphite bulk. We agree with the reviewer that the lithium ions diffusion efficiency quite high for graphite bulk, and that is not the critical limiting factor for fast charging. At room temperatures, graphite anode has a good rate performance, and the industry can currently achieve 7 C fast charge and discharge. However, under the low temperatures and high rate conditions (especially at the charging rates of >3-5 C), the fast-charging capability of graphite presents obvious decreases due to the huge polarization, and even propels the intercalation potential of the graphite anode below 0 V (vs. Li⁺/Li), which makes it difficult to charge the batteries limited by the cut-off voltage. A more critical

concern is that the lithium plating on the graphite anode tends to become “dead lithium” on account of the lithium dendrite damage, leading to the loss of active lithium and rapid capacity decay^[1-4]. Therefore, the raised safety issue of Li plating on graphite anodes is a key limiting factor for fast charging especially at low temperatures, rendering the practical application of high-energy-density batteries in cold regions challenging.

References:

R1. Nan, B., Chen, L., Rodrigo, N. D., Borodin, O., Piao, N., Xia, J., Pollard, T., Hou, S., Zhang, J., Ji, X., Xu, J., Zhang, X., Ma, L., He, X., Liu, S., Wan, H., Hu, E., Zhang, W., Xu, K., Yang, X.-Q., Lucht, B., Wang, C. Enhancing Li⁺ transport in NMC811||graphite lithium-ion batteries at low temperatures by using low-polarity-solvent electrolytes. *Angew. Chem. Int. Ed.* **61**, e202205967 (2022).

R2. Huang, Y., Wang, C., Lv, H., Xie, Y., Zhou, S., Ye, Y., Zhou, E., Zhu, T., Xie, H., Jiang, W., Wu, X., Kong, X., Jin, H., Ji, H. Bifunctional interphase promotes Li⁺ de-solvation and transportation enabling fast-charging graphite anode at low temperature. *Adv. Mater.* **36**, 2308675 (2024).

R3. Gao, Y., Pan, Z., Sun, J., Liu, Z., Wang, J. High-energy batteries: beyond lithium-ion and their long road to commercialisation. *Nano-Micro Lett.* **14**, 94 (2022).

R4. Hubble, D., Brown, D. E., Zhao, Y., Fang, C., Lau, J., McCloskey, B. D., Liu, G. Liquid electrolyte development for low-temperature lithium-ion batteries. *Energy Environ. Sci.* **15**, 550-578 (2022).

Based on the above discussion, the detail has been added to the revised manuscript, which is highlighted in yellow as shown below,

Revision in the manuscript

Page 3:

Particularly, the fast-charging capability of conventional graphite presents obvious decreases under the low temperatures and high rate conditions due to the huge polarization, and even propels the intercalation potential of the graphite anode below 0 V (vs. Li⁺/Li), which makes it difficult to charge the batteries limited by the cut-off

voltage. A more critical concern is that the lithium plating on the graphite anode tends to become “dead lithium” on account of the lithium dendrite damage, leading to the loss of active lithium and rapid capacity decay, rendering the practical application of high-energy-density batteries in cold regions challenging^[9,10].

References:

Ref 9. Huang, Y., Wang, C., Lv, H., Xie, Y., Zhou, S., Ye, Y., Zhou, E., Zhu, T., Xie, H., Jiang, W., Wu, X., Kong, X., Jin, H., Ji, H. Bifunctional interphase promotes Li⁺ de-solvation and transportation enabling fast-charging graphite anode at low temperature. *Adv. Mater.* **36**, 2308675 (2024).

Ref 10. Hubble, D., Brown, D. E., Zhao, Y., Fang, C., Lau, J., McCloskey, B. D., Liu, G. Liquid electrolyte development for low-temperature lithium-ion batteries. *Energy Environ. Sci.* **15**, 550-578 (2022).

Comment 2: According to the XPS, both the N doping and O defects present in this TNO-138 x@N anode material. Which factor is the main contribution for the superior electrochemical performance? Please provide the electrochemical performance of the benchmark sample with just containing N doping feature or just O defects feature.

Response 2: Many thanks reviewer for this helpful suggestion. As the reviewer mentioned, it is crucial which factor of the N doping and O defects is the main contribution for achieving superior electrochemical performance. We are very sorry that the preparation of samples containing only N doping features is difficult or even impossible because an absence of oxygen atmosphere (Ar or NH₃) is required during the nitrogen doping process, which would at the same time generate oxygen defects. According to the reviewer's suggestion, the TNO-x sample with just containing O defects feature are synthesized by calcining TNO under the Ar/H₂ atmosphere (700 °C for 1 h), which can also explain the main reason for the excellent electrochemical

properties of the $\text{TNO}_{-x}@\text{N}$ sample. From XRD in **Figure R1a**, the characteristic peaks of the TNO_{-x} sample are similar to the TNO and $\text{TNO}_{-x}@\text{N}$ sample, which can be well indexed as the monoclinic phase of TiNb_2O_7 (JCDPS No. 72-0116), confirming the formation of O-vacancies does not change the bulk structure of TiNb_2O_7 .

Figure R1 (a) XRD patterns of TNO, TNO_{-x} and $\text{TNO}_{-x}@\text{N}$; (b) High-resolution XPS O 1s spectra of TNO_{-x} and TNO; (c) EPR spectra of TNO_{-x} and TNO.

To investigate the successful synthesis of oxygen defects in TNO_{-x} , X-ray photoelectron spectroscopy are investigated. As displayed in **Figure R1b**, the O 1s peaks in TNO_{-x} are fitted with three peaks at 529.7 eV, 531.4 eV, and 532.9 eV, which are attributed to the lattice O (metal–oxygen bands), oxygen vacancies, and oxygen species in hydroxyl oxygen^[5,6], respectively. The peak area ratio of oxygen vacancies to lattice O for TNO_{-x} (0.28) is much higher than that of TNO (0.12), which indicates that the reducing atmosphere can effectively trigger the lattice oxygen loss. Further analysis based on electron paramagnetic resonance (EPR) confirm the XPS results. In **Figure R1c**, TNO_{-x} displays an obvious oxygen vacancy signal peak at $g=2.003$, whereas TNO exhibits only a slight resonance at this position, which further confirms the presence of bulk-phase oxygen vacancies in TNO_{-x} in addition to surface vacancies.

The cycling performances of the TNO, TNO_{-x} and $\text{TNO}_{-x}@\text{N}$ electrode are shown and compared in **Figure R2a**. The TNO_{-x} electrode delivers the reversible capacity of

251.6 mAh g⁻¹ after 100 cycles at 0.5 C with the capacity retention of 91.6%, which is preferred to that of pure TNO flower (250.4 mAh g⁻¹, 87.0%), but lower than that of TNO_{-x}@N (257.8 mAh g⁻¹, 94.1%). This result suggests that the O-vacancies feature probably not significantly improve the cycling stability of the TNO material, and therefore the N doping feature is the main reason for the excellent cycling performance.

Figure R2b further compares the fast charging capability of the TNO, TNO_{-x} and TNO_{-x}@N in the current density range from 0.2 C to 60 C. The TNO_{-x} electrode delivers a capacity of 179.1 mAh g⁻¹ at a high current density 60 C, which far exceeds that of TNO (113.6 mAh g⁻¹), approaching TNO_{-x}@N (186.7 mAh g⁻¹). This result indicates that the O-vacancies feature is the main contribution to the improvement of the rate performances of the TNO material, which may be due to O-vacancies can provide abundant active sites to accelerate near-surface reaction, leading to the more obvious intercalation pseudo-capacitance behavior. Therefore, the O-vacancies feature is the main contribution to the improved rate performances, whereas the N doping feature is the main factor in achieving excellent cycling performance.

Figure R2. (a) Cycling performance of the TNO, TNO_{-x} and TNO_{-x}@N electrode at 0.5 C; (b) Rate capability the TNO, TNO_{-x} and TNO_{-x}@N electrode. The mass loading of the electrode is about 1.5 mg cm⁻². (1 C=300 mA g⁻¹)

References:

R5. Chen, Z., Yu, Z., Wang, L., Huang, Y., Huang, H., Xia, Y., Zeng, S., Xu, R., Yang, Y., He, S., Pan, H., Wu, X., Rui, X., Yang, H., Yu, Y. Oxygen defect engineering toward zero-strain $V_2O_{2.8}$ @porous reticular carbon for ultrastable potassium storage. *ACS Nano* **17**, 16478-16490 (2023).

R6. Ou, G., Xu, Y., Wen, B., Lin, R., Ge, B., Tang, Y., Liang, Y., Yang, C., Huang, K., Zu, D., Yu, R., Chen, W., Li, J., Wu, H., Liu, L. M., Li, Y. Tuning defects in oxides at room temperature by lithium reduction. *Nat. Commun.* **9**, 1302 (2018).

The detail has been added to the revised manuscript and Supplementary Information (**Supplementary Fig. 26**), which is highlighted in yellow as shown below,

Revision in the manuscript

Page 19:

Additionally, the impacts of TNO_{-x} sample with just containing O defects feature on the battery performances are conducted. As shown in **Figure S26**, the TNO_{-x} electrode exhibits the specific capacity of 251.6 mAh g⁻¹ after 100 cycles at 0.5 C with the capacity retention of 91.6%. As the rate increased to 60 C, it delivers a capacity of 179.1 mAh g⁻¹, approaching TNO_{-x}@N (186.7 mAh g⁻¹), further revealing the O-vacancies feature is the main contribution to the improvement of the rate performances of the TNO material.

Revision in the Supplementary Information

Page 29:

Fig. S26. Structural characterization and electrochemical properties of TNO-x. (a) XRD patterns of TNO, TNO-x and TNO-x@N; (b) High-resolution XPS O 1s spectra of TNO and TNO-x; (c) EPR spectra of TNO and TNO-x; (d) Cycling performance of the TNO, TNO-x and TNO-x@N electrode at 0.5 C; (e) Rate capability the TNO, TNO-x and TNO-x@N electrode. The mass loading of the electrode is about 1.5 mg cm⁻². (1 C=300 mA g⁻¹)

From XRD in **Supplementary Fig. 26a**, the characteristic peaks of the TNO-x sample are similar to the TNO and TNO-x@N sample, which can be well indexed as the monoclinic phase of TiNb₂O₇ (JCDPS No. 72-0116), confirming the formation of O-vacancies does not change the bulk structure of TiNb₂O₇. To investigate the successful synthesis of oxygen defects in TNO-x, X-ray photoelectron spectroscopy are investigated. As displayed in **Supplementary Fig. 26b**, the O 1s peaks in TNO-x are fitted with three peaks at 529.7 eV, 531.4 eV, and 532.9 eV, which are attributed to the lattice O (metal-oxygen bands), oxygen vacancies, and oxygen species in hydroxyl oxygen, respectively. The peak area ratio of O vacancy to lattice O for TNO-x (0.28) is much higher than that of TNO (0.12), which indicates that the reducing atmosphere can effectively trigger the lattice oxygen loss. Further analysis based on electron paramagnetic resonance (EPR) confirm the XPS results. In **Supplementary Fig. 26c**, TNO-x displays an obvious oxygen vacancy signal peak at g=2.003, whereas TNO

exhibits only a slight resonance at this position, which further confirms the presence of bulk-phase oxygen vacancies in TNO_{-x} in addition to surface vacancies.

The cycling performances of the TNO, TNO_{-x} and TNO_{-x}@N electrode are compared and the results are shown in **Supplementary Fig. 26d**. The TNO_{-x} electrode delivers the specific capacity of 251.6 mAh g⁻¹ after 100 cycles at 0.5 C with the capacity retention of 91.6%, which is preferred to that of pure TNO flower (250.4 mAh g⁻¹, 87.0%), but lower than that of TNO_{-x}@N (257.8 mAh g⁻¹, 94.1%). This result suggests that the O-vacancies feature does not significantly improve the cycling stability of the TNO material, and therefore the N doping feature is the main reason for the excellent cycling performance. **Supplementary Fig. 26e** further compares the fast charging capability of the TNO, TNO_{-x} and TNO_{-x}@N electrode in the current density range from 0.2 to 60 C. The TNO_{-x} electrode delivers a capacity of 179.1 mAh g⁻¹ at a high current density 60 C, which far exceeds that of TNO (113.6 mAh g⁻¹), approaching TNO_{-x}@N (186.7 mAh g⁻¹). This result indicates that the O-vacancies feature is the main contribution to the improvement of the rate performances of the TNO material, which may be due to O-vacancies can provide abundant active sites to accelerate near-surface reaction, leading to the more obvious intercalation pseudo-capacitance behavior.

Comment 3: What is the N doping position for TNO@N sample? Besides the theoretical calculations, is there some experiment characterization to conform the N-doping position?

Response 3: We thank the reviewer for the in-depth comments, which really helps us improve the quality and value of this work. We apologize for not being able to

specifically state the N doping position for TNO@N sample in the original manuscript. Indeed, the introduction of N doping during the synthesis process is random, and it is impossible to determine the precise location of the N doping^[7,8]. The lowest energy point obtained by theoretical calculations only indicates the relative stability of the position in terms of thermodynamic substitution, suggesting that N doping has the highest probability at this position^[9-11]. As shown in **Supplementary Table 1**, the substitution sites show that N element is relatively stable when it occupies the tetra-coordination site, corresponding to the lowest substitution energy, indicating that N is preferentially doped at the oxygen site of the tetra-coordination. To further determine the content of N doping in TNO_x@N, we supplemented elemental analysis tests. The test results show that the content of N element is determined to be 0.15%. Therefore, we believe that N is most probably to occupy the oxygen site of the tetra-coordination, and other positions may be substituted, but it is a very low probability.

Table S1. The calculated relative energies of N doping at the different oxygen sites after structure optimization. (2C: Di-coordinated oxygen; 3C: Tri-coordinated oxygen; 4C: tetra-coordinated oxygen)

Site	Relative energy (eV)
2C-1	0.35
2C-2	0.71
2C-3	0.64
3C-1	0.41
3C-2	0.10
3C-3	1.31
3C-4	0.52
4C	0.00

References:

- R7. Chen, M., Zhou, L., Wang, T., Xia, H., Liu, H., Dou, S., Chou, S. Nitrogen as an anionic center/dopant for next-generation high-performance lithium/sodium-ion battery electrodes: key scientific issues, challenges and perspectives. *Adv. Funct. Mater.* **33**, 2214786 (2023).
- R8. Xiang, L., Xu, Q., Zhang, H., Geng, S., Cui, R., Xiao, T., Chen., P., Wu, L., Yu, W., Peng, H., Mai, Y., Sun, H. Ultrahigh-rate Na/Cl₂ batteries through improved electron and ion transport by heteroatom-doped bicontinuous-structured carbon. *Angew. Chem. Int. Ed.* **62**, e202312001 (2023).
- R9. Han, X., Zhao, L., Wang, J., Liang, Y., Zhang, J. Delocalized electronic engineering of Ni₅P₄ nanoroses for durable Li-O₂ batteries. *Adv. Mater.* **35**, 2301897 (2023).
- R10. Liang, S., Yu, Z., Ma, T., Shi, H., Wu, Q., Ci, L., Tong, Y., Wang, J., Xu, Z., Mechanistic insights into the structural modulation of transition metal selenides to boost potassium ion storage stability. *ACS Nano* **15**, 14697-14708 (2021).
- R11. Guo, Y., Zhang, C., Xin, S., Shi, J., Wang, W., Fan, M., Chang, Y. X., He, W. H., Wang, E., Zou, Y. G., Yang, X., Meng, F., Zhang, Y. Y., Lei, Z. Q., Yin, Y. X., Guo, Y. G. Competitive doping chemistry for nickel-rich layered oxide cathode materials. *Angew. Chem. Int. Ed.* **61**, e2021168 (2022).

The detail has been added to the revised manuscript, which is highlighted in yellow as shown below,

Revision in the manuscript

Page 11:

Density functional theory (DFT) calculations are performed to understand the N doping preference position and the delocalized electronic engineering on the kinetics nature of TNO_{-x}@N (Supplementary Fig. 16). As shown in Supplementary Table 1, the substitution sites show that N element is relatively stable when it occupies the tetra-coordination site, corresponding to the lowest substitution energy, indicating that N is preferentially doped at the oxygen site of the tetra-coordination.

Comment 4: In Figure 4b, the TNO and TNO@N show different rate performance.

Please clarify if the loading mass of TNO and TNO@N electrode are similar.

Response 4: We deeply appreciate this constructive comment. It's our negligence that the explanation of the loading mass of TNO and TNO@N electrode is deficient in the original manuscript. The loading mass of the TNO_x@N, TNO and solid TNO electrode are all similar at about 1.5 mg cm⁻².

The detail has been added to the revised manuscript, which is highlighted in yellow as shown below,

Revision in the manuscript

Page 18:

Fig. 4... b, rate capability of the three electrodes with the loading mass of about 1.5 mg cm⁻²...

Comment 5: Figure 4f demonstrates a cycling performance in low temperature. We suggestion the loading mass and the discharging rate should be same with Figure 4d.

Response 5: We thank the reviewer for this constructive comment. We agree with the reviewer's comment to the loading mass and the discharging rate in **Figure 4f** should be same with **Figure 4d**. According to the reviewer's suggestion, we supplemented the cycling performance of the TNO_x@N electrode with the mass loading of 10 mg cm⁻² operated at 1.0 C and -30 °C. As shown in Figure **R3**, the thick TNO_x@N electrode exhibits a stable capacity of 149.3 mA h g⁻¹ and an ultrahigh capacity retention of 92.9%

after 250 cycles at 1 C, further demonstrating the effective promotion of the electron delocalization effect on the Low-T performance.

Figure R3. Long-term cycling stability of the TNO_x@N electrode with the mass loading of 10 mg cm⁻² operated at 1 C and -30 °C.

The detail has been modified to the revised manuscript (**Fig. 4f**), which is highlighted in yellow as shown below,

Revision in the manuscript

Page 20:

Moreover, the thick TNO_x@N electrode exhibits a stable capacity of 149.3 mAh g⁻¹ and an ultrahigh capacity retention of 92.9% after 250 cycles at 1 C. (**Fig. 4f**).

Fig. 4... f, Cyclability of the TNO_x@N electrode with the mass loading of 10 mg cm⁻² operated at 1 C and -30 °C...

Comment 6: We notice the authors prepared TNO@N|LiNi_{0.8}Co_{0.1}Mn_{0.1}O₂ punch

cell, please the energy density and low temperature cycling performance of this pouch cell.

Response 6: We are delighted to receive this constructive comment, which really helps us improve the quality and value of this work. According to the reviewer's suggestion, we supplemented the energy density and low temperature cycling performance of the 3.5 Ah-level pouch cells employed a commercial $\text{LiNi}_{0.8}\text{Co}_{0.1}\text{Mn}_{0.1}\text{O}_2$ as the cathode. For the cathode electrode, the slurries consisting of $\text{LiNi}_{0.8}\text{Co}_{0.1}\text{Mn}_{0.1}\text{O}_2$ as the active materials, Kejten black & carbon nanotube as the conducting agent, and polyvinylidene fluoride (PVDF) binder in a weight ratio of 95.2:3:1.8 are prepared. The single-sided cathode electrode has a mass loading of approximately 16.1 mg cm^{-2} and a compaction density of 3.45 g cm^{-3} . For the anode electrode, the pastes consisting of $\text{TNO}_x@\text{N}$ as the active materials, Kejten black & carbon nanotube as the conducting agent, and PVDF binder in a weight ratio of 92:5:3 are fabricated. The single-sided anode electrode has a mass loading of approximately 15.4 mg cm^{-2} and a compaction density of 2.30 g cm^{-3} . The N/P ratio of the pouch cell is set to 1.0, and the dimensions of the electrode are $4.6 \text{ cm} \times 8.0 \text{ cm}$. The amount of electrolyte was 14 mL. The 1C-rate is defined as 3.5 A at the voltage range of 1.0-3.0 V. **Figure R4a** shows the charge/discharge curves of the 3.5 Ah-level pouch cells at $-30 \text{ }^\circ\text{C}$. The initial charge capacity of the $\text{TNO}_x@\text{N}$ -based pouch cell is 2.15 Ah at the current density of 0.2 C. Meanwhile, the pouch cell exhibits remarkable stability during cycling at 0.5 C rate, maintaining a charge capacity of 2.05 Ah after 80 cycles, with a high capacity retention ratio of 95.7% (**Figure R4b**), further demonstrating the practical potential of $\text{TNO}_x@\text{N}$. The mass energy density of

TNO_{-x}@N-based pouch cell can be calculated by the equation: mass energy density = cell capacity × average voltage/ electrode weight, giving 168.4 Wh kg⁻¹. The volumetric energy density of TNO_{-x}@N-based pouch cell can be calculated by the equation: volumetric energy density = cell capacity × average voltage/(thickness × width × length), giving 231.5 Wh L⁻¹.

Figure R4. (a) Charge/discharge profiles of the TNO_{-x}@N|LiNi_{0.8}Co_{0.1}Mn_{0.1}O₂ pouch cell at $-30\text{ }^{\circ}\text{C}$; (b) Cycling performance the TNO_{-x}@N|LiNi_{0.8}Co_{0.1}Mn_{0.1}O₂ pouch cell at 0.5 C and $-30\text{ }^{\circ}\text{C}$ after an activation process of 1 cycle at 0.2 C. (1 C=3.5 A)

The detail has been added to the revised manuscript and Supplementary Information (**Supplementary Fig. 37**), which is highlighted in yellow as shown below,

Revision in the manuscript

Page 21:

As the temperature further drops to $-30\text{ }^{\circ}\text{C}$, the 3.5 Ah-level TNO_{-x}@N|LiNi_{0.8}Co_{0.1}Mn_{0.1}O₂ pouch cell still retains a capacity of 2.05 Ah after 80 cycles, with a capacity retention of 95.7% (**Supplementary Fig. 37**). The calculated mass energy density and volumetric energy density of TNO_{-x}@N anode are obtained as 168.4 Wh kg⁻¹ and 231.5 Wh L⁻¹, respectively, further demonstrating the practical potential of TNO_{-x}@N.

Revision in the Supplementary Information

Page 41:

Fig. S37. Electrochemical properties of TNO-x@N|LiNi_{0.8}Co_{0.1}Mn_{0.1}O₂ Ah-level pouch cell at -30 °C. (a) Charge/discharge profiles; (b) Cycling performance at 0.5 C after an activation process of 1 cycle at 0.2 C. (1 C=3.5 A)

Overall, we are very grateful to the editor and reviewers for providing such constructive and insightful comments on our manuscript. We have meticulously incorporated these suggestions while preparing the revised version of our paper. We believe that these modifications have significantly enhanced the overall quality of this paper. We sincerely hope the editor and reviewers can give us a chance to publish the paper on Nature Communications. If further revision is needed, please do not hesitate to inform us, and we will try our best to address them and improve the quality of the paper. Thank you!

REVIEWERS' COMMENTS

Reviewer #3 (Remarks to the Author):

Some issues still have not be sovled completely, such as the position of N doping and the low energy density (168 Wh/kg). However, I think it is the time to accept it for publication considering the high quality and well organization of this manuscript.

Reviewers' comments:

Reviewer #3 (Remarks to the Author):

Some issues still have not be sovled completely, such as the position of N doping and the low energy density (168 Wh/kg). However, I think it is the time to accept it for publication considering the high quality and well organization of this manuscript.

Response to Reviewer's Comments

(Manuscript ID: NCOMMS-23-54010B)

Response to Reviewer #3

Comment: Some issues still have not been solved completely, such as the position of N doping and the low energy density (168 Wh/kg). However, I think it is the time to accept it for publication considering the high quality and well organization of this manuscript.

Response: Thank you very much for your positive recommendation in acceptance of our manuscript. Those comments are valuable and helpful for revising and improving our paper. We have studied the comments carefully and made corresponding corrections in the revised manuscript. A point-by-point response can be found below, and we hope our responses can well resolve your concerns and questions.

1) the position of N doping

Response 1: Many thanks for this constructive comment. We are so sorry that it is hard to specifically determine the N doping position of the $\text{TNO}_{-x}\text{@N}$ sample because the introduction of N is random during the ammonia calcination process^[1,2]. Indeed, a large number of previously-reported literature employ theoretical calculations to calculate the thermodynamically stable doping site that is considered to have the highest probability of substitution at that site^[3-9]. Therefore, we optimized the model with different positions of N substitution for O and calculated the substitution energies for different substitution positions in the original manuscript. As can be seen from **Supplementary Table 1**, the substitution energy is lowest when it occupies the tetra-

coordination site, suggesting that it is thermodynamically relatively stable at this substitution site. Based on this, we induce that the tetra-coordination is probably the N doped sites in the TNO- x @N sample .

Table S1. The calculated relative energies of N doping at the different oxygen sites after structure optimization. (2C: Di-coordinated oxygen; 3C: Tri-coordinated oxygen; 4C: tetra-coordinated oxygen)

Site	Relative energy (eV)
2C-1	0.35
2C-2	0.71
2C-3	0.64
3C-1	0.41
3C-2	0.10
3C-3	1.31
3C-4	0.52
4C	0.00

References:

- R1. Yan, L., Cheng, X., Yu, H., Zhu, H., Liu, T., Zheng, R., Zhang, R., Shui, M., Shu, J. Ultrathin $W_9Nb_8O_{47}$ nanofibers modified with thermal NH_3 for superior electrochemical energy storage. *Energy Storage Mater.* **14**, 159-168 (2018).
- R2. Chen, M., Zhou, L., Wang, T., Xia, H., Liu, H., Dou, S., Chou, S. Nitrogen as an anionic center/dopant for next-generation high-performance lithium/sodium-ion battery electrodes: key scientific issues, challenges and perspectives. *Adv. Funct. Mater.* **33**, 2214786 (2023).
- R3. Yang, Y., Shen, C., Sun, K., Mei, D., Liu, C. Enhanced surface charge localization over nitrogen-doped In_2O_3 for CO_2 hydrogenation to methanol with improved stability. *ACS Catal.* **13**, 6154–6168 (2023).
- R4. Kou, T., Smart, T., Yao, B., Chen, I., Thota, D., Ping, Y., Li, Y. Theoretical and experimental insight into the effect of nitrogen doping on hydrogen evolution activity of Ni_3S_2 in alkaline medium. *Adv. Energy Mater.* **8**, 1703538 (2018).
- R5. Cui, G., Zeng, Y., Wu, J., Guo, Y., Gu, X., Lou, X. Synthesis of nitrogen-doped KMn_8O_{16} with oxygen vacancy for stable zinc-ion batteries. *Adv. Sci.* **9**, 2106067 (2022).
- R6. Yang, Y., Zhu, H., Yang, F., Chen, D., Wen, Z., Wu, D., Ye, M., Zhang, Y., Zhao, J., Liu, Q., Lu, X., Gu, M., Li, C., He, W. Ten thousand-cycle ultrafast energy storage of wadsley-roth phase Fe–Nb oxides with a desolvation promoting interfacial layer. *Nano Lett.* **21**, 9675–9683 (2021).

R7. Han, X., Zhao, L., Wang, J., Liang, Y., Zhang, J. Delocalized electronic engineering of Ni₅P₄ nanoroses for durable Li-O₂ batteries. *Adv. Mater.* **35**, 2301897 (2023).

R8. Gu, X., Wang, J., Zhao, X., Jin, X., Jiang, Y., Dai, P., Wang, N., Bai, Z., Zhang, M., Wu, M. Engineered nitrogen doping on VO₂(B) enables fast and reversible zinc-ion storage capability for aqueous zinc-ion batteries. *J. Energy Chem.* **85**, 30–38 (2023).

R9. Sun, H., Fan, W., Li, Y., Cheng, X., Li, P., Hao, J., Zhao, X. Origin of improved visible photocatalytic activity of nitrogen/hydrogen codoped cubic In₂O₃: first-principles calculations. *Phys. Chem. Chem. Phys.* **13**, 1379–1385 (2011).

The detail has been added to the revised manuscript, which is highlighted in yellow as shown below,

Revision in the manuscript

Page 10:

As shown in **Supplementary Table 1**, the substitution energy is lowest when it occupies the tetra-coordination site (**Supplementary Fig. 16** and **Fig. 2a**), indicating that the tetra-coordination is probably the N-doped sites in the TNO_{-x}@N sample.

2) the low energy density (168 Wh/kg)

Response 2: We appreciate this meaningful comment. We agree with the reviewer that the mass-energy density of 168 Wh kg⁻¹ for the TNO_{-x}@N-based pouch cell at -30 °C is not high. Actually, it is much higher than commercial niobium-based batteries (84 Wh kg⁻¹, Toshiba) and Li₄Ti₅O₁₂ batteries (100 Wh kg⁻¹, GREE). Although not higher than graphite and silicon-based materials, it needs to be emphasized that niobium-based materials are potential candidates for power-type batteries, but not energy-type batteries. We believe that the comment can bring about consideration for this work, and we will focus on improving the low-temperature energy density of niobium-based materials in the future to facilitate their commercial applications.

Overall, we are very grateful for the reviewer's constructive comments and suggestions for reinforcing the quality of this manuscript, as well as the important guiding significance to our research. Thank you again for your precious time and recommendations.